# ZCWPW1 is recruited to recombination hotspots by PRDM9 and is essential for meiotic double strand break repair

**Daniel Wells[1,2†]\*, Emmanuelle Bitoun[1,2†]\*, Daniela Moralli[1], Gang Zhang[1], Anjali Hinch[1], Julia Jankowska[1], Peter Donnelly[1,2], Catherine Green[1], Simon R Myers[1,2]\***

[1]The Wellcome Centre for Human Genetics, Roosevelt Drive, University of Oxford, Oxford, United Kingdom; [2]Department of Statistics, University of Oxford, Oxford, United Kingdom

**Abstract** During meiosis, homologous chromosomes pair and recombine, enabling balanced segregation and generating genetic diversity. In many vertebrates, double-strand breaks (DSBs) initiate recombination within hotspots where PRDM9 binds, and deposits H3K4me3 and H3K36me3. However, no protein(s) recognising this unique combination of histone marks have been identified. We identified *Zcwpw1*, containing H3K4me3 and H3K36me3 recognition domains, as having highly correlated expression with *Prdm9*. Here, we show that ZCWPW1 has co-evolved with PRDM9 and, in human cells, is strongly and specifically recruited to PRDM9 binding sites, with higher affinity than sites possessing H3K4me3 alone. Surprisingly, ZCWPW1 also recognises CpG dinucleotides. Male *Zcwpw1* knockout mice show completely normal DSB positioning, but persistent DMC1 foci, severe DSB repair and synapsis defects, and downstream sterility. Our findings suggest ZCWPW1 recognition of PRDM9-bound sites at DSB hotspots is critical for synapsis, and hence fertility.

**\*For correspondence:**
wells@stats.ox.ac.uk (DW);
ebitoun@well.ox.ac.uk (EB);
myers@stats.ox.ac.uk (SRM)

†These authors contributed equally to this work

**Competing interests:** The authors declare that no competing interests exist.

## Introduction

Meiosis is a specialised cell division, producing haploid gametes essential for reproduction. Uniquely, during this process homologous maternal and paternal chromosomes pair and exchange DNA (recombine) before undergoing balanced independent segregation. Alongside de novo mutations, this generates all genetic diversity – providing a substrate on which natural selection can act.

In humans, mice, and likely many other vertebrates (*Baker et al., 2017*), the locations at which recombination occur are determined by the binding locations of PRDM9, which recognises a specific sequence motif encoded by its zinc finger array (*Baudat et al., 2010*; *Myers et al., 2010*; *Parvanov et al., 2010*). At these sites, SPO11 catalyses recombination-initiating double strand breaks (DSBs), a subset of which are repaired by recombination with the homologous chromosome. The sister chromatid can also provide a repair substrate, for example for DSBs on the X chromosome in males, and there is evidence that this also occurs at some autosomal hotspots (*Allen and Latt, 1976*; *Li et al., 2019b*; *Lu and Yu, 2015*).

In *Prdm9*-null mice, DSBs form at promoter regions rich in H3K4me3 (*Brick et al., 2012*), and fail to repair resulting in severe asynapsis, meiotic arrest at pachytene, and sterility in both males and females on the B6 background (*Hayashi et al., 2005*). We previously showed (*Davies et al., 2016*) that across DSB sites, increased binding of PRDM9 to the homologous chromosome aids synapsis and fertility in hybrid mice, implying an impact of binding downstream of DSB positioning. Mice for which most DSBs occur at sites where PRDM9 does *not* bind the homologue show widespread asynapsis (*Davies et al., 2016*; *Gregorova et al., 2018*).

**eLife digest** Sexual reproduction – that is, the combination of sex cells from two different individuals to produce an embryo – is one of the many mechanisms that have evolved to maintain genetic diversity. Most human cells contain 23 pairs of chromosomes, with each chromosome in a pair carrying either a paternal or maternal copy of the same gene. To form an embryo with the right number of chromosomes, each sex cell (the egg or sperm cell) must only contain one chromosome from each pair.

Sex cells are produced from parent cells containing two sets of paternal and maternal chromosomes: these cells then divide twice to form four sex cells which contain only one chromosome from each pair. Before the parent cell divides, a process known as 'recombination' takes place, which allows chromosomes in a pair to exchange bits of genetic information. This reshuffling ensures that each chromosome in a sex cell is unique. A protein called PRDM9 helps control which sections of genetic information are recombined by modifying proteins attached to the chromosomes, marking them as locations for exchange. The DNA at each of these sites is then broken and repaired using the genetic sequence of the chromosome it is paired with as a template, thus causing the two chromosomes to swap genes.

In 2019, a group of researchers found a set of genes in the testis of mice that are expressed at the same time as the gene for PRDM9. This suggested that another protein called ZCWPW1 is likely involved in recombination, but the precise role of this protein was unclear. To answer this question, Wells, Bitoun et al. – including many of the researchers involved in the 2019 study – examined human cells grown in the laboratory to determine where ZCWPW1 binds to in the chromosome.

This revealed that ZCWPW1 can be found at the same sites as PRDM9, which is responsible for bringing it there. Furthermore, cells from male mice lacking the gene for ZCWPW1 cannot complete the exchange of genetic information between chromosomes, meaning that the mice are infertile. As such, ZCWPW1 seems to connect location selection by PRDM9 to the DNA repair mechanisms needed for gene exchange between chromosomes.

Infertility is a significant issue for humans affecting as many as one in every six couples. Fertility is complex and many of the biological mechanisms involved are not fully understood. This work suggests that both PRDM9 and ZCWPW1 are key to the production of sex cells and may be worth investigating as factors that affect fertility in humans.

Following their formation, DSB sites are processed by resection, resulting in single-stranded DNA (ssDNA) that becomes decorated with DMC1 (*Hong et al., 2001*; *Neale and Keeney, 2006*; *Sehorn et al., 2004*). In wild-type (WT) mice, DMC1 foci start to appear in early zygotene cells. From mid-zygotene to early pachytene, as part of the recombinational repair process, DMC1 dissociates from the ssDNA and counts decrease until all breaks (except those on the XY chromosomes) are repaired at late pachytene (*Moens et al., 2002*). At DSB sites where the homologous chromosome is not bound by PRDM9, the DMC1 signal is strongly elevated (*Davies et al., 2016*), suggesting persistent DMC1 foci and consistent with delayed DSB repair at these sites. In addition, fewer homologous recombination events occur at these sites (*Hinch et al., 2019*; *Li et al., 2019b*) implying that (eventual) DSB repair may sometimes use a sister chromosome pathway. However, the underlying mechanism(s) by which PRDM9 effectively contributes to DSB repair are not yet known.

PRDM9 deposits both H3K4me3 and H3K36me3 histone methylation marks at the sites it binds, and this methyltransferase activity is essential for its role in DSB positioning (*Diagouraga et al., 2018*; *Powers et al., 2016*). What factors 'read' this unique combination of marks at recombination sites, however, is currently unknown. Notably, outside of hotspots and the pseudoautosomal region (PAR) on sex chromosomes, H3K4me3 and H3K36me3 occur at largely non-overlapping locations (*Powers et al., 2016*), suggesting potentially highly specialised reader(s). Indeed, in somatic cells H3K4me3 is deposited mainly at promoters, in particular by the SET1 complex targeted by CXXC1/CFP1 and Wdr82 which binds Ser5 phosphorylated polymerase II (*Barski et al., 2007*; *Lee and Skalnik, 2008*; *Lee and Skalnik, 2005*). H3K36me3 is deposited by different methyltransferases, including SETD2 bound to Ser2 phosphorylated (elongating) polymerase II, and is enriched at exon-bound

nucleosomes, particularly for 3' exons (reviewed in *McDaniel and Strahl, 2017*; *Wagner and Carpenter, 2012*). H3K36me3 has multiple important roles, including in directing DNA methylation by recruiting DNMT3B, somatic DSB repair by homologous recombination (*Aymard et al., 2014*; *Carvalho et al., 2014*; *Pfister et al., 2014*), mismatch repair by recruiting MSH6 (*Huang et al., 2018*; *Li et al., 2013*), and V(D)J recombination during lymphopoiesis (*Ji et al., 2019*).

Using single-cell RNA-sequencing (RNA-seq) of mouse testis, we identified a set of genes co-expressed in (pre)leptotene cells which are highly enriched for genes involved in meiotic recombination (*Jung et al., 2019*). *Zcwpw1*, which ranks 3$^{rd}$ in this set after *Prdm9* (2$^{nd}$), is of unknown function but contains two recognised protein domains: CW and PWWP, shown to individually bind H3K4me3 and H3K36me3 respectively (*He et al., 2010a*; *Rona et al., 2016*). This raises the attractive possibility that ZCWPW1 might recognise and physically associate with the same marks deposited by PRDM9 (*Jung et al., 2019*).

In humans, *ZCWPW1* is expressed in testis (*Carithers et al., 2015*; *Uhlén et al., 2015*; *Figure 1—figure supplement 1*). It is also one of 104 genes specifically expressed in meiotic prophase within murine fetal ovaries (*Soh et al., 2015*), further suggesting a conserved meiotic function. This was confirmed by a recent study showing that ZCWPW1 is required for male fertility in mice, with ZCWPW1 hypothesised to recruit the DSB machinery to hotspot sites (*Li et al., 2019a*). Here, we show that ZCWPW1 co-evolves with PRDM9 and is recruited to recombination hotspots by the combination of histone marks deposited by PRDM9. However ZCWPW1 is not in fact required for the positioning of DSBs at PRDM9-bound sites, which occurs normally in *Zcwpw1*-null mice. Instead, ZCWPW1 is required for proper inter-homologue interactions: synapsis and the repair of DSBs. In *Zcwpw1*-null mice, DMC1 dynamics are strongly perturbed, with signals at autosomal hotspots resembling those on the X-chromosome, which does not have a homologue. Thus, ZCWPW1 represents the first protein directly positioned by PRDM9's dual histone marks, but impacting homologous DSB repair, not positioning.

## Results

### *ZCWPW1* co-evolves with *PRDM9*

Based on publicly available databases, we identified likely ZCWPW1 orthologues in 167 species (Materials and methods, *Figure 1—source data 1*), aligned each to the human reference ZCWPW1 protein sequence, and compared against a previous analysis of PRDM9 (*Baker et al., 2017*). The regions containing the CW and PWWP domains are well-conserved (*Figure 1A,B*), while other parts of the protein appear absent in some species. Additionally, there is a region of moderate conservation downstream of the PWWP domain, not overlapping any known domain. Further, an SYCP1 (SCP1) domain is annotated in the mouse protein only which, although only suggestive, is notable given that SYCP1 physically connects homologous chromosomes in meiosis. The C-terminal end of ZCWPW1 may include a methyl-CpG-binding domain (Materials and methods, *Lobley et al., 2009*), at least in some species.

Incomplete currently available DNA and protein sequences mean we are almost certain to miss some species where either *ZCWPW1* and/or *PRDM9* (*Baker et al., 2017*) are present. Despite this, we see extremely high overlap between *PRDM9* and *ZCWPW1* occurrence (*Figure 1C,D*). *ZCWPW1* appears to mainly occur in species possessing *PRDM9* (and even more clearly in groups of species possessing *PRDM9*). Furthermore, with the exception of a small number of amphibians, *ZCWPW1* orthologues are only identified in clades previously identified to possess *PRDM9* (*Baker et al., 2017*), and the majority of species (78% of 167) possessing *ZCWPW1* also possess a close relative (species or family) where *PRDM9* is identified (*Figure 1C*).

We leveraged a previous analysis of *PRDM9* evolution (*Baker et al., 2017*) to extend this comparison (*Figure 1D*, *Figure 1—source datas 1* and *2*). Among seven clades previously shown to possess *PRDM9* copies with all N-terminal domains (KRAB, SSXRD and PR/SET domains) intact, *ZCWPW1* is found in each case, typically in the same species (*Figure 1D*). In some species with several *PRDM9* copies, including many teleost fish and two other confirmed independent events, *PRDM9* has lost N-terminal domain sequences in some but not all copies, while in at least three further species clades, particular N-terminal domains have been lost in *all PRDM9* copies, but the PR/SET domain

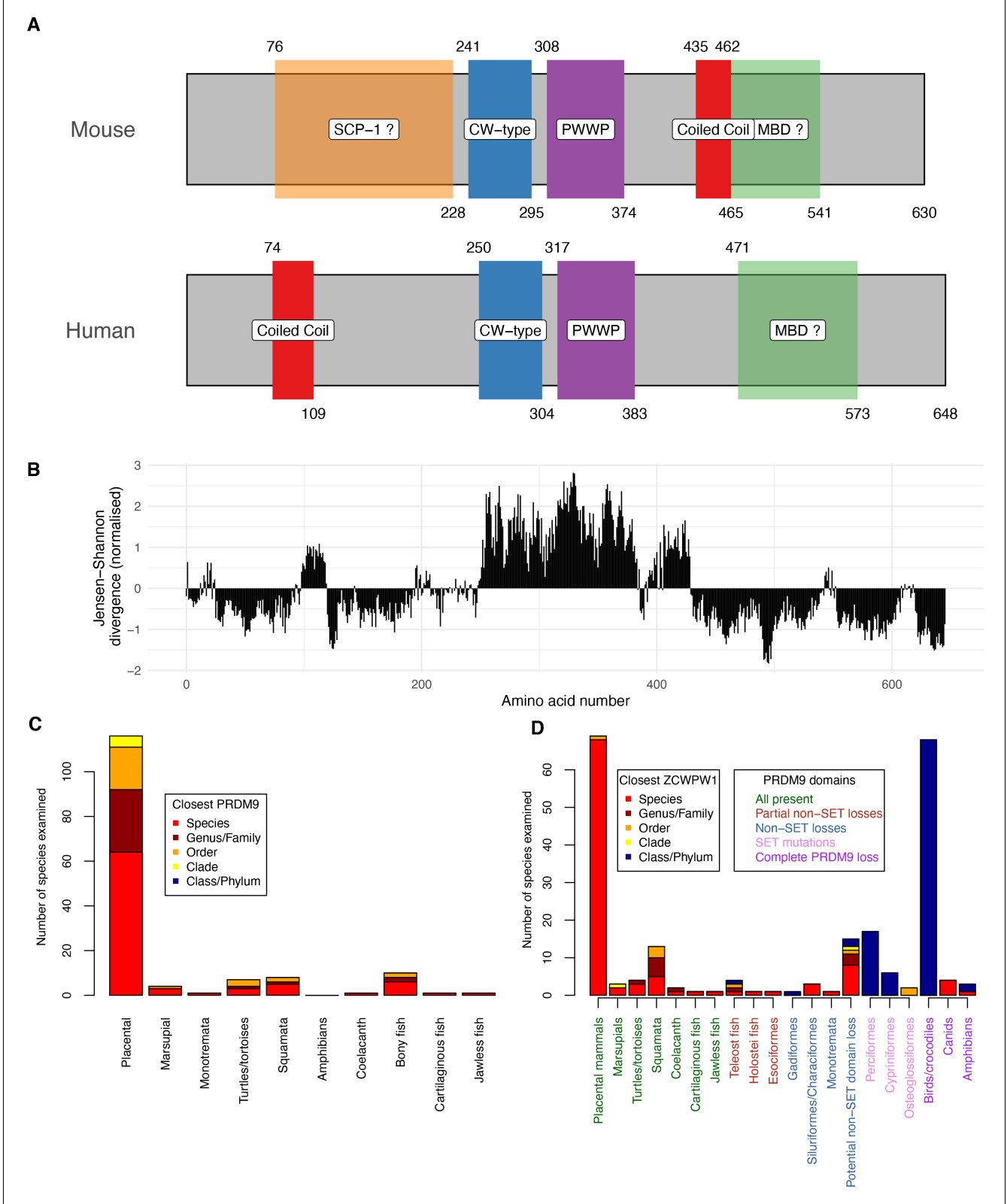

**Figure 1.** Domain organisation (**A**) and evolutionary conservation (**B**) of ZCWPW1, and co-evolution with PRDM9 (**C,D**). (**A**) Protein domains in the human and mouse proteins (source: UniProt). Amino acid start and end positions of each domain are shown above and below the rectangles, respectively. Prediction of SCP-1 (SYCP1) domain from **Marchler-Bauer and Bryant, 2004** and of MBDs (methyl-CpG binding domain) from **Lobley et al., 2009** (Materials and methods). (**B**) Conservation of human amino acids, normalised Jensen-Shanon divergence normalised to mean of 0

*Figure 1 continued on next page*

Figure 1 continued

and standard deviation of 1 is shown on the y-axis (a measure of sequence conservation, see *Capra and Singh, 2007* and *Johansson and Toh, 2010*) computed from using multiple alignment of 167 orthologues (Materials and methods). (C) All species we identified as possessing *ZCWPW1* copies were phylogenetically grouped into clades as previously (*Baker et al., 2017*) (x-axis) and each clade divided (stacked bars) according to whether *ZCWPW1*-possessing species within it also possess *PRDM9* ('Species', red) or instead their closest *PRDM9*-possessing relative is respectively in the same genus/family, order, clade or order/phylum, with colours as given in the 'Closest PRDM9' legend. (D) As (C), but now showing the closest relative possessing *ZCWPW1* ('Closest ZCWPW1' legend) for species possessing complete, partial or no identified *PRDM9* copies. As in (C), the x-axis groups species into clades, now further divided based on data from *Baker et al., 2017* into subclades according to the domains of PRDM9 lost or mutated across that subclade in all observed copies, reflecting multiple partial losses of particular PRDM9 domains, or complete loss of all *PRDM9* copies (Main text). The x-axis labels are ordered and coloured according to the PRDM9 domains present ('PRDM9 domains' legend, where 'SET' refers to PRDM9's PR/SET domain and the KRAB and SSXRD domains are grouped as 'non-SET', and 'partial' losses are seen in some but not all *PRDM9* copies in that species). Further details are presented in Materials and methods, and the raw data in *Figure 1—source datas 1* and *2*.

The online version of this article includes the following source data and figure supplement(s) for figure 1:

Source data 1. ZCWPW1 identified orthologues.
Source data 2. PRDM9 orthologues (*Baker et al., 2017*).
Figure supplement 1. *ZCWPW1* is specifically expressed in testis in humans.

remains intact. *ZCWPW1* is also found across these cases (*Figure 1D*), so it is widely conserved across species possessing one or more copies of *PRDM9* with an intact PR/SET domain.

Previously (*Baker et al., 2017*), three independent groups of species have been identified possessing mutations in the PR/SET domain of PRDM9, at key catalytic residues for the deposition of the H3K4me3/H3K36me3 mark(s). Complete loss of *PRDM9* is also identified in three groups: bird/crocodiles, canids, and (tentatively) amphibians. Given that ZCWPW1 is predicted to recognise the H3K4me3/H3K36me3 marks catalysed by the PR/SET domain of PRDM9, we would expect each of these six events to disrupt any potential interaction between the two proteins. Strikingly, we failed to identify *ZCWPW1* orthologues in any of the previously examined species impacted by these six events (*Figure 1D*), except in canids (and one amphibian species). Thus, at least four of the six may be completely mirrored by *ZCWPW1* loss. Even if *PRDM9* is retained, species with PR/SET domain mutations are much more likely not to possess an identified *ZCWPW1* orthologue within their class/phylum compared to species without such mutations (odds ratio = 285; indicative $p < 10^{-15}$ by FET, although we note that these observations are not all independent).

In conclusion, among vertebrates at least, it appears the set of species with *ZCWPW1* is extremely similar to those species possessing *PRDM9* with an intact PR/SET domain, with considerable evidence of co-evolution of gain/loss events for each protein. In some species with apparently intact PR/SET domains (for example Tasmanian Devil, Mexican Tetra, Platypus), it is hypothesised that PRDM9 does not position DSBs due to loss of other N-terminal domains and/or the PRDM9 zinc-finger array (*Baker et al., 2017*). These example species still possess identified *ZCWPW1* orthologues (*Figure 1—source data 1*), so it does not seem that PRDM9's ability to position DSBs is essential for retention of *ZCWPW1*. Instead, our results are consistent with ZCWPW1's main function involving recognition of the histone modifications catalysed by PRDM9-SET during early meiosis, in agreement with the functional evidence from ZCWPW1's CW and PWWP domains.

## Localisation of ZCWPW1 in meiosis and analysis of asynapsis in infertile *Zcwpw1*⁻/⁻ male mice

We previously detected *Zcwpw1* transcript expression in pre-leptotene to leptotene spermatocytes of adult WT mice (*Jung et al., 2019*). To investigate the role of ZCWPW1 during meiosis in vivo, we produced and validated the specificity of an antibody against the full-length recombinant mouse protein (*Figure 2—figure supplement 1*), and studied the phenotype of a newly generated knock-out (KO) mouse line for *Zcwpw1*, with a particular focus on fertility and meiotic recombination.

In testes from WT mice, we observe a dynamic localisation of ZCWPW1 protein (*Figure 2*), similar but non-identical to that reported in a recent study (*Li et al., 2019a*). ZCWPW1 shows a strong, punctate nuclear staining excluding the pericentromeric regions (clustered into chromocenters brightly stained with DAPI) in zygotene and early pachytene cells. In mid-pachytene cells, ZCWPW1 levels drop, with the protein now mainly localised in the XY body and as bright foci at the ends of the synaptonemal complex labeled by SYCP3, not previously observed using an antibody raised

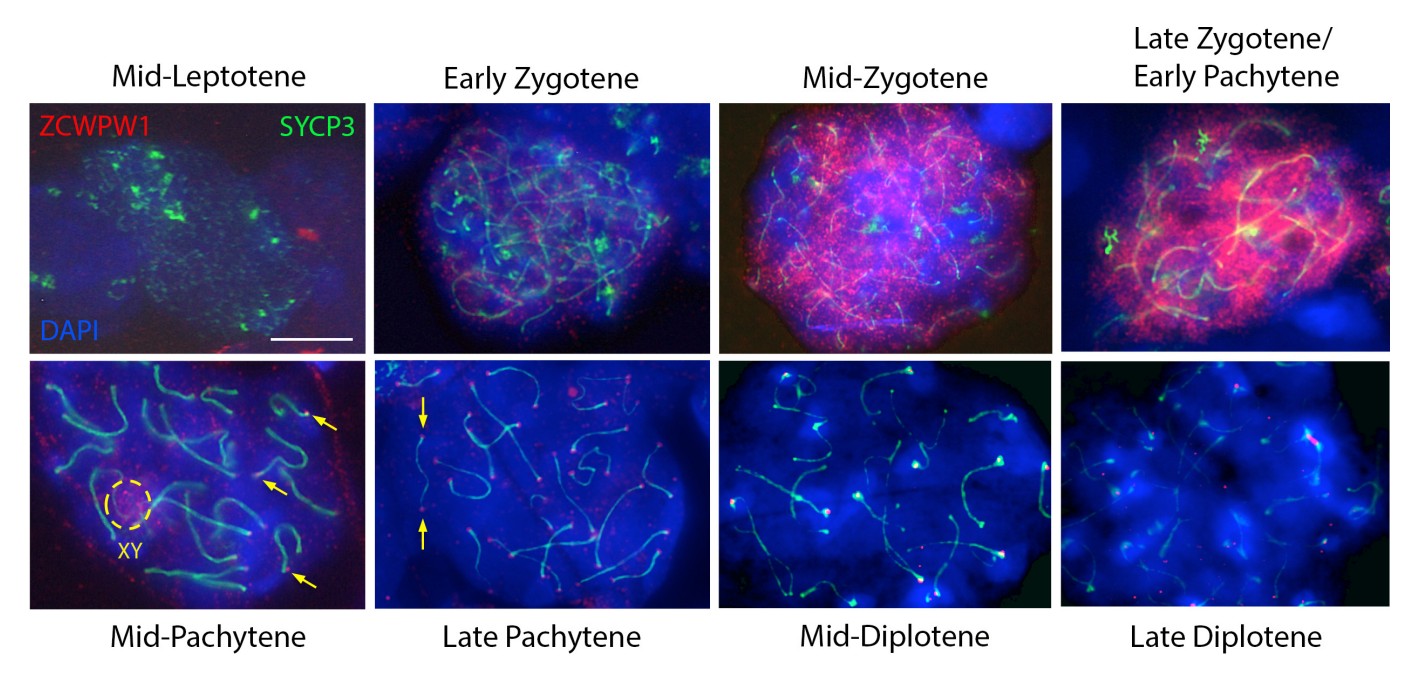

**Figure 2.** Expression of ZCWPW1 across meiosis prophase I in mouse testis. Nuclear spreads from 9 to 10 weeks old WT mice were immunostained with antibodies against ZCWPW1 (red) and the synaptonemal complex protein SYCP3 (green) which labels the chromosome axis, and counterstained with DAPI (blue) to visualise nuclei. Developmental stages are indicated above and below. Yellow arrows point to ZCWPW1 foci clearly visible at both ends of the synaptonemal complex in mid-late pachytene. Additional evidence is provided in *Figure 2—figure supplement 2*. The dashed circle shows staining in the XY body. Images for the individual channels are provided in *Figure 3—figure supplement 1*. These images are representative of the results obtained in three mice. Scale bar: 10 μm.

The online version of this article includes the following source data and figure supplement(s) for figure 2:

**Source data 1.** Immuno-FISH analysis of ZCWPW1 foci localisation at the synaptonemal complex ends in WT testis (mid-Pachytene to Late Diplotene cells).
**Figure supplement 1.** ZCWPW1 antibody generation and validation.
**Figure supplement 2.** ZCWPW1 localises to both subtelomeric and subcentromeric regions of chromosomes in pachytene cells.

against a 174 amino acid C-terminal region of the protein (*Li et al., 2019a*). By diplotene, little expression is visible. Using fluorescent in-situ hybridisation (FISH) to label centromeric and telomeric (distal and proximal to centromeres) regions of chromosomes, we established that these discrete foci of ZCWPW1 are consistently positioned at the ends of the synaptonemal complex, with the majority at both ends simultaneously, lying close to telomeres and centromeres themselves (*Figure 2—figure supplement 2*).

We next studied mice from a constitutive *Zcwpw1* KO line (Materials and methods), carrying a ~ 1.5 kb frameshift deletion encompassing exons 5 to 7 upstream of the CW and PWWP domains. This creates a premature stop codon that would produce a predicted heavily truncated 492bp (vs 1893bp for WT) transcript lacking either of these domains (*Figure 3A*), and a candidate for nonsense-mediated mRNA decay (*Kurosaki et al., 2019*). Indeed, we observed a complete lack of detectable ZCWPW1 protein expression in testis protein extracts (*Figure 3—figure supplement 1A*) and chromosome spreads from *Zcwpw1$^{-/-}$* mice, in zygotene cells (*Figure 3B*) and all other meiotic stages of prophase I where expression is detected in WT mice (*Figure 3—figure supplement 1B*). Confirming recent findings in a different *Zcwpw1$^{-/-}$* mouse (*Li et al., 2019a*), we observed no overt fertility phenotype in either sex in the heterozygous *Zcwpw1$^{+/-}$* mice (data not shown). However, *Zcwpw1$^{-/-}$* male mice were sterile with complete azoospermia and reduced testis size and weight (*Figure 3C,D*, *Figure 3—source data 1*), while female mice retained fertility until around 7–8 months of age (*Figure 3—source data 2*), and otherwise both sexes develop normally. As in *Li et al., 2019a*, in male mice we observe no meiotic progression beyond (pseudo)pachytene with widespread asynapsed chromosomes marked by γ-H2AX and HORMAD2, failure to form the sex body (*Figure 3B*, *Figure 3—figure supplement 2A*), a complete absence of MLH1 foci marking

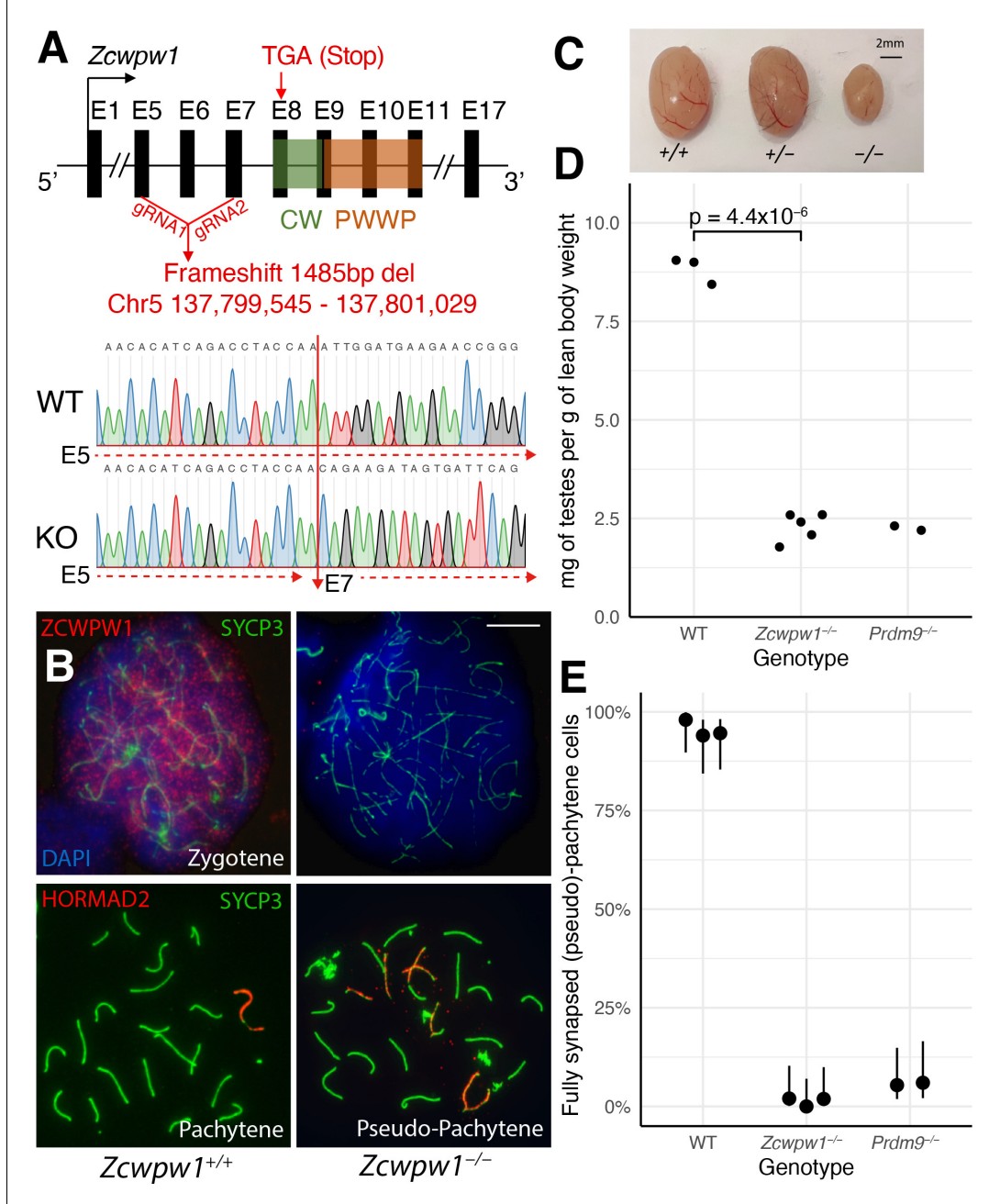

**Figure 3.** *Zcwpw1⁻/* male mice show reduced testis size and asynapsis, similar to the *Prdm9⁻/⁻* mutant. (**A**) Schematic of the *Zcwpw1* knockout (KO) mouse line. E: Exon. gRNA: guideRNA. Sanger sequencing DNA chromatograms of wild-type (WT) and KO mice encompassing the deletion are shown. The intron-exon organisation is not to scale. (**B**) Immunofluorescence staining of testis nuclear spreads from 9- to 10-week-old *Zcwpw1⁺/⁺* and *Zcwpw1⁻/⁻* mice for ZCWPW1 or HORMAD2 (red) which marks asynapsed chromosomes, and the synaptonemal complex protein SYCP3 (green) which labels the chromosome axis. Cells were counterstained with DAPI (blue) to visualise nuclei (top images). These images are representative of the data obtained for three mice per genotype. Scale bar: 10 μm. (**C**) Representative testes from 9- to 10-week-old WT (+/+), Het (+/−) and Hom (−/−) *Zcwpw1* KO mice are shown. (**D**) Paired testes weight was normalised to lean body weight. Each datapoint represents one mouse. The p-value is from Welch's two sided, two sample t-test. Raw data in *Figure 3—source data 1*. (**E**) Synapsis quantification in testis chromosome spreads immunostained with HORMAD2, as in (B). The percentage of mid-Pachytene (WT) or pseudo-Pachytene (*Zcwpw1⁻/⁻* and *Prdm9⁻/⁻*) cells with all autosomes fully synapsed is plotted by genotype; each datapoint represents one mouse, each with n≥ 49 cells analysed. Vertical lines are 95% Wilson binomial confidence intervals. Raw data in *Figure 3—figure supplement 2—source data 1*.

The online version of this article includes the following source data and figure supplement(s) for figure 3:

**Source data 1.** Fertility measures in WT (+/+), *Zcwpw1⁻/⁻* and *Prdm9⁻/⁻* males.

*Figure 3 continued on next page*

*Figure 3 continued*

**Source data 2.** Breeding performance of *Zcwpw1⁻/⁻* females.
**Figure supplement 1.** Loss of ZCWPW1 expression in *Zcwpw1⁻/⁻* mouse testis.
**Figure supplement 2.** Asynapsis, and lack of XY body formation and crossover sites in *Zcwpw1⁻/⁻* mouse testis.
**Figure supplement 2—source data 1.** Impaired synapsis in *Zcwpw1⁻/⁻* males.

recombination crossover sites (*Figure 3—figure supplement 2B*), and persistent DMC1 foci marking unrepaired DSBs (*Figure 4*).

Each of these properties resembles observations in the *Prdm9⁻/⁻* mutant (*Hayashi et al., 2005*), and so we compared to this mutant. In our *Zcwpw1⁻/⁻* male mice, >98% of pachytene cells failed to properly synapse at least one pair of chromosomes (*Figure 3B,E*), similar to *Prdm9⁻/⁻* males. However, the nature of the synaptic defects observed differed (*Figure 3—figure supplement 2—source data 1*). In the *Prdm9⁻/⁻* mutant, 63.2% of the pseudo-pachytene cells showed mispairing of non-homologous chromosomes in a typical branched structure (referred to as 'tangled', *Figure 3—figure supplement 2—source data 1*). In contrast, we only observed 24.8% of *Zcwpw1⁻/⁻* pseudo-pachytene cells with this type of error, while the majority (74.5%) of cells contained multiple bundles of HORMAD2-positive asynapsed chromosomes, which resemble the XY body and may merge with the sex chromosomes (thus referred to as 'multibodies', *Figure 3—figure supplement 2*). These results imply that the *Prdm9⁻/⁻* mutant often mispairs chromosomes, while *Zcwpw1⁻/⁻* spermatocytes

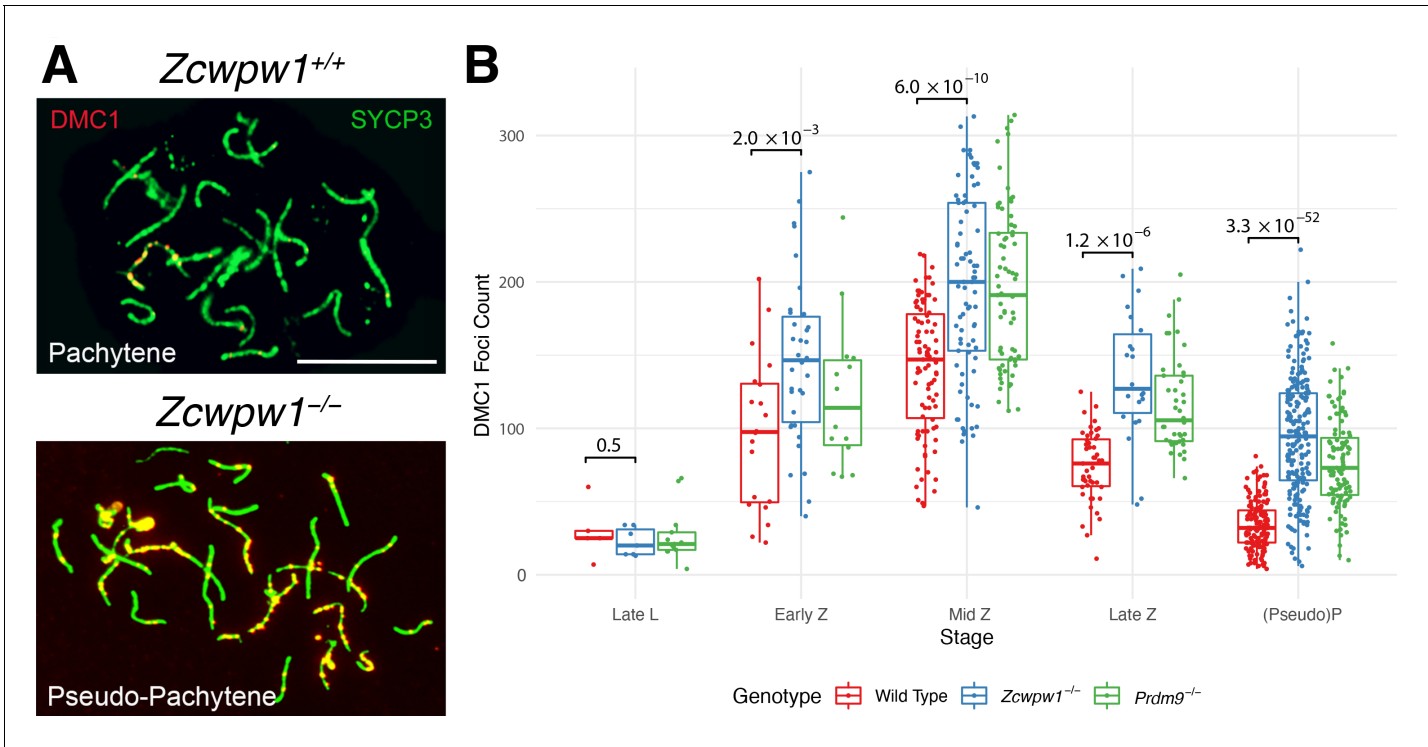

**Figure 4.** Similar DMC1 count elevation in *Zcwpw1⁻/⁻* and *Prdm9⁻/⁻* mice, compared to wild-type. (**A**) Testis chromosome spreads from 9- to 10-week-old *Zcwpw1⁺/⁺* and *Zcwpw1⁻/⁻* mice were immunostained for DMC1 and SYCP3. Late (pseudo)-Pachytene cells are shown. These images are representative of the data obtained for three mice per genotype. Scale bar: 10 μm. (**B**) The number of DMC1 foci in cells from the indicated stages of prophase I were counted; see *Figure 4—source data 1* for number of cells per stage per mouse. p-values are from Welch's two sided, two sample t-test. L: Leptotene, Z: Zygotene, P: Pachytene. n = 3 mice for *Zcwpw1⁻/⁻* and wild-type, n = 2 for *Prdm9⁻/⁻*. Raw data in *Figure 4—source data 1*.
The online version of this article includes the following source data and figure supplement(s) for figure 4:

**Source data 1.** Raw data for DMC1, RAD51 and RPA2 foci counts.
**Figure supplement 1.** Similar RAD51 count elevation in *Zcwpw1⁻/⁻* and *Prdm9⁻/⁻* mice, compared to wild-type.
**Figure supplement 2.** RPA2 count elevation in the *Zcwpw1⁻/⁻* mouse.
**Figure supplement 3.** DSB repair is delayed with accumulation of DMC1 on asynapsed chromosomes in the *Zcwpw1⁻/⁻* mouse.

mainly fail to pair a subset of chromosomes at all. The expression levels and staining pattern of ZCWPW1 were not visibly altered in testis nuclear spreads (data not shown) and protein extracts (*Figure 3—figure supplement 1A*) from *Prdm9$^{-/-}$* mice. However, since the *Zcwpw1* KO and *Prdm9* KO lines are on a different genetic background (C57/BL6N and B6×129, respectively), we cannot rule out the possibility that this explains partly or fully the differences in meiotic defects observed between the two mutants.

Comparing levels of DMC1 foci as a proxy for DSB repair, or a potential cause for the asynapsis in *Zcwpw1$^{-/-}$* males, in both *Zcwpw1$^{-/-}$* and *Prdm9$^{-/-}$* mice foci count was significantly elevated from zygotene onwards, indicating delayed repair of DSBs (*Figure 4*). However, we observed a wider spread for *Zcwpw1$^{-/-}$* males. A similar increase was observed at pachytene stage in the levels of RAD51 (*Figure 4—figure supplement 1*), a strand exchange protein which functions in concert with DMC1, with the large majority of RAD51 and DMC1 foci co-localising (*Brown et al., 2015*; *Tarsounas et al., 1999*). Like DMC1, RPA2 levels were also significantly elevated in the *Zcwpw1$^{-/-}$* mouse from zygotene onwards (*Figure 4—figure supplement 2*). In *Zcwpw1$^{-/-}$* males, like in the *Prdm9$^{-/-}$* mutant, DSBs form and recruit RPA2, RAD51 and DMC1 in similar numbers (*Huang et al., 2020*; *Li et al., 2019a*; *Mahgoub et al., 2020*), but fail to repair efficiently, accompanied by asynapsis and meiotic arrest at pseudo-pachytene. Indeed, we observe late unrepaired DMC1 foci mainly on HORMAD2-positive asynapsed chromosomes (*Figure 4—figure supplement 3*).

## ZCWPW1 is recruited to PRDM9-binding sites in an allele-specific manner

We previously studied the binding properties of human PRDM9 and established a genome-wide map in transfected human mitotic (HEK293T) cells by ChIP-seq (*Altemose et al., 2017*), observing binding to the majority of human meiotic recombination hotspots. Based on the presence of H3K4me3 and H3K36me3 recognition domains in ZCWPW1, we hypothesised that it would be recruited to PRDM9-bound genomic sites, where these marks are deposited upon binding in HEK293T cells (*Altemose et al., 2017*).

To test this, we co-transfected HEK293T cells with full-length human HA-tagged ZCWPW1 and either no other protein, or full-length *PRDM9* alleles carrying the human or chimpanzee zinc finger array, as studied previously (*Altemose et al., 2017*), and then performed ChIP-seq against the ZCWPW1 tag (the endogenous levels of these proteins in HEK293T cells, and of the closely related family member ZCWPW2, are extremely low so unlikely to confound the analysis - see *Figure 5—source data 1*). Confirming the recruitment hypothesis, in the presence of human PRDM9 and compared to cells lacking human PRDM9, ZCWPW1 shows a strong enrichment at human PRDM9-binding sites with higher enrichment at sites with higher PRDM9 enrichment (*Figure 5A,B*, Pearson's correlation = 0.43 - *Figure 5—figure supplement 1*). Notably, even without PRDM9 we observed many ZCWPW1 binding peaks across the genome, some coinciding with PRDM9-binding sites (*Figure 5C* and *Figure 5—figure supplement 2*). However, upon co-transfection with PRDM9, 92% of the strongest 10,000 ZCWPW1 peaks are at PRDM9-bound sites (98% of the strongest 3,000 vs 7% expected overlap with randomised peaks, and only 47% overlap for the strongest 3,000 ZCWPW1 peaks in cells without PRDM9 transfection). Hence, PRDM9 is able to strongly reprogram ZCWPW1 binding (*Figure 5C*, see also *Figure 5—figure supplement 3*), suggesting that loci with peaks in cells without PRDM9 transfection are bound more weakly. Because our transfection has <100% efficiency (*Figure 5—figure supplement 4*), some cells containing ZCWPW1 will not possess PRDM9, and so it is possible that an even greater fraction of ZCWPW1 is redirected to PRDM9 binding sites in cells where both proteins *are* present.

Importantly, when co-transfecting with a modified version of PRDM9 in which the zinc finger array is replaced with that from chimp (which binds different locations in the genome [*Altemose et al., 2017*]), we find that the enrichment at human binding sites disappears, but instead ZCWPW1 is enriched at *chimp* PRDM9-binding sites (*Figure 5A* and *Figure 5—figure supplement 5*). This perturbation experiment provides strong evidence that PRDM9 *causes* recruitment of ZCWPW1 (as opposed to for example independent recruitment of both proteins). Notably, the strength of ZCWPW1 binding in HEK293T cells at human PRDM9-binding sites provides a better predictor of human meiotic DMC1 binding status (a proxy for DSB formation) than does human PRDM9 binding strength itself (*Figure 5—figure supplement 6*). We show (see below) that ZCWPW1 is *not* directly

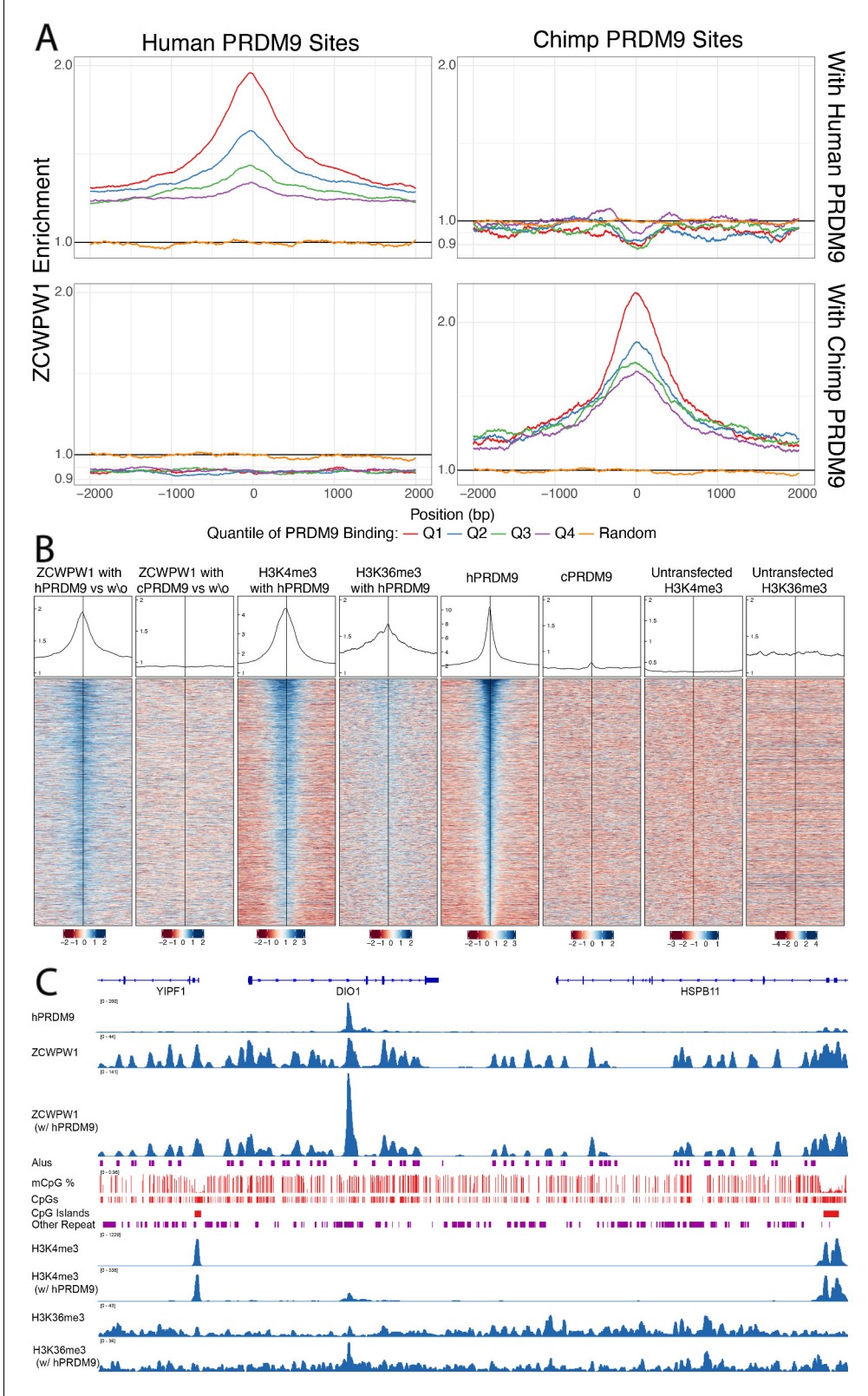

**Figure 5.** Enrichment and binding profiles of ZCWPW1 and other factors. (**A**) Enrichment of ZCWPW1 (with vs without PRDM9) at PRDM9-binding sites when co-transfected with PRDM9 with either Human or Chimp Zinc Finger (Materials and methods section 'Enrichment Profiles'). Q = quartile. Human PRDM9 sites are centered and stranded by the motif. Y-axis is log10 scale (y-axis labels remaining in linear space). (**B**) Profiles and heatmaps of reads from cells co-transfected with human (h) or chimp (c) PRDM9 around the top 25% of individual human PRDM9-binding sites (rows). Heatmaps: log-fold

*Figure 5 continued on next page*

*Figure 5 continued*

change of target (indicated in column titles, Materials and methods) vs input, for various labelled target proteins, ordered by human PRDM9. ZCWPW1, H3K4me3 and H3K36me3 each become enriched at human PRDM9 sites, following (co-)transfection with human PRDM9. Profiles: Sum of all target coverage divided by sum of all input coverage for all regions shown in the heatmap, shown on a linear scale. w\o, without. (C) ChIP-seq data and annotation in a genome plot illustrate the behaviour of ZCWPW1 and other factors. ChIP-seq tracks show fragment coverage. Tracks where PRDM9 is present are labeled 'w/PRDM9', and below, corresponding tracks without PRDM9. ZCWPW1 binds to Alus, CpG islands and other CpG-rich sequences even in the absence of PRDM9. On addition of PRDM9, ZCWPW1 becomes strongly enriched at PRDM9 binding locations (center left peak within DIO1). mCpG, methylated CpG.

The online version of this article includes the following source data and figure supplement(s) for figure 5:

**Source data 1.** RT-PCR analysis of *PRDM9*, *ZCWPW1* and *ZCWPW2* transcript expression in HEK293T cells.
**Figure supplement 1.** Correlation between PRDM9 enrichment and ZCWPW1 enrichment at sites of PRDM9 binding.
**Figure supplement 2.** Proportion of ZCWPW1 peaks, ordered by enrichment of ZCWPW1 binding over input, overlapping various other marks.
**Figure supplement 3.** Enrichment of ZCWPW1 when co-transfected with human or chimp PRDM9 is dependent on the ability of ZCWPW1 to bind, more weakly, in the absence of PRDM9 (there are no peaks with high co-transfected enrichment [y-axis] when the untransfected enrichment [x-axis] is close to 0) and co-transfecting with PRDM9 increases the enrichment.
**Figure supplement 4.** Co-expression of ZCWPW1 and PRDM9 in HEK293T cells.
**Figure supplement 5.** Profiles and heatmaps of reads at locations of either chimp PRDM9 binding or ZCWPW1 binding when co-transfected with human PRDM9.
**Figure supplement 6.** Among human PRDM9 binding sites, we identified those at which male recombination hotspots occur, defined by the presence/absence of an overlapping human DMC1 peak, and fitted a linear model to predict this hotspot status based on PRDM9 binding strength (PRDM9 Only), ZCWPW1 enrichment (with human PRDM9 vs without, referring to enrichment of ZCWPW1 co-transfected with PRDM9 relative to ZCWPW1 transfected alone), or both (see Materials and methods 'DMC1 prediction'). We fitted a logistic regression model, and present the results in the form of standard Receiver Operating Characteristic curves (A) and Precision Recall Curves (B).

involved in DSB positioning, this result might therefore instead suggest that similar features to those ZCWPW1 recognises are involved in recruiting DSB formation machinery.

We tested whether the recruitment of ZCWPW1 to sites bound by PRDM9 might be mediated by the dual histone modifications H3K4me3 and H3K36me3. Consistent with this idea, ZCWPW1 binding is positively associated with levels of both H3K4me3 and H3K36me3 marks (*Figure 6—figure supplements 1* and *2*, tested separately, not necessarily 'dual' marks coincident within an individual cell, although locations of these marks in meiotic cells are rarely coincident except at hotspots [*Powers et al., 2016*]). While we cannot rule out a direct interaction between ZCWPW1 and PRDM9, others have shown that dual modified peptides have a greater affinity to ZCWPW1 than peptides carrying the single modifications, supporting that the marks themselves are responsible for ZCWPW1's recruitment (*Mahgoub et al., 2020*). We examined transcription start sites, which possess H3K4me3 at high levels, but lack H3K36me3, observing some ZCWPW1 signal at these sites, but with a uniformly lower mean ZCWPW1 enrichment compared to those with evidence of PRDM9 binding (and hence both histone marks) (*Figure 6*). Although it is not possible to measure H3K4me3 presence/absence in *individual* cells in our system, even the most weakly bound PRDM9 sites – which must therefore possess H3K4me3 in only a fraction of cells – show stronger ZCWPW1 enrichment than the strongest promoters, which are likely to have near 100% H3K4me3 marking, and possess >2 fold more H3K4me3 than even the strongest PRDM9-binding sites. We conclude that H3K4me3 alone endows only relatively weak binding, while PRDM9 is therefore able to recruit ZCWPW1 with a much greater efficiency than sites marked by H3K4me3 alone, suggesting that both histone modifications might aid efficient binding. While the increase in ZCWPW1 enrichment at sites with pre-existing H3K4me3 is consistent with both marks contributing to recruitment, we have no evidence that these weaker binding sites serve a functional role, and indeed concurrent investigations did not find strong evidence of ZCWPW1-driven transcriptional changes (*Huang et al., 2020*).

## DSBs occur at their normal locations in *Zcwpw1*$^{-/-}$ mice but show increased DMC1 levels

Previous work has shown that *Prdm9*$^{-/-}$ mice use a new set of DSB hotspots, localising at CpG islands and/or promoter regions (*Brick et al., 2012*). Given that PRDM9 recruits ZCWPW1, one possible function of ZCWPW1 may be that, in turn, it recruits the DSB machinery and hence forms part of the causal chain in normal positioning of DSBs at PRDM9-specified hotspots. Alternatively, the *Zcwpw1*$^{-/-}$ mutant phenotypes we observe might reflect a more downstream role. To distinguish

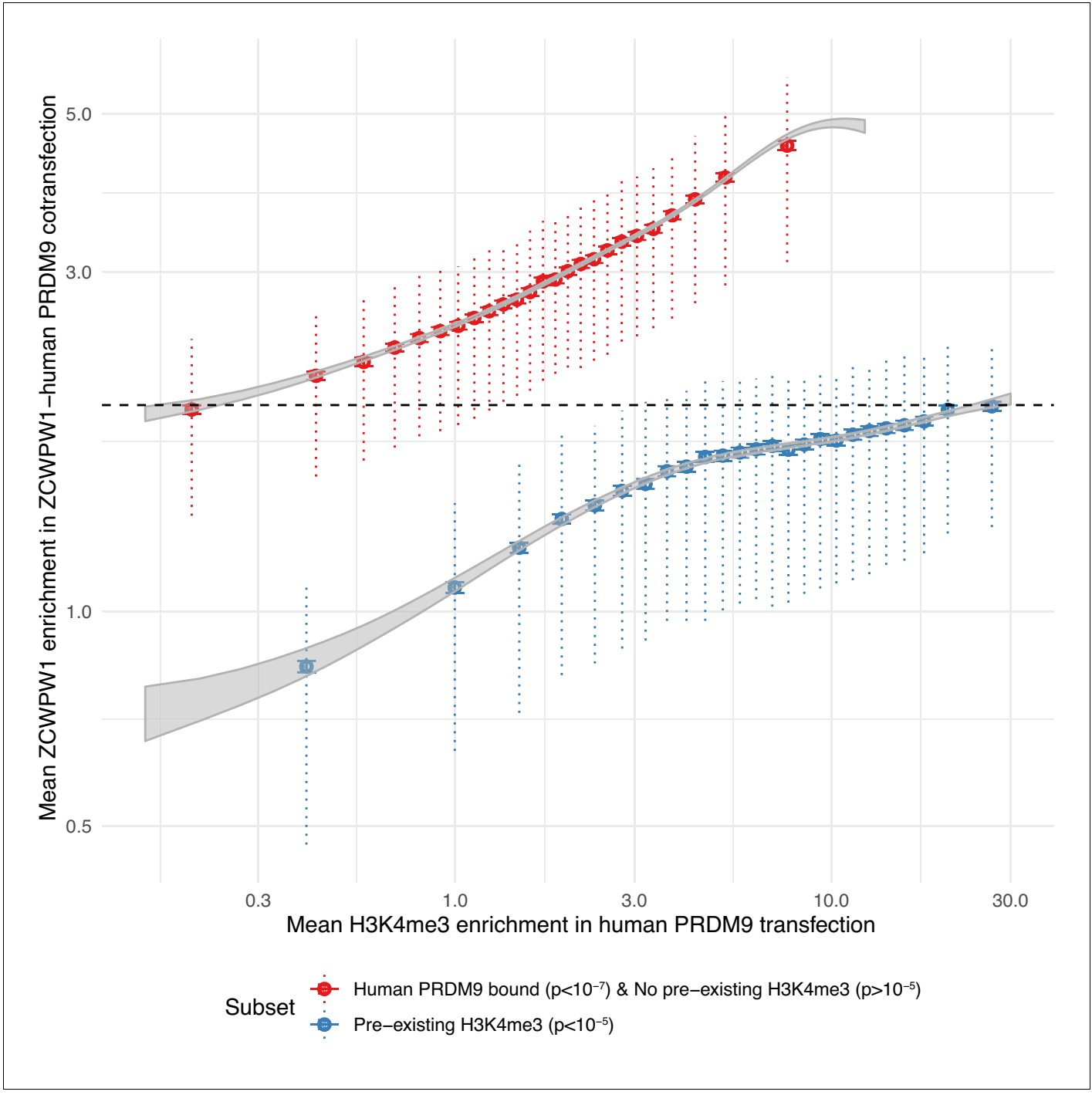

**Figure 6.** PRDM9-bound regions (H3K4me3 and H3K36me3) are a stronger recruiter of ZCWPW1 than promoters (H3K4me3 only). For any given level of H3K4me3 (x-axis), ZCWPW1 enrichment (y-axis) is higher at PRDM9-bound regions (red) than regions with pre-existing H3K4me3 (promoters, blue). H3K4me3 and ZCWPW1 were force called in 100bp windows across all autosomes. These windows were split into two sets defined as indicated in the legend (where 'p' is the p-value from peak-calling required for a window to be included in the subset) with the additional constraint of requiring input fragment coverage >5 for ZCWPW1 and >15 for H3K4me3. p: p-value for non-zero level of input corrected coverage in that bin. 'pre-existing H3K4me3' refers to H3K4me3 that is present without transfection (of either PRDM9 or ZCWPW1), which is mainly found at promoter regions. For each subset, H3K4me3 was split into 25 bins with equal number of data points. Horizontal bars: two standard errors of the mean. Vertical dotted bars: upper and lower quartiles. Grey ribbons show two standard errors for a Generalized additive model on log(mean H3K4me3 enrichment + 0.1). Dashed black horizontal line highlights that the mean enrichment of the highest bin for promoters is similar to that of the lowest bin for PRDM9-bound sites. The online version of this article includes the following figure supplement(s) for figure 6:

*Figure 6 continued on next page*

Figure 6 continued

**Figure supplement 1.** ZCWPW1 binding is positively associated with levels of both H3K4me3 and H3K36me3 marks.

**Figure supplement 2.** Enrichment from 100-bp non-overlapping windows, genome-wide, is binned into 100 equal sample size bins by either.

these hypotheses, we gathered data on DSB positioning and repair dynamics genome-wide, by carrying out single ssDNA sequencing (SSDS) by ChIP-seq against DMC1 (*Khil et al., 2012*).

In the *Zcwpw1*$^{-/-}$ mutant males, we observed normal localisation of DMC1 at the expected B6 WT hotspots for this background, with no signal at *Prdm9*$^{-/-}$ hotspots (*Figure 7A*). We only fail to see evidence of DMC1 signal in a subset of the weakest hotspots, where we are likely to lack statistical power (*Figure 7—figure supplement 1*). Thus, DSBs occur in unchanged hotspot regions in *Zcwpw1*$^{-/-}$ males, relative to WT. To check if they occur at the same locations *within* hotspot regions, we leveraged data for SPO11-mapped DSB sites (*Lange et al., 2016*). Specifically, we identified three sets of hotspots whose respective mapped breaks occur mainly upstream, central, or downstream of the PRDM9-binding site (Materials and methods), and compared their DMC1 ChIP-seq signal profiles, as well as their SPO11 signals (*Figure 7—figure supplement 2*). This revealed that in the *Zcwpw1*$^{-/-}$ males, as in the WT, DMC1 signals mirror this break positioning. Therefore, ZCWPW1 is not required to specify hotspot locations, and neither does it strongly influence where DSBs occur *within* hotspots.

One perhaps important difference between the *Zcwpw1*$^{-/-}$ and WT DMC1 profiles is that the latter signal is slightly wider (up to ~200bp) in hotspots with increasing PRDM9-induced H3K4me3 (*Figure 7B*), while this effect is absent in the *Zcwpw1*$^{-/-}$ case. This might be explained by small differences in chromatin accessibility subtly impacting SPO11 locations, DNA end resection which generates the 3' ssDNA tail to which DMC1 binds, or downstream processing differences altering DMC1 span in the mutant mice.

Despite very similar DSB locations, we saw much greater, and systematic, differences in the strength of the DMC1 signal at individual hotspots. As previously (*Davies et al., 2016*; *Khil et al., 2012*), we note that observed average DMC1 signal strength at a hotspot reflects the product of the frequency at which DSBs occur there, and the average length of time DMC1 remains bound to the ssDNA repair intermediate, with the latter reflecting DSB processing/repair time. In contrast, available SPO11 oligo-seq data (*Lange et al., 2016*) reflect mainly the frequency of DSBs. We therefore compared DMC1 and SPO11 signal strength at each autosomal and X-chromosome hotspot, in WT and *Zcwpw1*$^{-/-}$ male mice. In WT mice, the non-PAR X-chromosome shows a very strong elevation of DMC1 signal strength (as seen previously in other mice [*Davies et al., 2016*] including, albeit somewhat more weakly, even sterile hybrids). This elevation reflects the persistent DMC1 foci on this chromosome also visible using microscopy, at DSB sites that eventually repair using the sister chromatid. Moreover, in WT mice there is a sub-linear (curved) relationship between SPO11 and DMC1 at hotspots (*Figure 7C*). This is thought to reflect a wider phenomenon of faster DSB repair occurring within those hotspots whose homologue is more strongly bound by PRDM9, that is those hotspots with a stronger H3K4me3 signal in the WT mouse (*Davies et al., 2016*; *Hinch et al., 2019*; *Li et al., 2019b*). The X-chromosome DMC1 elevation is in a sense an extreme case of slower repair of DSBs whose homologue is not PRDM9-bound, because no homologue exists in this case.

However, we see a striking departure in the *Zcwpw1*$^{-/-}$ mouse, where the X-chromosome behaves more similarly to the autosomes with respect to DMC1 vs WT SPO11 signal strength (*Figure 7C*). Moreover, we see a simple linear relationship between DMC1 and WT SPO11 binding in this mouse (*Figure 7C*). This implies that the effect of PRDM9 binding on DMC1 removal is eliminated in this mouse (assuming SPO11 levels are not greatly affected by *Zcwpw1* KO); so PRDM9 appears unable to 'assist' in homologue pairing and synapsis in this mouse (whether because ZCWPW1 directly aids the process, or indirectly). Indeed, the data imply a widespread perturbation of DSB repair in this mouse, with autosomal DMC1 foci persisting as long as those on the X-chromosome. In the previously studied *Hop2*$^{-/-}$ mouse (*Khil et al., 2012*; *Petukhova et al., 2003*; *Smagulova et al., 2011*), we identified a very similar pattern (*Figure 7—figure supplement 3*): in this mouse, DMC1 is loaded onto ssDNA, but DSBs are not repaired at all, indicating that DSB repair pathways in *Zcwpw1*$^{-/-}$ mice are profoundly altered.

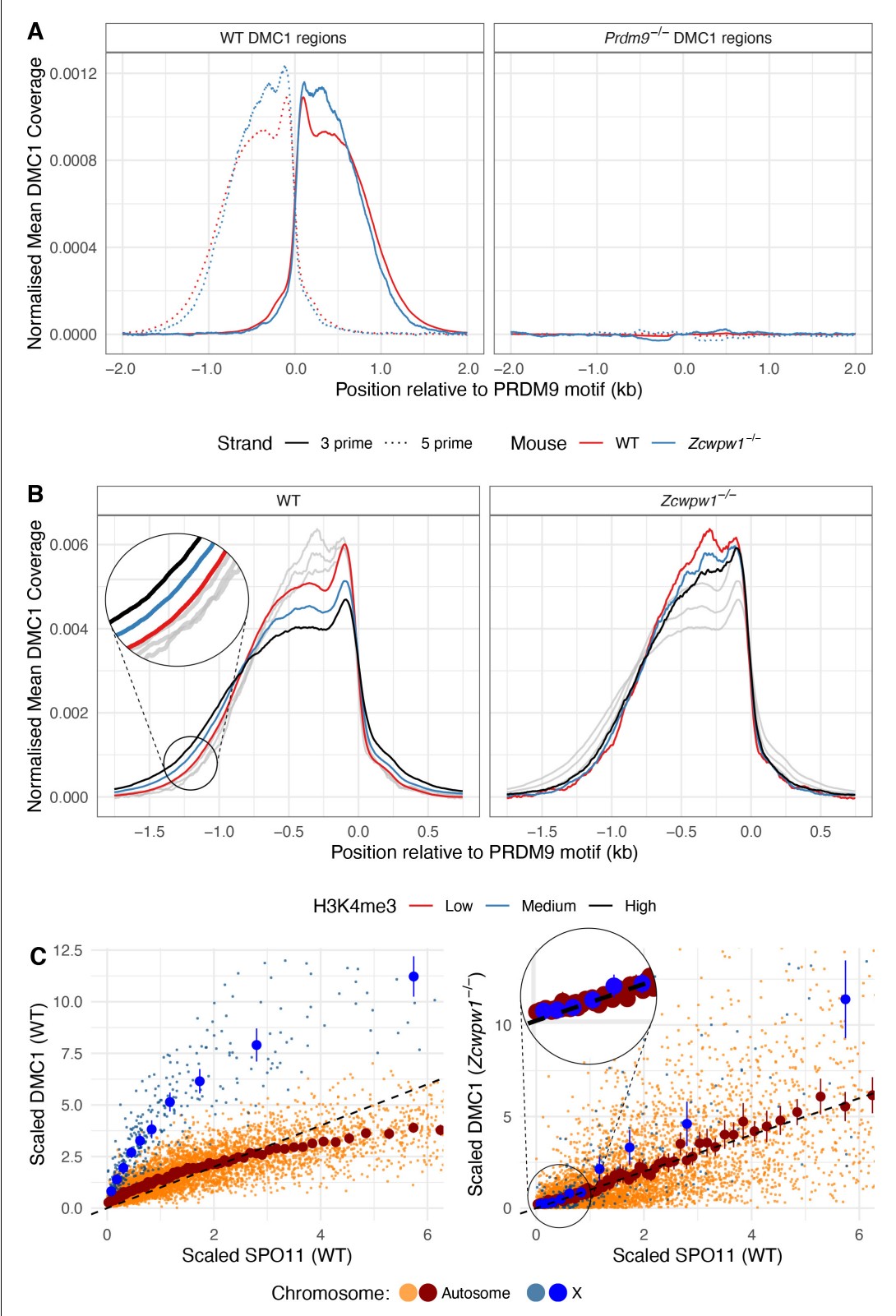

**Figure 7.** DMC1 levels in the *Zcwpw1⁻/⁻* mouse compared to DMC1 and SPO11 levels in WT. (**A**) DSBs occur at normal hotspot locations in the *Zcwpw1⁻/⁻* male mouse. Average coverage of reads from DMC1 SSDS ChIP-seq in a 10-week-old mouse at previously mapped regions (Materials and methods) in B6 WT (left) and *Prdm9⁻/⁻* (right) mice is shown, centered at the PRDM9 motif (left). DMC1 profiles from a WT mouse are shown in red, data from *Brick et al., 2012*. (**B**) Normalised DMC1 profile (both strands combined) is plotted for WT and *Zcwpw1⁻/⁻*, stratified by

*Figure 7 continued*

H3K4me3 (a proxy for PRDM9 binding). Low: <50th percentile cumulative enrichment, High: >75th percentile cumulative enrichment, with Medium being the remaining data. Greyed out lines show the alternative genotype for comparison. (C) Relationship between WT SPO11-oligos (measuring the number of DSBs) vs DMC1 (a measure of the number and persistence of DSBs) at each B6 hotspot for WT and *Zcwpw1*$^{-/-}$. Unlike WT mice, DMC1 signals in *Zcwpw1*$^{-/-}$ mice are approximately linearly associated with WT SPO11. The DMC1 enrichment was force called at the positions of B6 WT hotspots. Black dashed line is y = x for reference. SPO11 and DMC1 enrichment have been scaled by dividing by the mean autosomal enrichment. Large dark blue and dark red points show mean DMC1 signal, binned into groups containing equal numbers of hotspots by WT SPO11 signal (vertical lines: corresponding 95% CIs), for X (10 bins) and autosomal data (100 bins) respectively (smaller lighter dots represent individual hotspots).

The online version of this article includes the following figure supplement(s) for figure 7:

**Figure supplement 1.** Fraction of wild-type (WT) hotspot locations seen in *Zcwpw1*$^{-/-}$ DMC1 ChIP-seq at different p-values.

**Figure supplement 2.** DSBs in *Zcwpw1*$^{-/-}$ are positioned at WT locations within hotspots.

**Figure supplement 3.** Relationship between WT SPO11-oligos (measuring the number of DSBs) vs DMC1 (a measure of the number and persistence of DSBs) at each B6 hotspot for *Hop2*$^{-/-}$ male mice (A) *Zcwpw1*$^{-/-}$ (B) and WT (C) as in *Figure 7C* are replotted, for comparison.

**Figure supplement 4.** Regression of the ratio of DMC1 signal in the *Zcwpw1*$^{-/-}$ (KO) vs wild-type (WT) male mice against H3K4me3 [a proxy of PRDM9 binding] (A), SPO11 (B), and DMC1 (C) in WT.

These results are consistent with, and extend, our microscopy observations (*Figure 4*) that many DSBs persist in the *Zcwpw1*$^{-/-}$ mouse, and some may never repair. We are unable to say whether DSB repair involves the homologue or not in this mouse, but the partial synapsis we observe suggests some repair does likely occur. To further understand DMC1 persistence at individual hotspots, we estimated the relative DMC1 heat of autosomal hotspots in the *Zcwpw1*$^{-/-}$ mouse, compared to the WT mouse. The best individual predictor of this ratio among H3K4me3, DMC1, and SPO11 in WT was the level of H3K4me3 (r = 0.60 Generalised Additive Model; *Figure 7—figure supplement 4*, while r = 0.63 when using all three predictors together). The average KO:WT DMC1 ratio increases around ninefold from the most weakly to the most strongly H3K4me3-marked hotspots. Therefore, if DMC1 signal changes are indeed explained by slower DSB processing in the *Zcwpw1*$^{-/-}$ mouse at hotspots bound strongly by PRDM9, then this implies a very strong effect of PRDM9's 'assistance' in aiding repair. Whatever the cause, the overall strong correlation implies perturbation of DMC1 behaviour is widespread, rather than impacting any small subset of hotspots. Moreover, it appears to be mainly controlled by local levels of PRDM9 histone modification, in keeping with the evidence that ZCWPW1 recognises this mark at PRDM9-bound sites, on either the broken or (identical) homologous chromosome.

## ZCWPW1 binds CpG dinucleotides

In addition to the strong PRDM9-dependent ZCWPW1 peaks described earlier, there are many locations in HEK293T cells at which ZCWPW1 binds, typically more weakly, and independently of PRDM9. Indeed we identified over 800,000 ZCWPW1 peaks. Surprisingly, a large proportion of these binding sites overlap Alu repeats (*Figure 8A*, *Figure 5C*, and *Figure 5—figure supplement 2*) (of which there are 1.1 million in the human genome [*Deininger, 2011*]). The weakest ZCWPW1 peaks overlap Alus most frequently, whilst the strongest peaks are depleted of Alus relative to chance overlap (*Figure 8A*).

Because Alus are rich in CpGs (containing >23% of human CpG dinucleotides, with deamination depleting CpGs in older Alus [*Luo et al., 2014*]), we tested if binding correlates with CpG presence/absence. Indeed, the binding of ZCWPW1 to Alus depends on the presence of CpGs, which are mainly methylated (*Xie et al., 2011*). Alus with no CpGs were bound at a rate lower than by chance, but almost all Alus containing 10–20 CpGs are bound (*Figure 8B*). This dependence is not Alu-specific: even outside genomic repeats, 300bp regions with zero CpGs have negligible probability of overlapping a ZCWPW1 binding site, while the overlap probability rises to >50% for regions containing 10 or more CpGs, suggesting CpGs are essential for PRDM9-independent ZCWPW1 binding (*Figure 8C,D* and *Figure 8—figure supplement 1*). ZCWPW1 appears to have greater affinity for methylated CpG pairs, but retains some affinity even for non-methylated regions. This binding mode was unexpected a priori, and does not appear to be easily explained by patterns of H3K4me3/H3K36me3 in the genome, which do not concentrate as strongly in these regions.

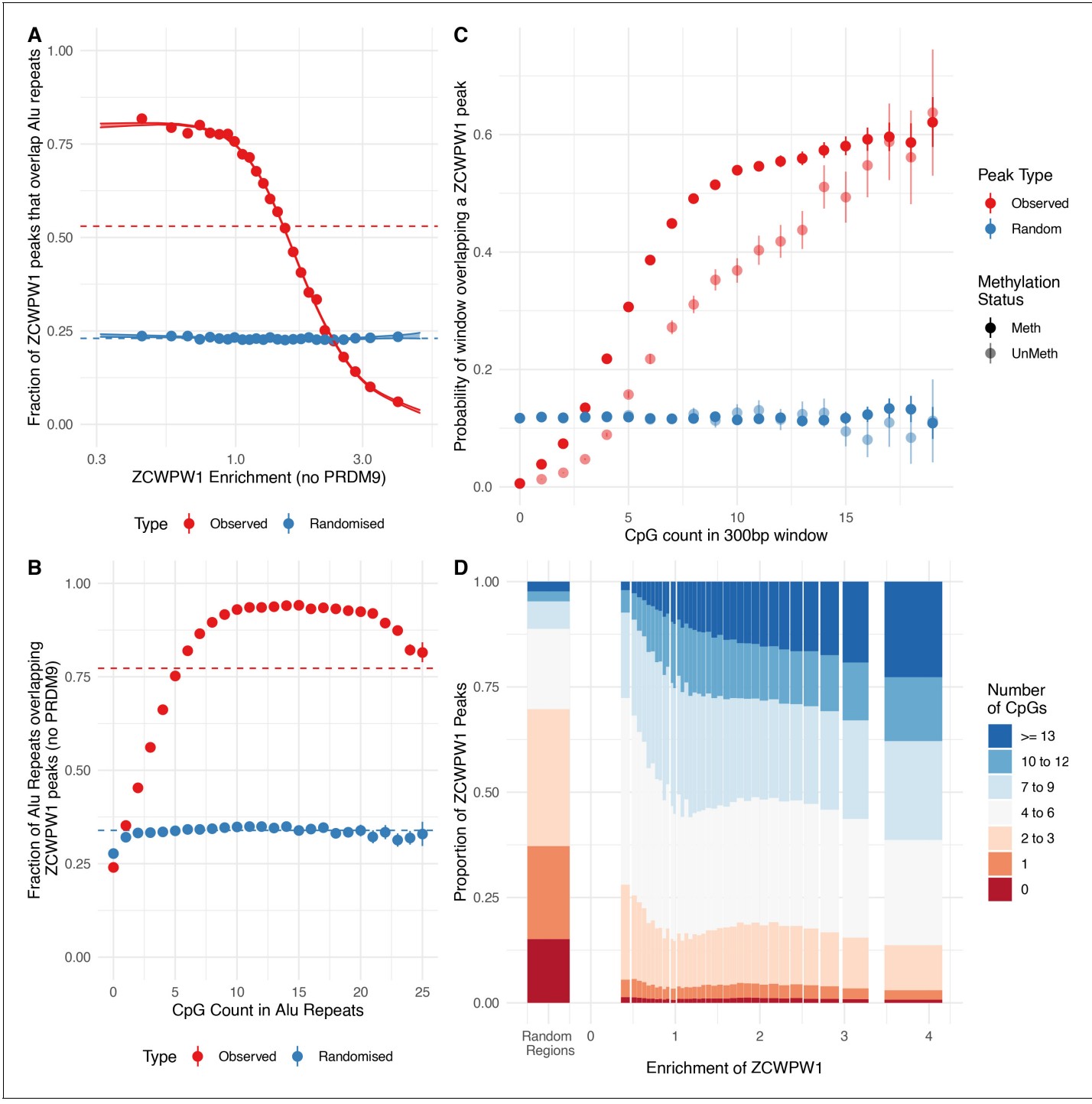

**Figure 8.** ZCWPW1 binds CpG-rich sequences such as Alu repeats. (**A**) Fraction of overlap of ZCWPW1 binding peaks, , with Alusin HEK293T cells transfected with ZCWPW1 alone, ordered by enrichment in ZCWPW1 binding. ZCWPW1 peaks are binned into 25 bins with equal number of data points, and means of both enrichment and overlap are plotted. Solid ribbons: prediction from GAM logistic regression. Dotted lines: overall means. Red points show actual observed peaks, blue points the same number of peaks placed at random genomic positions. (**B**) Rate of overlap of Alu repeats with ZCWPW1 peaks, for Alus with different numbers of CpG dinucleotides. Other details as A. (**C**) The probability of a 300bp window on an autosome overlapping a ZCWPW1 peak increases with increasing CpG count in that window. Windows overlapping (by 10bp or more) Alus, other repeats, or CpG islands have been excluded. Methylated CpG regions (full colour) are those with a methylated to unmethylated reads ratio of >0.75, and unmethylated <0.25 (semi-transparent, Materials and methods). (**D**) Relative proportion of peaks with given numbers of CpGs (stacked bars) +/− 150bp from peak center, within peaks binned by ZCWPW1 enrichment (x-axis). ZCWPW1 peaks are enriched in CpGs compared to random peak locations (leftmost bar). 'Meth', methylated; 'UnMeth', unmethylated.

*Figure 8 continued on next page*

*Figure 8 continued*

The online version of this article includes the following figure supplement(s) for figure 8:

**Figure supplement 1.** CpG count around ZCWPW1 peaks (+/− 150bp, for those peaks with input coverage >5) is positively associated with ZCWPW1 enrichment score (measuring the level of ZCWPW1 recruitment) in both peaks overlapping Alus and peaks not overlapping Alus, but not at L1M1-3, L1MA or L1P repeats.

## Discussion

Here, we exploited previous work on single-cell transcript profiling of mouse testis, which has established spatio-temporal (and likely functional) components of co-expressed genes (*Jung et al., 2019*), to identify co-factors relevant to PRDM9 function in specification of recombination hotspots and DSB repair during meiosis. We show that the histone methylation reader ZCWPW1, which is both highly co-expressed and co-evolving with PRDM9, is recruited by PRDM9 to its binding sites, likely mediated through the recognition of the dual H3K4me3-H3K36me3 histone mark that PRDM9 deposits.

Like Mahgoub and colleagues in a parallel analysis (*Mahgoub et al., 2020*), we find co-evolution of ZCWPW1 with PRDM9, consistent with proteins functioning in the same pathway. We find that this association appears to depend on particular domains. ZCWPW1 co-occurs with PRDM9 in species where PRDM9 possesses a SET domain predicted to retain catalytic activity for H3K4me3/H3K36me3, which the domains of ZCWPW1 are predicted to recognise, strongly suggesting a partnership totally dependent on this recognition. This co-occurrence can persist even in species where PRDM9 is not believed to position DSBs (*Baker et al., 2017*), suggesting the co-operation between these proteins does not completely depend on this aspect of PRDM9's function. Interestingly, a predicted functional SET domain of PRDM9 mainly occurs when PRDM9 possesses a SSXRD domain, which in other proteins (*Banito et al., 2018*) recruits CXXC2 which binds to CpG islands – whether this connects to our observation of a potential CpG-binding domain in ZCWPW1, and CpG-binding behaviour in a human cell line, remains to be explored.

A recent study reported phenotyping of an independent *Zcwpw1*$^{-/-}$ mouse (*Li et al., 2019a*), and we observe similar complete loss of fertility in males, but not in young females. Why females deficient for ZCWPW1 are initially fertile (but lose fertility by ~8 months; *Li et al., 2019a*) is unknown, but resembles sexual dimorphism in loss-of-function mutants for other meiotic genes involved in homologous recombination, where females exhibit a milder fertility phenotype (*Cahoon and Libuda, 2019*; *Morelli and Cohen, 2005*; *Zhang et al., 2019*). However, we observe a distinct intracellular localisation of ZCWPW1 from that reported (*Li et al., 2019a*): while we also detect ZCWPW1 as a diffuse nuclear signal across the nucleus (although excluding the chromocenters) and localisation to the XY body in pachytene, we see an additional signal in mid-pachytene to diplotene cells at the ends of the synaptonemal complex. This might reflect the specificity of our polyclonal antibody, which is raised against the full-length mouse protein vs a monoclonal antibody raised against a C-terminal region, hence recognising a wider spectrum of ZCWPW1. Given that the *Zcwpw1*$^{-/-}$ mouse arrests in early pachytene, it is not possible to test later stages, so a caveat is that we cannot completely exclude off-target antibody recognition. If accurate, the late-stage localisation of ZCWPW1 at subtelomeric regions resembles other proteins (Speedy A, CDK2, TFR1, SUN1, TERB1/2, MAJIN, and KASH), which form a complex tethering the telomeres to the nuclear envelope to facilitate chromosome movement and pairing — essential for synapsis and fertility (*Ashley et al., 2001*; *Ding et al., 2007*; *Horn et al., 2013*; *Shibuya et al., 2014*; *Tu et al., 2017*). However, no evidence of impaired telomeric attachment was reported in the other *Zcwpw1*$^{-/-}$ mutant, based on normal localisation of TRF1 (*Li et al., 2019a*).

In parallel with this work, two other studies of ZCWPW1 *Huang et al., 2020*; *Mahgoub et al., 2020* have performed complementary analyses, and together with our own results allow some clear conclusions to be drawn. We mapped positions of ZCWPW1 recruitment (of the human protein) in mitotic HEK293T cells, in the presence of human or chimpanzee PRDM9 and compared to assayed peaks for PRDM9, H3K4me3 and H3K36me3 (*Altemose et al., 2017*). We find that in co-transfected HEK293T cells ZCWPW1 is strongly recruited to PRDM9-bound sites, in an allele-specific manner. *Mahgoub et al., 2020* also report *PRDM9* allele-specific recruitment of ZCWPW1, but in hybrid mice carrying *Prdm9*$^{Dom2}$ and *Prdm9*$^{Cast}$ alleles which encode proteins with distinct zinc finger array

DNA binding specificities. Together with ZCWPW1 binding measurements from *Huang et al., 2020*, these results imply that across PRDM9 alleles and species - humans, mice, and likely chimpanzees - PRDM9 binding alone is sufficient to strongly recruit ZCWPW1 without requiring DSB formation (which does not occur in our system) or meiosis-specific co-factors (which would not be expected to be present in our cellular system).

A likely explanation of this recruitment is recognition of H3K4me3 and H3K36me3 that PRDM9 deposits by the CW and PWWP domains of ZCWPW1 respectively. Previously, Li and colleagues (*Li et al., 2019a*) showed co-immunoprecipitation of H3K4me3 with ZCWPW1. In HEK293T cells we observed that ZCWPW1 recruitment correlates with the levels of H3K4me3 and H3K36me3 that PRDM9 deposits, while *Huang et al., 2020* reported that in mice with a non-functional CW domain no ZCWPW1 binding peaks were observed, suggesting that H3K4me3 recognition at least might be necessary for the recruitment of ZCWPW1 at hotspots in vivo. However, H3K4me3 cannot be the only factor because we observed much stronger recruitment of ZCWPW1 to PRDM9 binding sites than to other sites possessing similar H3K4me3 levels, which might be explained by the additional presence of H3K36me3 at PRDM9 binding sites increasing ZCWPW1's recognition and/or its stable association at these sites. Indeed in cells with PRDM9, the strongest ZCWPW1-binding sites are almost all PRDM9-bound sites. Similarly in mice, >85% of ZCWPW1 peaks overlap both marks concomitantly (*Huang et al., 2020*; *Mahgoub et al., 2020*). *Mahgoub et al., 2020* additionally reported that in vitro binding affinity of recombinant mouse ZCWPW1 is at least five fold greater for dual-modified H3K4me3/K36me3 peptides compared to peptides carrying the single modifications. Together, this paints a clear picture that ZCWPW1 likely directly recognises sites possessing the dual histone marks deposited by the PRDM9-HELLS complex (*Spruce et al., 2020*), the first such protein identified. Our evolutionary analyses suggest that this function might occur across almost all species where PRDM9 deposits these marks, even species where PRDM9 does not specify DSB locations (*Baker et al., 2017*).

Although it was suggested that ZCWPW1 might recruit the DSB machinery to PRDM9 binding sites (*Li et al., 2019a*; *Spruce et al., 2020*), we and *Mahgoub et al., 2020* (using distinct assays) find that DSB positioning is completely unaffected by loss of ZCWPW1. However, the additional role PRDM9 plays besides DSB positioning, in enabling homologous chromosome pairing (*Davies et al., 2016*; *Hinch et al., 2019*; *Li et al., 2019b*), apparently malfunctions in *Zcwpw1$^{-/-}$* mice. Our DMC1 data, together with microscopy data, are most easily explained if, unlike in WT, DSBs at sites bound more strongly by PRDM9 do not repair more quickly. Instead, in *Zcwpw1$^{-/-}$* males, all DSBs behave similarly to those on the X-chromosome — which repairs DSBs very slowly, even in WT animals, and does not synapse. This implies an unexpected role for ZCWPW1: in aiding PRDM9 in its second function, synapsis. ZCWPW1 therefore offers a (thus far) unique protein, directly recruited by PRDM9 binding and so linking PRDM9 binding with its downstream functions in aiding rapid DSB processing/repair at hotspots whose homologue is bound by PRDM9 (*Davies et al., 2016*; *Hinch et al., 2019*; *Li et al., 2019b*).

PRDM9 is thought to aid homology search/chromosome pairing by such binding to the homologous chromosome, and hybrids where hotspots are highly 'asymmetric' (i.e. the homologues are not bound at the sites where DSBs occur, due to evolutionary hotspot erosion) show multiple features identical to *Zcwpw1$^{-/-}$* mice. These include asynapsis, persistent DMC1 foci (at the asymmetric hotspots), and complete sterility in males only (*Davies et al., 2016*). In *Zcwpw1$^{-/-}$* mice, unlike in WT mice, there are many persistent DMC1 foci and we observe that even strong PRDM9 binding to the homologue causes no reduction in DMC1 persistence. These mice therefore behave, in many ways, as if all hotspots are completely asymmetric. Thus, a parsimonious explanation of our findings is that ZCWPW1 functions by attaching to PRDM9-bound sites on the matching homologous chromosome, thus marking possible repair partner locations for DSB sites, to help pair homologues. This might for example lead to recruitment to the chromosomal axis (where DMC1 foci are found), for potential use as the homologous repair template of DSBs, perhaps mediated by the SCP-1-like domain identified in some (although not all) orthologous copies. Given we observe ZCWPW1 throughout the nucleus in early meiosis, such recruitment would likely be transient in nature. Alternatively, ZCWPW1's part in mediating PRDM9's downstream roles might also depend (or even, depend only) on it binding the chromosome on which DSBs occur.

Given that we still observe some synapsis, similar to *Prdm9$^{-/-}$* mice, we suggest that a 'back-up' mechanism, which is independent of PRDM9 binding level, is acting to attempt homology search/

chromosome pairing in *Zcwpw1*$^{-/-}$ mice. In fact, some such mechanism is likely necessary, to repair DSBs happening within a subset of hotspots that are asymmetric due to naturally occurring variation (as seen in for example humans and heterozygous mice). If our observed >9 fold changes in relative DMC1 levels for strongly vs weakly bound hotspots purely reflect increased DMC1 persistence, it might be that this mechanism is greatly slower than the PRDM9-dependent pathway for WT mice. In turn, this could explain many – or all – of the downstream phenotypes in these mice, as consequences of delayed homology search.

The simplest explanation of our observation that ZCWPW1 binds weakly to many sites in a CpG-dependent manner in HEK293T cells is that these sites might be directly bound by a methyl-CpG binding domain (MBD) putatively observed in the human, mouse, and coelacanth *Zcwpw1*, among others. Alternatively, recruitment might be indirect, with ZCWPW1 forming a complex with an MBD containing protein, for example SETDB1 which is expressed in HEK293T cells, co-expressed with ZCWPW1 and PRDM9 (*Jung et al., 2019*; *Schultz et al., 2002*), and interacts with several other identified PRDM9 partner proteins (*Mulligan et al., 2008*; *Parvanov et al., 2017*). Given the precisely regulated developmental changes in methylation status, ZCWPW1/PRDM9 protein abundance, and chromatin during meiosis (*Gaysinskaya et al., 2018*; *Seisenberger et al., 2013*) it is unclear whether ZCWPW1 binding will also be affected by CpGs in vivo. Although more modest, *Huang et al., 2020* did also report enrichment of ZCWPW1 at B1 elements in mice, the closest equivalent of human Alus. Unlike *Mahgoub et al., 2020*, *Huang et al., 2020* also observed overlap of ZCWPW1 peaks with CpG islands (CGI) more frequently than expected. Given (i) they and *Mahgoub et al., 2020* detect a much lower number of peaks (14,688 and 4,487 respectively) than we do in our HEK293T system (800,000), and (ii) we only find Alu enrichment in weaker ZCWPW1 peaks (of our top 10,000 peaks, >90% would be PRDM9-bound sites), these results are not conflicting, and instead might depend on detection/behaviour of weaker peaks, and/or presence of PRDM9 in ZCWPW1-containing cells. If CpG dinucleotides do (although weakly, relative to PRDM9) affect in vivo ZCWPW1 binding affinity and given that CpG methylation is associated with transposons, it is interesting to speculate whether ZCWPW1 might play a role in germline transposon recognition, silencing, or facilitate repair of DSBs occurring within transposable elements.

*Zcwpw1* possesses a paralogue, *Zcwpw2*, which is also co-expressed in testis with *Prdm9* (although at a lower level), contains both CW and PWWP domains, and is able to recognise H3K4me3 at least (*Liu et al., 2016*). Further studies are required to investigate what function if any *Zcwpw2* has in meiosis. One interesting possibility is that ZCWPW2, rather than ZCWPW1, might act to help position DSBs – if so, its low abundance relative to ZCWPW1 would be analogous to the low number of DSBs per meiosis (~300) relative to the number of PRDM9 binding sites.

In this study, we have shown that ZCWPW1 co-evolves with PRDM9, binds to sites marked by PRDM9, and is required for the proper processing, removal of DMC1, and repair of DSBs. How exactly it mediates repair, if for example by recruitment of other proteins, remains to be determined. This study also further demonstrates that genes co-expressed with *Prdm9* represent a rich source of undiscovered meiotic genes, with important functional implications for recombination, meiosis progression and ultimately fertility. ZCWPW1 impacts synapsis, downstream of DSB formation. Therefore the protein(s) responsible for recognising PRDM9-marked sites for DSB formation, and for aiding synapsis, are at least partly different, and we anticipate that future studies will likely uncover additional PRDM9-recruited proteins.

## Materials and methods

**Key resources table**

| Reagent type (species) or resource | Designation | Source or reference | Identifiers | Additional information |
|---|---|---|---|---|
| Gene (167 species) | *ZCWPW1* | This paper using BlastP and tBLASTn (www.blast.ncbi.nlm.nih.gov), NCBI (www.ncbi.nlm.nih.gov) and Ensembl (www.ens.embl.org) | Details in Materials and methods | Also see *Figure 1—source data 1* |

*Continued on next page*

*Continued*

| Reagent type (species) or resource | Designation | Source or reference | Identifiers | Additional information |
|---|---|---|---|---|
| Gene (225 species) | *PRDM9* | *Baker et al., 2017* (doi: 10.7554/eLife.24133) | | Also see *Figure 1—source data 2* |
| Genetic reagent (*Mus musculus*) | *Zcwpw1^{-/-}* | Toronto Centre for Phenogeno-mics (Canada) | RRID:IMSR_CMMR:ADVN; Strain name C57BL/ 6N-Zcwpw1^{em1(IMPC)Tcp} | Constitutive knock out for Zcwpw1 carrying a 1485bp CRISPR/ Cas9-induced deletion (chr5:137799545 –13780101029) |
| Genetic reagent (*Mus musculus*) | *Prdm9^{-/-}* | RIKEN BioResource Research Center (Japan) | RRID:MGI:3624989; Strain name B6.129P2- Prdm9 < tm1Ymat>, strain number RBRC05145 | Originating article *Hayashi et al., 2005*. |
| Cell line (*Homo sapiens*) | Embryonic Epithelial Kidney | ATCC | Cat. CRL-3216 | |
| Transfected construct (*Homo sapiens*) | hPRDM9-V5-YFP | *Altemose et al., 2017* (doi: 10.7554/eLife.28383) | | Human *PRDM9* B allele cloned into pLENTI CMV/TO Puro DEST vector (Addgene plasmid #17293; *Campeau et al., 2009*) in frame with a Twin-strep tag, a V5 tag, and a self-cleaving YFP tag due to the presence of an upstream P2A sequence |
| Transfected construct (*Homo sapiens*/ Pan troglodyte hybrid) | cPRDM9-V5-YFP | *Altemose et al., 2017* (doi: 10.7554/eLife.28383) | | hPRDM9-V5-YFP construct where Exon 10 encoding the human zinc finger array was replaced with the equivalent sequence from the chimp *PRDM9* w11a allele |
| Transfected construct (*Homo sapiens*) | hZCWPW1-HA | GenScript | Clone ID OHu16813 | *ZCWPW1*, transcript variant 1, mRNA (NM_017984.5) cloned into pCDNA3.1^+/C-HA |
| Transfected construct (*Homo sapiens*) | hZCWPW2-HA | GenScript | Clone ID OHu31001C | *ZCWPW2*, transcript variant 1, mRNA (NM_001040132.3) cloned into pCDNA3.1^+/C-HA |
| Transfected construct (*M. musculus*) | mZCWPW1-FLAG | OriGene | Clone ID MR209594 | *Zcwpw1*, Transcript variant 2 mRNA (NM_001005426) cloned with a C-terminal Myc- DDK(FLAG) tag |
| Transfected construct (*M. musculus*) | mZCWPW2-FLAG | This paper | pCMV6-Entry (OriGene, Cat. PS100001) | Generated by cloning custom-synthesised mZCWPW2 into pCMV6-Entry |
| Transfected construct (*M. musculus*) | mZCWPW1-His | This paper | pET22b(+) Novagen (Sigma-Aldrich, Cat. 69744) | Generated by sub-cloning mZCWPW1 from clone ID MR209594 (OriGene) into pET22b(+) in frame with a C-terminal 6-histidines tag |
| Recombinant DNA reagent | mZCWPW2 | Origene | Mouse *Zcwpw2*-206; Transcript ID ENSMUST00000238919.1 | Custom synthesis of full-length cDNA sequence |
| Strain, strain background (*Escherichia coli*) | BL21(DE3) | Thermo Fisher Scientific | Cat. C600003 | Chemically competent cells |
| Peptide, recombinant protein | mZCWPW1-His | This paper | | Used to produce a rabbit polyclonal antibody against mouse ZCWPW1 by immunisation (Eurogentec) |

*Continued on next page*

*Continued*

| Reagent type (species) or resource | Designation | Source or reference | Identifiers | Additional information |
|---|---|---|---|---|
| Antibody | Anti-mouse ZCWPW1 antiserum, and pre-immune serum (rabbit polyclonal) | This paper | Custom generation (Eurogentec) | IF (1:100), WB (1:1000), IP (5 µl on transfected cells, 10 µl on mouse testis) |
| Antibody | Anti-Human ZCWPW1 (mouse monoclonal) | Sigma-Aldrich | Cat. SAB1409478 | WB (1:2000) |
| Antibody | Anti-SYCP3 (mouse monoclonal) | Santa Cruz Biotechnology | Cat. sc-74569, RRID:AB_2197353 | IF (1:100) |
| Antibody | Anti-SYCP3 (biotinylated, rabbit polyclonal) | Novus | Cat. NB300-232, RRID:AB_2087193 | IF (1:100) |
| Antibody | Anti-DMC1 (rabbit polyclonal) | Santa Cruz Biotechnology | Cat. sc-22768, RRID:AB_2277191, Discontinued | IF (1:100) |
| Antibody | Anti-DMC1 2H12/4 (mouse monoclonal) | Novus | Cat. NB100-2617, RRID:AB_2245859 | ChIP (5 µg) |
| Antibody | Anti-HORMAD2 (rabbit polyclonal) | Santa Cruz Biotechnology | Cat. sc-282192, RRID:AB_2121124 | IF (1:300) |
| Antibody | Anti-RAD51 (mouse monoclonal) | Abcam | Cat. ab88572, RRID:AB_2042762 | IF (1:50) |
| Antibody | Anti-RPA2 (rabbit polyclonal) | Abcam | Cat. ab10359, RRID:AB_297095 | IF (1:1000) |
| Antibody | Anti-phospho-H2AX (mouse monoclonal) | Sigma-Aldrich | Cat 05–636, RRID:AB_309864 | IF (1:250) |
| Antibody | Anti-phospho γ-H2AX (chicken polyclonal) | Biorbyt | Cat. orb195374 Discontinued | IF (1:1000) |
| Antibody | Anti-rabbit IgG Alexa Fluor 488 secondary (goat polyclonal) | Thermo Fisher Scientific | Cat. A-11008, RRID:AB_143165 | IF (1:250) |
| Antibody | Anti-mouse IgG Alexa Fluor 488 secondary (goat polyclonal) | Thermo Fisher Scientific | Cat. A-11001, RRID:AB_2534069 | IF (1:250) |
| Antibody | Anti-rabbit IgG Alexa Fluor 594 secondary (goat polyclonal) | Thermo Fisher Scientific | Cat. A-11012, RRID:AB_141359 | IF (1:250) |
| Antibody | Anti-mouse IgG Alexa Fluor 594 secondary (goat polyclonal) | Thermo Fisher Scientific | Cat. A-11005, RRID:AB_141372 | IF (1:250) |
| Antibody | Anti-mouse IgG Alexa Fluor 647 secondary (goat polyclonal) | Thermo Fisher Scientific | Cat. A-21235, RRID:AB_2535804 | IF (1:250) |
| Antibody | Anti-chicken IgY Alexa Fluor 647 secondary (goat polyclonal) | Thermo Fisher Scientific | Cat. A-21449, RRID:AB_2535866 | IF (1:250) |
| Antibody | Streptavidin, Alexa Fluor 647 | Thermo Fisher Scientific | Cat. S32357 | IF (1:50) |
| Antibody | Anti-poly-His (mouse monoclonal) | Sigma-Aldrich | Cat. H1029, RRID:AB_260015 | WB (1:2000) |
| Antibody | Anti-HA (rabbit polyclonal) | Abcam | Cat. ab9110, RRID:AB_307019 | IF (1:100), WB (1:1000), IP (2 µg), ChIP (5 µg) |
| Antibody | Anti-HA (mouse monoclonal) | Sigma-Aldrich | Cat. H3663, RRID:AB_262051 | IF (1:500) |
| Antibody | Anti-V5 (rabbit polyclonal) | Abcam | Cat. ab9116, RRID:AB_307024 | IF (1:500) |

*Continued on next page*

*Continued*

| Reagent type (species) or resource | Designation | Source or reference | Identifiers | Additional information |
|---|---|---|---|---|
| Antibody | Anti-FLAG M2 (mouse monoclonal) | Sigma-Aldrich | Cat. F3165, RRID:AB_259529 | IF (1:500), WB (1:2000), IP (3 μg) |
| Antibody | Anti-β-Actin (mouse monoclonal) | Sigma-Aldrich | Cat. A1978, RRID:AB_476692 | WB (1:2000) |
| Antibody | ECL Rabbit IgG, HRP-linked whole Ab (donkey polyclonal) | GE Healthcare | Cat. NA934, RRID:AB_772206 | WB (1:10000) |
| Antibody | ECL Mouse IgG, HRP-linked whole Ab (sheep polyclonal) | GE Healthcare | Cat. NA931, RRID:AB_772210 | WB (1:10000) |
| Sequence-based reagent | pIRESMinor | *Chan et al., 2017* | biotin labelled minor satellite probe | |
| Sequence-based reagent | GAPDH_F (Human) | OriGene | PCR primers, transcript detection, NM_002046 | GCTCCTCTGACTT CAACAGCGGCT |
| Sequence-based reagent | GAPDH_R (Human) | OriGene | PCR primers, transcript detection, NM_002046 | ACCACCCTGTTG CTGTAGCCAA |
| Sequence-based reagent | PRDM9_F (Human) | OriGene | PCR primers, transcript detection, NM_020227 | ACGAAGAGGCAG CCAACAATGG |
| Sequence-based reagent | PRDM9_R (Human) | OriGene | PCR primers, transcript detection, NM_020227 | GCCACCAGGTT CTGCTCTTCAT |
| Sequence-based reagent | ZCWPW1_F (Human) | OriGene | PCR primers, transcript detection, NM_017984 | GATGGCTCAAGA GGCAGAACAG |
| Sequence-based reagent | ZCWPW1_R (Human) | OriGene | PCR primers, transcript detection, NM_017984 | TGGGCTGTTCAA ACCAGAGAGC |
| Sequence-based reagent | ZCWPW2_F (Human) | OriGene | PCR primers, transcript detection, NM_001040432 | AAGAGCTGGAG CAAATGCTGCAG |
| Sequence-based reagent | ZCWPW2_R (Human) | OriGene | PCR primers, transcript detection, NM_001040432 | CAGGAGCTTCTG GGCTGCATTT |
| Commercial assay or kit | Telomere PNA FISH Kit/Cy3 | Agilent | Cat. K5326 | |
| Commercial assay or kit | Pierce BCA protein assay kit | Thermo Fisher Scientific | Cat. 23227 | |
| Commercial assay or kit | ECL Prime Western Blotting Detection Reagent | GE Healthcare | Cat. 10308449 | |
| Commercial assay or kit | Minelute Reaction Cleanup Kit | QIAGEN | Cat. 28204 | |
| Commercial assay or kit | Qubit dsDNA HS Assay kit | Thermo Fisher Scientific | Cat. Q32851 | |
| Chemical compound, drug | IPTG | Sigma-Aldrich | Cat. I5502 | 0.5 mM final |
| Other | Fast SYBR Green Master Mix | Applied Biosystems | Cat. 4385610 | RNA extraction and RT-qPCR |
| Other | Dynabeads M-280 Sheep anti-Rabbit IgG | Thermo Fisher Scientific | Cat. 11203D, RRID:AB_2783009 | IP and ChIP experiments; IP (25–75 ul), ChIP (65 ul) |
| Other | Dynabeads M-280 Sheep anti-Mouse IgG | Thermo Fisher Scientific | Cat. 11202D, RRID:AB_2783640 | IP and ChIP experiments; IP (25 ul), ChIP (65 ul) |

*Continued on next page*

*Continued*

| Reagent type (species) or resource | Designation | Source or reference | Identifiers | Additional information |
|---|---|---|---|---|
| Other | TALON Metal Affinity Resin | Takara | Cat. 635502 | Expression and purification of ZCWPW1 recombinant protein; 2 ml per L of IPTG-induced bacterial culture |
| Other | TRI Reagent | Sigma-Aldrich | Cat. T9424 | RNA extraction and RT-qPCR |
| Other | Protease Inhibitor Cocktail | Sigma-Aldrich | Cat. P8340 | IP and WB detection; 1:100 dilution |
| Other | Complete Mini Protease Inhibitor Cocktail | Sigma-Aldrich | Cat. 11697498001 | ChIP; 1 tablet in 10 ml volume |
| Other | Novex WedgeWell 4%to 20%, Tris-Glycine, Protein Gel | Thermo Fisher Scientific | Cat. XP04200BOX | IP and WB detection |
| Other | Novex WedgeWell 8%, Tris-Glycine, Protein Gel | Thermo Fisher Scientific | Cat. XP00080BOX | IP and WB detection |
| Software, Algorithm | MAPeakCaller | *Altemose et al., 2017* (doi: 10.7554/eLife.28383) | https://github.com/MyersGroup/PeakCaller/ (archived at https://doi.org/10.5281/zenodo.3783600) | |
| Software, Algorithm | BWA MEM | *Li, 2013* (arXiv:1303.3997) | bwa mem (version 0.7.17-r1188) | |
| Software, Algorithm | bwtool | *Pohl and Beato, 2014* (doi:10.1093/bioinformatics/btu056) | RRID:SCR_003035; v 1.0 | https://github.com/CRG-Barcelona/bwtool |
| Software, Algorithm | Picard | 'Picard Toolkit.' 2019. Broad Institute, GitHub Repository. http://broadinstitute.github.io/picard/; Broad Institute | RRID:SCR_006525; version 2.20.4-SNAPSHOT | |
| Software, Algorithm | SAMtools | PMID:19505943 | RRID:SCR_002105; v1.9 | https://www.htslib.org/download/ |
| Software, Algorithm | BEDtools | *Quinlan and Hall, 2010* (doi:10.1093/bioinformatics/btq033) | RRID:SCR_006646; v2.28.0 | bedtools.readthedocs.io |
| Software, Algorithm | SEQkit | *Shen et al., 2016* (doi:10.1371/journal.pone.0163962) | | |
| Software, Algorithm | IGV | *Thorvaldsdóttir et al., 2013* (doi: 10.1093/bib/bbs017) | | |

## Orthologue alignment

We identified ZCWPW1 orthologues across species using four data sources: first, we used BlastP (blast.ncbi.nlm.nih.gov), against the full-length human reference ZCWPW1 sequence (identifier NP_060454.3, against nr_v5 database) storing the top 1000 hits using the default parameters (set 1). Secondly, we downloaded the two sets of Ensembl identified ZCWPW1 orthologues (against ENSMUSG00000037108.13; ensembl.org, 105 orthologues), and identified NCBI ZCWPW1 protein orthologues (146 species, www.ncbi.nlm.nih.gov). Initial examination of identified orthologues revealed conservation among orthologues mainly of the CW and PWWP domains; therefore, to find additional orthologous sequences we performed tBLASTn (against the nr/nt nucleotide collection 17[th] July 2019; top 1000 hits) to identify orthologues in the NCBI nucleotide database, to the partial sequence (amino acids 256–339 of the reference sequence NP_060454.3), corresponding to the CW and PWWP domains of ZCWPW1. Protein sequences were then aligned against full-length human ZCWPW1 (NP_060454.3) using BLASTP2.9 (*Altschul et al., 2005*; *Altschul et al., 1997*). We

obtained taxonomy information from the NIH classification https://ftp.ncbi.nlm.nih.gov/pub/taxonomy/new_taxdump/, for comparisons among species and between ZCWPW1 and PRDM9.

Previous work has identified highly conserved 'KWR' and 'PWWP' patterns among CW and PWWP domains, respectively (*He et al., 2010b*; *Qin and Min, 2014*). We thus identified an initial set of 'clear' ZCWPW1 orthologues, and then used these to identify further less precise matches. 'Clear' orthologues are defined as proteins within set one containing perfect matches to both these sequences, and such that in the NCBI alignment, at least 39% of each sequence aligns to the human protein, and conversely. This second step is required to avoid spurious matches to, for example, ZCWPW2, which overlaps 19% of ZCWPW1, by requiring >2 fold this match length; although several *ZCWPW2* copies are in our initial list, no gene annotated as most similar to *ZCWPW2* attains the 39% overlap. For proteins within the 'clear' set, we identified the longest alignment for each given species, resulting in 136 species with a likely ZCWPW1 orthologue. This initial screen identified three groups of fish, placental mammals, marsupials, monotremes and reptiles as likely possessing ZCWPW1, similar to our final conclusions. Moreover, it defined amino acids 247–428 in the human reference sequence as being conserved in the alignment (at most one sequence not aligning), including the annotated CW and PWWP domains.

Using this initial set, we identified additional orthologous sequences and refined our results, by identifying conserved bases within ZCWPW1. Specifically, we divided the 136 species into major clades, as in *Baker et al., 2017*, and gave each sequence a weighting so that the overall weight for each clade was the same (so, for example, each Placental mammal sequence was downweighted as this clade was over-represented). Within the consistently aligned region 247–428, we calculated the overall weighted probability of each of the 20 amino acids, or a gap (adding $10^{-5}$ to exclude zero weights). This identified 31 completely conserved amino acids, and 78 amino acids whose entropy was below 1 (equivalent to two amino acids having equal probability, so implying one amino acid present in >50% of species). Finally, we defined ZCWPW1 orthologues as those sequences matching at least 90% of those 31 perfectly conserved bases which were aligned, and aligning to at least 50% of these bases (this last condition allows for inclusion of incomplete sequencing or protein assembly). We note that while this 90% condition is arbitrary, by definition all annotated orthologues exceed this threshold, while thresholds below ~70–80% are exceeded by orthologues of *ZCWPW2* among other genes, offering some justification. Nonetheless, it is worth pointing out that our analysis does not rule out the existence of more poorly conserved *ZCWPW1* copies in more distantly related species. This approach identified a final set of 167 genes, which we annotated as likely *ZCWPW1* orthologues and used for the majority of results. For each *ZCWPW1* orthologue, we also identified the taxonomic relationship of the closest species possessing such a *PRDM9* copy (*Figure 1—source data 1*). We also reciprocally annotated each PRDM9 copy previously identified (*Baker et al., 2017*) according to the closest species also possessing ZCWPW1 (*Figure 1—source data 1*).

Our analyses revealed a relationship between ZCWPW1 predicted functional domains and PRDM9 histone modifications. Therefore, we identified additional potential conserved sites by identifying perfectly conserved bases among those species possessing *both* ZCWPW1 and PRDM9 orthologues, and where three key SET domain catalytic amino acids within PRDM9 are intact, meaning PRDM9 is predicted to have normal histone modification activity (*Baker et al., 2017*). This identified a slightly larger number of conserved amino acids (37). Only eight of these varied in any of the 167 species with a potential ZCWPW1 orthologue: 260, 404 and 411 (in two *Xenopus* frogs, with 411 also in white-headed capuchin), 19 and 22 (in three canines), 25 (in Ocelot gecko), 257 (in Anolis lizard and in Wombat), 325 (in elephantfish). While these changes might alter ZCWPW1 function, the true behaviour of ZCWPW1 is uncertain in these cases, although it is interesting that canids and frogs represent clades that all appear to have lost PRDM9, and possess multiple, clustered, amino acid changes.

## Domain search

pDomThreader (*Lobley et al., 2009*) was used via the PSIPRED server (http://bioinf.cs.ucl.ac.uk/psipred/) on the following uniProt amino acid sequences (Q9H0M4|ZCPW1_HUMAN, Q6IR42|ZCPW1_MOUSE, E2RFJ2|E2RFJ2_CANLF, M3XJ39_LATCH, A0A3S5ZP38_BOVIN, M3WDY6_FELCA, and G3ULT5|G3ULT5_LOXAF) all of which identified a match to 1ub1A00 in the C terminal section after the PWWP domain, except in LOXAF in which this match was slightly below the p-value threshold of 0.001.

SCP1 prediction used NCBI conserved domain search server (https://www.ncbi.nlm.nih.gov/Structure/cdd/wrpsb.cgi?) with the uniProt amino acid sequences Q9H0M4|ZCPW1_HUMAN and Q6IR42|ZCPW1_MOUSE.

## Mice and genotyping

KO mice for *Zcwpw1* (C57BL/6N-Zcwpw1^em1(IMPC)Tcp) were generated by the Toronto Centre for Phenogenomics (Canada). A 1485bp deletion on chromosome 5 spanning exons 5 to 7 of *Zcwpw1* (chr5:137799545–13780101029) was engineered by CRISPR/Cas9 using guide RNAs 5'-GACTGCAC TCACGGCCATCT-3' and 5'-GCCCGGTTCTTCATCCAATT-3'. The frameshift deletion introduces a stop codon in Exon 8, leading to a predicted unstable short truncated 492bp transcript. Mice were genotyped at the *Zcwpw1* locus using the following primers, and standard cycling conditions: KO allele-Forward, 5'-CACAGGCTCATGTATGTTTGTCTC-3'; KO allele-Reverse, 5'-CTGCTTCGTCCTC TTTCCTTATCTC-3'; WT allele-Forward, 5'-TGCCACCACACTTCATTTGT-3'; WT allele-Reverse CC TGTTTCCTTCCCAACTCA-3'. The deletion was verified by direct Sanger sequencing of the KO genotyping PCR product (Source Bioscience, UK), following purification with the QIAquick PCR Purification Kit (Qiagen). Sequence analysis was carried out using Chromas LITE (version 2.1.1).

KO mice for *Prdm9* were described previously (*Hayashi et al., 2005*; *Mihola et al., 2019*) and obtained from the RIKEN BioResource Research Center in Japan (strain name B6.129P2-Prdm9 < tm1Ymat>, strain number RBRC05145). Mice were genotyped at the *Prdm9* locus using the following primers, and standard cycling conditions: WT allele-Forward 5'-AGGAATCTTCCTTCC TTGCTGTCG-3'; WT allele-Reverse 5'-ATTTCCCTGTATCTTCTTCAGGACT-3'; KO allele-Reverse 5'-CGCCATTCAGGCTGCGCAACTGTT-3'.

All animal experiments received local ethical review approval from the University of Oxford Animal Welfare and Ethical Review Body (Clinical Medicine board) and were carried out in accordance with the UK Home Office Animals (Scientific Procedures) Act 1986.

## Fertility measurements

Fertility was assessed in mice ranging from 9 to 12 weeks of age, either by mating with WT littermates and recording the average litter size and frequency, or by measuring paired testes weight (normalized to lean body weight), sperm count (per paired epididymides) and chromosome synapsis rate (by immunostaining of pachytene spermatocytes) in males. Lean body weight was measured using the EchoMRI-100 Small Animal Body Composition Analyzer.

## Immunostaining of spermatocytes

Mouse testis chromosome spreads were prepared using surface spreading (*Barchi et al., 2008*; *Peters et al., 1997*) and immunostained as previously described (*Davies et al., 2016*). The following primary antibodies were used: custom ZCWPW1 rabbit antiserum (1:100 dilution), mouse anti-SYCP3 (Santa Cruz Biotechnology sc-74569, RRID:AB_2197353) or biotinylated rabbit anti-SYCP3 (Novus NB300-232, RRID:AB_2087193), rabbit anti-DMC1 (Santa Cruz Biotechnology sc-22768, RRID:AB_2277191, discontinued; 1:100 dilution), rabbit anti-HORMAD2 (Santa Cruz Biotechnology sc-82192, RRID:AB_2121124; 1:300 dilution), mouse anti-RAD51 (Abcam ab88572, RRID:AB_2042762; 1:50 dilution), rabbit anti-RPA2 (Abcam ab10359, RRID:AB_297095; 1:1000 dilution), mouse (Sigma-Aldrich 05–636, RRID:AB_309864; 1:250 dilution) or chicken (Biorbyt orb195374, discontinued; 1:1000 dilution) anti-phospho γ-H2AX and Alexa Fluor 488-, 647- or 594-conjugated secondary antibodies against rabbit, mouse or chicken IgG/Y (A-11008, RRID:AB_143165; A-11001, RRID:AB_2534069; A11012, RRID:AB_141359; A-1105, RRID:AB_141372; A-21235, RRID:AB_2535804; A-21449, RRID:AB_2535866; Thermo Fisher Scientific; 1:250 dilution), as well as streptavidin Alexa Fluor 647 (Thermo Fisher Scientific) were used to detect the primary antibodies. Images were acquired using either a BX-51 upright wide-field microscope equipped with a JAI CVM4 B and W fluorescence CCD camera and operated by the Leica Cytovision Genus software, or a Leica DM6B microscope for epifluorescence, equipped with a DFC 9000Gt B and W fluorescence CCD camera, and operated via the Leica LASX software. Image analysis was carried out using Fiji (ImageJ-win64). The specificity of the ZCWPW1 signal was verified using the pre-immune serum from the same rabbit: no staining was visible at any stage of prophase I (*Figure 2—figure supplement 1B* shows representative images of mid-zygotene and early pachytene cells). Staging and sub- (early, mid and

late) staging of spermatocytes during prophase I was carried out according to defined staining patterns against SYCP1, SYCP3, and DAPI-stained DNA appearance, and morphological criteria previously published (*Gaysinskaya et al., 2014*). Briefly, cells showing large chromocentres (brightly stained by DAPI and corresponding to pericentromeric regions) with short tracts of SYCP3 were assigned as (mid-)leptotene. Cells with larger pooled chromocentres, associated with longer tracts of SYCP3 were identified as early zygotene. Cells where some synapsis was initiated (with more than two paired SYCP3 tracts, resulting in a thicker synaptonemal complex region), were assigned as (mid-)zygotene. Cells where the majority of chromosomes were synapsed, with three or less completely or partially asynapsed, were identified as late zygotene.

## Fluorescent in-situ hybridisation (FISH)

Following immunostaining of spermatocytes, centromeres and telomeres were labelled using a biotin labelled minor satellite probe (pIRESMinor, *Chan et al., 2017*) together with the Telomere PNA FISH Kit/Cy3 (Agilent), following the manufacturer's instructions, but without protease treatment. The hybridisation and signal detection were carried out using standard techniques.

## Plasmids

Constructs encoding full-length human PRDM9 (B allele) in frame with a C-terminal V5 tag and a self-cleaving YFP tag (hPRDM9-V5-YFP), or the chimp construct (cPRDM9-V5-YFP) where the sequence that encodes the human zinc finger array in hPRDM9-V5-YFP was replaced with the equivalent sequence from the chimp PRDM9 w11a allele, for expression in mammalian cells (pLENTI CMV/TO Puro DEST backbone vector) were described previously (*Altemose et al., 2017*). Constructs encoding full-length human (h) and mouse (m) ZCWPW1 or ZCWPW2 for expression in mammalian cells were purchased from GenScript (hZCWPW1 and hZCWPW2 cDNA with a C-terminal HA tag in pCDNA3.1 clones ID OHu16813 and OHu31001C, respectively) and OriGene (mZCWPW1 cDNA with a FLAG-Myc dual tag in pCMV6-Entry; clone ID MR209594), respectively. mZCWPW2 cDNA was synthesised by GenScript and subcloned into pCMV6-Entry in-frame with a C-terminal FLAG tag. For expression in *E. coli*, full-length mZCWPW1 cDNA was subcloned into the pET22b(+) vector (Novagen) with a C-terminal poly-His tag.

## ZCWPW1 antibody production and validation

pET22b-mZCWPW1-His construct was transformed into BL21 (DE3) *E. coli* cells (Thermo Fisher Scientific). Bacterial cultures were grown to a density with $O.D_{600}$ ~0.7, and expression of recombinant His-tagged ZCWPW1 protein was induced overnight at 20°C by addition of IPTG to 0.5 mM. The protein was purified using TALON metal affinity resin, according to the manufacturer's instructions (Takara 635502). Further purification was carried out by size exclusion chromatography (Superdex HiLoad 200 16/60, GE Life Sciences). WB validation was carried out as described previously (*Altemose et al., 2017*), using mouse anti-human ZCWPW1 (Sigma-Aldrich SAB1409478; 1:2000 dilution) and mouse anti-polyhistidine (Sigma-Aldrich H1029, RRID:AB_260015; 1:2000 dilution) antibodies. The purified protein was used to immunise two rabbits (Eurogentec, Belgium), and the resulting immune antisera (and pre-immune sera from the same rabbits) were tested against the recombinant antigen by ELISA (Eurogentec), the overexpressed FLAG-tagged mouse ZCWPW1 protein (and the closely related ZCWPW2 protein to address specificity) in transfected HEK293T cells, and the endogenous protein in mouse testes (*Figure 2—figure supplement 1*). Both antisera showed high titers and similar reactivity by IP, WB and IF staining. The antiserum with the lowest level of background (corresponding to the signal detected by the pre-immune serum) was chosen to carry out the experiments.

## Cell line, transfection and immunofluorescence staining

Human embryonic kidney (HEK) 293 T cells were purchased from the ATCC (ATCC CRL-3216), with a certificate of analysis confirming cell line identity by Short Tandem Repeat profiling and lack of mycoplasma contamination. All experiments were carried out on cells cultured for less than five passages from the purchased stock reference strain. Cells were cultured and transfected using Fugene HD as previously described (*Altemose et al., 2017*). High and comparable expression of the target proteins across all samples was verified by IF staining of duplicate transfected cultures before

proceeding to ChIP (*Figure 5—figure supplement 4*). PRDM9 expression was visualised either directly from live cells through YFP fluorescence, or by IF staining of fixed cells using rabbit anti-V5 (Abcam ab9116, RRID:AB_307024; 1:500 dilution) as described previously (*Altemose et al., 2017*). ZCWPW1-HA was detected by immunostaining as above, using mouse anti-HA (Sigma-Aldrich H3663, RRID:AB_262051, 1:500 dilution). The fraction of cells co-expressing ZCWPW1-HA and PRDM9-V5 was determined by co-staining with both antibodies, as above (*Figure 5—figure supplement 4*). To assess the specificity of the ZCWPW1 antiserum, cells transfected with FLAG-tagged mZCWPW1 or mZCWPW2 were co-immunostained with the antiserum (1:100 dilution) and a mouse anti-FLAG (Sigma-Aldrich F3165, RRID:AB_259529; 1:500 dilution). Images were acquired using a ZOE fluorescent cell imager (Bio-Rad).

## RNA extraction and reverse transcription-quantitative PCR (RT-qPCR)

RT-qPCR analysis of human *PRDM9*, *ZCWPW1* and *ZCWPW2* transcript expression in HEK293T cells was performed as previously described (*Altemose et al., 2017*), with minor modifications listed here. RNA was extracted from one biological replicate per sample using TRI Reagent (Sigma-Aldrich T9424), following the manufacturer's guidelines. For qPCR, each sample was analysed in triplicate (technical replicates) using primer sets from Origene: *GAPDH* (NM_002046) forward 5'-GCTCCTC TGACTTCAACAGCG-3' and reverse 5'-ACCACCCTGTTGCTGTAGCCAA-3'; *PRDM9* (NM_020227) forward 5'-ACGAAGAGGCAGCCAACAATGG-3' and reverse 5'-GCCACCAGGTTCTGCTCTTCAT-3'; *ZCWPW1* (NM_017984) forward 5'-GATGGCTCAAGAGGCAGAACAG-3' and reverse 5'-TGGGC TGTTCAAACCAGAGAGC-3'; *ZCWPW2* (NM_001040432) forward 5'- AAGAGCTGGAGCAAATGC TGCAG-3' and reverse 5'-CAGGAGCTTCTGGGCTGCATTT-3'. PCR reactions were carried out in triplicate (three technical replicates per sample), using Fast SYBR Green Master Mix according to the manufacturer's instructions (Applied Biosystems 4385610). Relative gene expression was calculated using the ΔΔCt method after averaging the technical replicates for each sample and normalising to the *GAPDH* gene. Raw Ct values and detailed calculations are given in *Figure 5—source data 1*.

## Immunoprecipitation and western blot detection

Cell and testes protein extracts were prepared in lysis buffer containing 50 mM Tris-HCl pH8.0, 150 mM NaCl, 1% Triton X-100 and a cocktail of protease inhibitors (Sigma-Aldrich P8340), followed by gentle rotation at 4°C for 30 to 45 min, respectively. Cell debris were pelleted by centrifugation at 4°C for 20 min at 20,000 g. Protein extracts were quantified using the Pierce BCA protein assay kit (Thermo Fisher Scientific 23227) and equal quantities were incubated overnight at 4°C with 5 or 10 µl (on transfected cells and mouse testis, respectively) of rabbit ZCWPW1 immune antiserum or pre-immune serum, or 3 µg of mouse anti-FLAG antibody (Sigma-Aldrich F3165, RRID:AB_259529), setting aside 5–100 µg of protein extract for direct western blot detection. Immunocomplexes were pulled down with 25–75 µl of Dynabeads M-280 sheep anti-rabbit (11203D, RRID:AB_2783009) or anti-mouse (11202D, RRID:AB_2783009) IgG (Thermo Fisher Scientific) for 2 hr at 4°C with gentle rotation. After five washes in lysis buffer, they were eluted from the beads by boiling in Laemmli sample buffer for 5 min. Immunocomplexes and total protein extracts were resolved on either 8% or 4–20% Novex Tris-Glycine precast gels (Thermo Fisher Scientific XP00080BOX, XP04200BOX). Proteins were transferred onto PVDF membranes and the proteins of interest were detected by western blotting following standard procedures. Blots were blocked for 1 hr at room temperature (RT) in PBS containing 0.2% Tween-20 (PBS-T) and 5% milk, and incubated for 1 hr at RT with rabbit ZCWPW1 immune antiserum or pre-immune serum (1:1000 dilution), mouse anti-β-actin (Sigma-Aldrich A1978, RRID:AB_476692, 1:2000 dilution), or mouse anti-FLAG antibody (Sigma-Aldrich F3165, RRID:AB_259529; 1,2000 dilution), washed three times in PBS-Tween buffer and incubated for 1 hr at RT with HRP-conjugated donkey anti-rabbit (NA934, RRID:AB_772206) or sheep anti-mouse (NA931, RRID:AB_772210) IgG antibody (1:10,000 dilution; GE Healthcare). Protein signals were revealed using the ECL Prime western blotting detection reagent according to the manufacturer's recommendations (GE Healthcare 10308449). The specificity of the ZCWPW1 signal observed in B6 testis extract was verified by immunoprecipitation using the pre-immune serum from the same rabbit: no signal was detected (*Figure 2—figure supplement 1C*). No cross-reactivity was observed with ZCWPW2 in either B6 testis extract or transfected HEK293T cells overexpressing FLAG-tagged ZCWPW2 (*Figure 2—figure supplement 1C,D*).

## ChIP-seq

### ChIP

ChIP against ZCWPW1-HA was carried out from transfected HEK293T cells as follows. Cells were crosslinked for 10 min in 1% formaldehyde, the reaction was quenched for 5 min by the addition of glycine to a final concentration of 125 mM, and the cells were washed twice in cold PBS. The cell pellet was resuspended in cold sonication buffer (50 mM Tris-HCl pH8, 10 mM EDTA, 1% SDS) supplemented with Complete Mini protease inhibitor cocktail (Sigma-Aldrich 11697498001), and chromatin was sheared to an average size of 200–500bp by sonication for 35 cycles (30 s ON/30 s OFF) using a Bioruptor Twin (Diagenode). After centrifugation for 10 min at 20,000 g, 4°C, the sonicate was diluted 10 fold in ChIP buffer (16.7 mM Tris pH8 1.2 mM EDTA, 167 mM NaCl, 1.1% Triton X-100) supplemented with protease inhibitors, pre-cleared for 2 hr at 4°C with 65 µl of Dynabeads M-280 sheep anti-rabbit IgG (Thermo Fisher Scientific 11203D, RRID:AB_2783009) and a 1% input chromatin sample was set aside. The rest of the sample was incubated overnight at 4°C with 5 µg of rabbit anti-HA antibody (Abcam ab9110, RRID:AB_307019). Immunocomplexes were washed once with each of low salt buffer (20 mM Tris pH8, 150 mM NaCl, 1% Triton X- 100, 0.1% SDS, 2 mM EDTA), high salt buffer (20 mM Tris pH8, 500 mM NaCl, 1% Triton X-100, 0.1% SDS, 2 mM EDTA), LiCl buffer (10 mM Tris pH8, 0.25 M LiCl, 1% NP-40, 1% sodium deoxycholate, 1 mM EDTA) and TE buffer (10 mM Tris pH8, 1 mM EDTA) and eluted from the beads in 100 mM NaHCO₃, 1% SDS for 30 min at 65°C with shaking. Both input and ChIP samples were reverse crosslinked overnight at 65°C in the presence of 200 mM NaCl, and proteins were digested for 90 min at 45°C by addition of proteinase K (0.3 mg/ml final concentration). DNA was purified using the MinElute Reaction Cleanup Kit (QIAGEN, 28204), and quantified using the Qubit dsDNA HS Assay Kit (Q32851) and a Qubit 2.0 Fluorometer (Thermo Fisher Scientific).

ChIP against DMC1 was performed from *Zcwpw1*⁻/⁻ testes using the published method by *Khil et al., 2012* with some modifications listed here. Chromatin shearing was carried out in 20 mM Tris-HCl pH8, 2 mM EDTA, 0.1% SDS using a Bioruptor Pico sonicator (Diagenode) for 4 cycles of 15 s ON/45 s OFF. ChIP was performed in 10 mM Tris-HCl pH8, 1 mM EDTA, 0.1% Sodium Deoxycholate, 1% Triton X-100, 500 mM NaCl using 5 µg of mouse anti-DMC1 2H12/4 (Novus NB100-2617, RRID:AB_2245859) pre-bound to 50 µl of Dynabeads M-280 sheep anti-mouse IgG (Thermo Fisher Scientific 11202D, RRID:AB_772210).

### Sequencing

ZCWPW1 ChIP and input libraries from transfected cells were prepared by the Oxford Genomics Centre at the Wellcome Centre for Human Genetics (Oxford, UK) using the Apollo Prep System (Wafergen, PrepX ILMN 32i, 96 sample kit) and standard Illumina multiplexing adapters following the manufacturer's protocol up to pre-PCR amplification, and sequenced on a HiSeq 4000 platform (75bp paired end reads, 48 million reads/sample). DMC1 ChIP libraries from *Zcwpw1*⁻/⁻ testes were prepared and sequenced as described previously (*Davies et al., 2016*) on an Illumina HiSeq2500 platform (Rapid Run, 51bp paired end reads, 110 million reads/sample).

## Read mapping

For the HEK293T experiments, reads were mapped to either hg38 (NCBI's GCA_000001405.15_GRCh38_no_alt_plus_hs38d1_analysis_set.fna.gz) using bwa mem (version 0.7.17-r1188) (*Li, 2013*). Duplicates were removed using picard's markDuplicates (version 2.20.4-SNAPSHOT). Unmapped, mate unmapped, non primary alignment, failing platform, and low MAPQ reads were removed using samtools with parameters '-q 30 F 3852 f 2 u' (version 1.9 (using htslib 1.9)). Other unmapped reads and secondary alignments were removed using samtools fixmate. Fragment position bed files were created using bedtools bamtobed (v2.28.0).

For the DMC1 mouse experimental data, we processed the data following the algorithm provided by *Khil et al., 2012* to map the reads to the mouse mm10 reference genome (*Lunter and Goodson, 2011*), and obtain type I reads.

## Peak calling

We called DMC1 peaks, as described previously (*Davies et al., 2016*). For the HEK293T experiments, peaks for ZCWPW1, PRDM9, H3K4me3, and H3K36me3 were called using a peak caller

previously described (*Davies et al., 2016*) and available at https://github.com/MyersGroup/PeakCaller (archived at https://doi.org/10.5281/zenodo.3783600). Single base peaks were called with parameters pthresh $10^{-6}$ and peakminsep 250 on 22 autosomes. For each experiment, we used available IP replicates, and sequenced input DNA to estimate background, in calling peaks.

Most of our analyses refer to these peaks, except in *Figure 5A,B* which plot the *change* in ZCWPW1 occupancy following PRDM9 transfection, for either the chimp or human *PRDM9* alleles. To estimate ZCWPW1 levels with vs without PRDM9, we called ZCWPW1 peaks in HEK293T cells also transfected with PRDM9 as before, but now replacing the 'input' lane with the IP data for ZCWPW1 in cells not transfected with PRDM9. *Figure 5—figure supplement 6* also uses this measure, of ZCWPW1 recruitment attributable to PRDM9 binding, to predict DMC1 levels, while *Figure 5—figure supplement 2* (right panel) checks that changes in ZCWPW1 occupancy occur largely at PRDM9-bound sites, and *Figure 5—figure supplement 1* shows the correlation of PRDM9 binding strength with the change in ZCWPW1 enrichment.

The peak calling algorithm allows for calculation of enrichment and p-values at arbitrary pre-specified locations or windows in the genome, we refer to these as 'force called'.

Note that the enrichment values from peak calling represent coverage due to signal, *after subtracting coverage due to background*, relative to background coverage. Hence a value less than one does not imply depletion — any value greater than 0 is enriched. For more details see *Altemose et al., 2017*.

## Enrichment profiles for HEK293T experiments

Peaks were filtered if: center within 2.5kb of PRDM9-independent H3K4me3 (promoters), input coverage $\leq 5$, in top five by likelihood, greater than 99.9 percentile input coverage. For human PRDM9 peaks the PRDM9 motif position (*Altemose et al., 2017*) was inferred using the getmotifs function in MotifFinder within the 300bp region around the peak (with parameters alpha = 0.2, maxits = 10, seed = 42, stranded_prior = T), and peaks were recentered and stranded at these locations. Peaks within 4kb of one another are removed to avoid double counting. Mean coverage was calculated with bwtool 1.0 using the aggregate command with parameters '-fill = 0 -firstbase' at a width of +/− 2kb. Each profile was normalised by the total coverage of all of the fragments and the normalised sample profile was divided by the relevant normalised comparison/control dataset to calculate enrichment. Random profiles were created using bedtools random with 10,000 locations, with seed 72346, and were also filtered such that none were within 4kb of one another.

## DSB profiles

Mapped type one reads were filtered by removing non-canonical chromosomes (with underscores in names), sex chromosomes, and mitochondrial reads. Bedgraphs were converted to bigWig format using UCSC bedGraphToBigWig v4. Profiles were created using bwtool 1.0 using the aggregate command with parameters 5000:2000 and -fill = 0. B6 WT hotspot locations were filtered to remove non-autosomal chromosomes, retain only B6 allele hotspots, and remove hotspots with PRDM9-independent H3K4me3. *Prdm9$^{-/-}$* hotspots were filtered to remove the X chromosome. Motif centered coordinates were used for WT hotspots where the motif positions were inferred using the MotifFinder software and PRDM9 position weight matrices from *Altemose et al., 2017*. To normalise background signal, the mean signal between −5000 and −3000 was subtracted from each strand-mouse combination. Mean coverage was normalised by the sum of coverage over each strand-mouse combination across both WT and KO locations.

## Mapping of alu CpGs

Alu locations were downloaded from UCSC tables and filtered for Alu repeats with a width between 250 and 350bp. DNA sequence was extracted from the genome (Ensembl 95 h38 primary assembly) at these locations using the bedtools getfasta command and CpG dinucleotides were counted using 'stringr::str_locate_all' command in R.

## CpG methylation

Data is from *Libertini et al., 2015* (GSE51867), non CpG methylated sites were removed, and the two strands were summed. High copy number (>1.5) CpGs were removed.

## DMC1 prediction using ZCWPW1 and PRDM9 binding strength

PRDM9-dependent ZCWPW1 enrichment was force-called at the human PRDM9 peaks. Human testis DMC1 sites are from *Pratto et al., 2014* (GSE59836) and were subsetted to 'A or B' autosomal peaks with a width less than 3kb. These peaks were then trimmed to a maximum of 800bp before overlapping with PRDM9 peaks to create a binary target variable. Regions with input coverage ≤5 in either predictor were removed in addition to outliers with input >200 and/or ZCWPW1 enrichment >10. A binomial generalised linear model with a logit link (logistic regression) was fitted using the 'glm' function from the 'stats' package in R (3.6.0). Chromosomes 1, 3, and 5 were used as the test data with the remaining autosomes being the training data.

## Heatmaps

Regions were extracted from bigWig using bwtool matrix-fill = 0 -decimals = 1 -tiled-averages=5, and width parameters 2000. Coverage was normalised by total coverage (scaled by $10^{10}$). A pseudo-count of 1 was added to both input and sample, and the sample was then normalised by the input for each region. Values outside the quantile range 0.01–0.99 were thresholded. The profile plots are created by taking the ratio of the mean coverage of the sample and input separately. Ordering of the regions was determined by the mean coverage of a 200bp window centered on the peak center.

## Software

Computational analysis was performed using R (*R Development Core Team, 2018*) and snakemake (*Köster and Rahmann, 2012*). BigWig aggregations by bwtool (*Pohl and Beato, 2014*), BED manipulation with BEDtools (*Quinlan and Hall, 2010*), and FASTA processing with seqkit (*Shen et al., 2016*). We also used BigWig tools (*Kent et al., 2010*), IGV (*Thorvaldsdóttir et al., 2013*), and GNU parallel (*Tange, 2018*). Plots were created using the ggplot2 package (*Wickham, 2016*) and extensions ggforce (*Pedersen, 2016*), RColorBrewer (*Neuwirth, 2014*), viridis (*Garnier, 2018*), drawProteins (*Brennan, 2018*), and cowplot (*Wilke, 2018*). In addition, the following R packages were used: data.table (*Dowle and Srinivasan, 2019*), mgcv (*Wood, 2011*), PRROC (*Grau et al., 2015*), sangerseqR (*Hill et al., 2014*), and ComplexHeatmap (*Gu et al., 2016*).

## External datasets summary

DMC1 SSDS ChIP-seq in human testis is from *Pratto et al., 2014* (GSE59836). DMC1 SSDS ChIP-seq (*Brick et al., 2012*) (GSE35498), H3K4me3 ChIP-seq (*Davies et al., 2016*) (GSE73833), and SPO11 oligo-seq (*Lange et al., 2016*) (GSE84689) are in WT B6 mouse testis. PRDM9 (h/c-V5 tagged), H3K4me3, and H3K36me3 ChIP-seq in HEK293T cells is from *Altemose et al., 2017* (GSE99407). Whole-genome bisulfite sequencing (BS-seq) of HEK293 cells is from *Libertini et al., 2015* (GSE51867).

## Acknowledgements

We thank the High-Throughput Genomics Group at the Wellcome Centre for Human Genetics (funded by Wellcome Trust grant 203141/Z/16/Z) for the generation of sequencing data. We also thank Dr. Benjamin Bishop and Prof. Christian Siebold in the Division of Structural Biology (Wellcome Centre for Human Genetics, Oxford) for their help with the purification of recombinant mouse ZCWPW1 (size exclusion) towards antibody production. Funding: This work was supported by the Wellcome Trust through Investigator Awards to S.R.M. (098387/Z/12/Z and 212284/Z/18/Z), a PhD Studentship to D.W. (109109/Z/15/Z), a Senior Investigator Award to P.D (095552/Z/11/Z), and core support grants to the Wellcome Centre for Human Genetics (20314/Z/16/Z and 090532/Z/09/Z).

## Additional information

### Funding

| Funder | Grant reference number | Author |
|--------|------------------------|--------|
| Wellcome | 098387/Z/12/Z | Simon R Myers |
| Wellcome | 212284/Z/18/Z | Simon R Myers |

| Wellcome | 109109/Z/15/Z | Daniel Wells |
| Wellcome | 095552/Z/11/Z | Peter Donnelly |

The funders had no role in study design, data collection and interpretation, or the decision to submit the work for publication.

## Author contributions

Daniel Wells, Conceptualization, Resources, Data curation, Software, Formal analysis, Funding acquisition, Validation, Investigation, Visualization, Methodology, Writing - original draft, Writing - review and editing; Emmanuelle Bitoun, Conceptualization, Resources, Data curation, Formal analysis, Funding acquisition, Validation, Investigation, Visualization, Methodology, Writing - original draft, Project administration, Writing - review and editing; Daniela Moralli, Resources, Data curation, Formal analysis, Validation, Investigation, Visualization, Methodology; Gang Zhang, Resources, Data curation, Investigation, Methodology; Anjali Hinch, Resources, Data curation, Software, Formal analysis, Validation, Investigation, Visualization, Methodology; Julia Jankowska, Resources, Data curation, Formal analysis, Validation, Investigation; Peter Donnelly, Catherine Green, Resources, Supervision; Simon R Myers, Conceptualization, Resources, Data curation, Software, Formal analysis, Supervision, Funding acquisition, Validation, Investigation, Visualization, Methodology, Writing - original draft, Project administration, Writing - review and editing

## Author ORCIDs

Daniel Wells (iD) https://orcid.org/0000-0002-2007-8978
Emmanuelle Bitoun (iD) https://orcid.org/0000-0003-3439-2113
Simon R Myers (iD) https://orcid.org/0000-0002-2585-9626

## Ethics

Animal experimentation: All animal experiments received local ethical review approval from the University of Oxford Animal Welfare and Ethical Review Body (Clinical Medicine board) and were carried out in accordance with the UK Home Office Animals (Scientific Procedures) Act 1986. The specific protocols used were authorised by the UK Home Office under Project Licence PPL 3003437.

## Decision letter and Author response

Decision letter https://doi.org/10.7554/eLife.53392.sa1
Author response https://doi.org/10.7554/eLife.53392.sa2

# Additional files

## Supplementary files

• Transparent reporting form

## Data availability

Source data files are provided for Figures 1-5. Raw and processed data for ChIP-seq (Figures 5-8) are available on the GEO database (identifier GSE141516). Codes used for analysis are available at https://github.com/MyersGroup/Zcwpw1 (copy archived at https://github.com/elifesciences-publications/Zcwpw1) and archived at Zenodo (DOI:https://doi.org/10.5281/zenodo.3559759).

The following dataset was generated:

| Author(s) | Year | Dataset title | Dataset URL | Database and Identifier |
| --- | --- | --- | --- | --- |
| Wells D, Bitoun E, Moralli D, Zhang G, Hinch AG, Donnelly P, Green C, Myers SR | 2019 | ZCWPW1 and DMC1 ChIP-seq data | https://www.ncbi.nlm.nih.gov/geo/query/acc.cgi?acc=GSE141516 | NCBI Gene Expression Omnibus, GSE141516 |

The following previously published datasets were used:

| Author(s) | Year | Dataset title | Dataset URL | Database and Identifier |
|---|---|---|---|---|
| Pratto F, Brick K, Khil P, Smagulova F, Petukhova G, Camerini-Otero R | 2014 | DMC1 SSDS ChIPseq from human testis | https://www.ncbi.nlm.nih.gov/geo/query/acc.cgi?acc=GSE59836 | NCBI Gene Expression Omnibus, GSE59836 |
| Davies B, Hatton E, Altemose N, Hussin JG, Pratto F, Zhang G, Hinch AG, Moralli D, Biggs D, Diaz R, Preece C, Li R, Bitoun E, Brick K, Green CM, Camerini-Otero RD, Myers SR, Donnelly P | 2016 | H3K4me3 ChIPseq from mice testis | https://www.ncbi.nlm.nih.gov/geo/query/acc.cgi?acc=GSE73833 | NCBI Gene Expression Omnibus, GSE73833 |
| Lange J, Yamada S, Tischfield SE, Pan J, Kim S, Zhu X, Socci ND, Jasin M, Keeney S | 2016 | SPO11 ChIPseq from mice testis | https://www.ncbi.nlm.nih.gov/geo/query/acc.cgi?acc=GSE84689 | NCBI Gene Expression Omnibus, GSE84689 |
| Brick K, Smagulova F, Khil P, Camerini-Otero RD, Petukhova GV | 2012 | DMC1 SSDS ChIPseq from WT B6 mouse testis | https://www.ncbi.nlm.nih.gov/geo/query/acc.cgi?acc=GSE35498 | NCBI Gene Expression Omnibus, GSE35498 |
| Altemose N, Noor N, Bitoun E, Tumian A, Imbeault M, Chapman JR, Aricescu AR, Myers SM | 2017 | PRDM9, H3K4me3 and H3K36me3 ChIPseq from HEK293 cells | https://www.ncbi.nlm.nih.gov/geo/query/acc.cgi?acc=GSE99407 | NCBI Gene Expression Omnibus, GSE99407 |
| Libertini E, Lebreton A, Lakisic G, Dillies MA, Beck S, Coppée JY, Cossart P, Bierne H | 2015 | whole-genome bisulfite sequencing (BS-seq) of HEK293 cells | https://www.ncbi.nlm.nih.gov/geo/query/acc.cgi?acc=GSE51867 | NCBI Gene Expression Omnibus, GSE51867 |

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
