## [Decision Letter]

**Acceptance summary:**

The identification of *Zcwpw1* is an important step in understanding the regulation of meiotic DSB repair, and the analysis performed both in mice and in cells provide novel information and complementary to those of the two other papers published on this same topic.

**Decision letter after peer review:**

Thank you for submitting your article "ZCWPW1 is recruited to recombination hotspots by PRDM9, and is essential for meiotic double strand break repair" for consideration by *eLife*. Your article has been reviewed by three peer reviewers, including Bernard de Massy as the Reviewing Editor and Reviewer #1, and the evaluation has been overseen by Jessica Tyler as the Senior Editor.

The reviewers have discussed the reviews with one another and the Reviewing Editor has drafted this decision to help you prepare a revised submission.

Summary:

The authors characterise a new protein, ZCWPW1, that is co-expressed with PRDM9 in mouse, a protein that has been shown to specify meiotic SPO11-DSB hotspot locations via the zinc-finger domain, and histone-trimethylation activity. The authors demonstrate the parallel phylogeny between ZCWPW1 and PRDM9. The authors analyse ZCWPW1 KO mice and show DSB repair and synapsis defects, with a proper DSB activity and localization. By assaying ZCWPW1 chromatin binding in HEK cells, the authors also show that ZCWPW1 has binding affinity for CpG rich DNA.

This study provides novel and interesting information about meiotic DSB control in mice. Most of the conclusions are convincing, however several interpretations required additional experimental support and the presentation of the data requires revision and clarification.

Importantly, as the authors have submitted their manuscript at the same time as the ones from Mahgoub et al. and Huang et al. , the results from these two papers have to be integrated in the Discussion to provide a more coherent view or to point out discrepancies if any — i.e. what is the meaning of the ZCWPW1 enrichment detected in HEK as compared to the signal detected in spermatocytes? What are the potential implications on the role of ZCWPW1 on gene and TE expression? What is the dynamic of RPA2 in ZCWPW1 KO?

Essential revisions:

1) Show the specificity of the antibody for ZCWPW1 vs ZCWPW2.

2) Revise cytology of ZCWPW1 KO: solve the discrepancy about DMC1 vs RPA2, and refer to other manuscripts for complementary data.

3) Revise interpretation about DSB repair phenotype: Conclusion should be toned down. ZCWPW1 may also affect DSB formation.

4) Clarify several points in data presentation (in particular HEK cells data, genomic data and figures). It is unfortunate that many figures are poorly prepared, difficult to understand and thus requiring a lot of time to read and evaluate.

5) Consider presenting DSB analysis by ChIP-seq DMC1 after the cytology results.

Other points:

1) General comment about mouse data presentation: for all data, provide age, number of mice and number of nuclei counted (complete the transparent reporting form accordingly).

2) General comment about NGS data: for all figures, make sure to label properly x and y axes with explanation of units and scales, explain color labels and how curves were calculated, explained which regions are being analysed.

3) Abstract: "persistent DMC1 foci, particularly those more strongly bound by PRDM9": This sentence is a very incorrect shortcut: persistence of DMC1 at specific hotspots has not been evaluated by cytology but by ChIP, and no direct correlation with strength of PRDM9 binding is presented.

4) Introduction final paragraph. There are many proteins that are positioned in response to DSBs generated at PRDM9-binding sites (e.g. MRE11, RAD51, DMC1 etc). Please revise.

5) Results first paragraph. It would help to have a clearer description of these data and how they relate to the graph in Figure 1B. What do the values on the Y-axis indicate?

6) Results paragraph two-four are hard to follow without small summary tables and/or simple bar plots or possibly pie-charts. Pointing at the large Table S1 was not helpful.

It is not fully clear if the six independent losses are birds + crocodiles, three groups of fishes, canids, amphibians; please clarify. Then, what is defined as the first four losses? Birds + crocodiles and fishes? What are these mutations? Are they predicted to be loss of function? The succession of comparisons of groups of species is confusing. What is the group discussed and mentioned as: "they mainly retain ZCWPW1"?. What is the need to add this statement at this point of the presentation of the results? In addition, it is not correct that this pattern is expected. The a priori expectation depends on whether ZCWPW1 is the only reader of those marks, for instance, it could also depend on whether this reading activity is essential.

7) Figure 2: The pachytene nucleus should be defined: if it is neither early nor late, it should be mid-?

Add antigen names on figure. Provide separate channels in supplement.

8) Paragraph two subsection “Localisation of ZCWPW1 in meiosis and details of asynapsis in infertile male *Zcwpw1^-/-^* mice” (with Jung reference). This is confusingly placed. I suggest opening this paragraph with this published information, then describe the new observations within the manuscript.

9) Figure 3—figure supplement 1B is referred to before Figure 3—figure supplement 1A.

10) The text mentions lack of MLH1 foci (Figure 3—figure supplement 2), yet no mention of these observations within this figure or elsewhere can be found in the manuscript. Please amend/clarify.

11) Please add relevant references to the PRDM9 phenotype/observations.

12) It is not the best definition of RAD51 to qualify it as a ssDNA binding protein. It is above all a strand exchange protein that binds to both ss and dsDNA (and three-strand DNA as well).

13) “ In contrast, there was no difference in the levels of RPA2 (Figure 4—figure supplement 2).”. This statement is inappropriate without clear evidence (and in contradiction with Li et al.). Please quantify RPA2 foci as graphs in a similar way to the DMC1 and RAD51 data. If unclear, the authors could also test MEIOB foci.

14) Figure 3E. I assume it is implicit that each value is for one mouse, and data are in triplicate or duplicate for *Prdm9*? Please clarify.

15) Figure 4. How were the categories Early Z, Z (MidZ?), Late Z defined?

16) Please aid reader by clarifying what asynapsed chromosomes are (HORMAD2-positive). Question: Sometimes unsynapsed is used, sometimes asynapsed is used. If there is a distinction, please clarify for the reader what is meant, or instead, please consolidate and choose a single term for clarity.

17) Please define FISH.

18) Figure S9. Why do most (but not all) groups overlap? (e.g. Chr 2, 2-3, 3-4, 4-5, 6-7, 7-8, 8.9, 16-17, 17-18, 18-19). Are observations sometimes placed in two bins, and sometimes not? This seems problematic and unnecessary to make the point.

19) Comparisons of synapsis between ZCWPW1 KO and *Prdm9* KO. An alternative hypothesis is for the distinct phenotype is a difference in the genetic background. ZCWPW1 is B6N and *Prdm9* is B6x129 hybrid probably with a certain number (?) of backcrosses. Subtle differences in checkpoint, cell cycle progression could affect the phenotype. Were mice compared at identical ages? Maybe complementary information could be available from other *Prdm9* deficient alleles. What about sterile hybrids?

20) Supplementary table 1 should have a legend, explain columns titles (for instance Y276, 341, 357, Max sum…)

21) Figure 2—figure supplement 2. Revise: This spread must be from WT not from *ZCWPW1^-/-^.*

22) ZCWPW1 staining is only at one of the two ends and it looks as if it is the centromeric end; please provide data to solve this ambiguity by labeling centromeres for instance. Also, as mentioned above, prove that this signal is not ZCWPW2.

23) Figure 3—figure supplement 1A. Where is the wild type? Wild type nuclei that have been treated in parallel and image acquisition with identical parameters must be shown.

24) Data not shown: please show data referred to in lanes 855 (western blots), 891 (IP with pre-immune serum).

25) Supplementary Figure 9: This analysis requires more documentation: representative images of the nuclei with immuno and FISH signals, and table of raw data (values for axis lengths and synapsis). Why are 18 and 19, or 18 and 17 not distinguished?

26) Figure 5—figure supplement 3. Clarify the legend title. It is unclear where the conclusion comes from. One assumes blue hexagons (actually misleading: the shape is invisible, just look like spheres) are *Prdm9* bound. I also understand the X axis is data from Zcpwp1 transfected alone. So, the binding at these non-*Prdm9* site does not change when *Prdm9* is co-transfected. Please clarify.

27) Are ZCWPW1, ZCWPW2 and *Prdm9* expressed in HEK cells? If yes, at which level and can it have consequence on interpretation of the data? Is the HA-ZCWPW1 fully functional?

28) Figure 5A. How is "Enrichment" calculated? It is mentioned with vs without PRDM9. Is it a ratio? Is this a log scale? For ZCWPW1 around human PRDM9 sites, it appears that the enrichment spans much more than just the peak region where PRDM9 binds. It would be extremely helpful to be able to compare these analyses to a similar aggregation of PRDM9 and H3K4me3 around the same sites. Y axis: homogenize: 2 or 2.0.

Why is the enrichment not down to 1 with *Prdm9* transfection?

Provide western blots to assay whether ZCWPW1 levels are identical in different analysis (with normalization for transfection efficiency).

Analyze quantitatively the correlation between ZCWPW1 and *Prdm9* binding.

29) Please clarify what is meant by "sequence-specific binding", and please describe how the data presented in Figure 5C and Figure 5—figure supplement 3 support this interpretation.

30) “ However, upon transfection, 91% (98%) of the strongest 10,132 (3,016) peaks are PRDM9 binding sites (cf. 7% expected overlap with randomised peaks), so PRDM9 is able to strongly reprogram ZCWPW1 binding, suggesting that non-PRDM9 peaks are bound more weakly.”. What does the value in parentheses (3,016) mean? This sentence as written is hard to follow. Please simplify.

31) Figure 5—figure supplement 5. Add labels to A and B panels.

Panel A: It seems that ZCWPW1 correlates poorly with cPRDM9. Why is this? What about transfecting chimp ZCWPW1? Do the same correlation plot with human proteins. Could there be some species-specific interactions, despite the nearly identical sequences of human and chimp proteins? Please provide the plot (heatmap/average plot) of H3K4me3 at cPRDM9 peaks.

Panel B: What is the difference with Figure 5B?

32) Paragraph three of subsection “ZCWPW1 is recruited to PRDM9 binding sites in an allele-specific manner”. Please expand this section so that the average reader can appreciate and understand the analysis that was performed. Figure 5—figure supplement 6 is hard to follow, please clarify. Where is the logistic regression methods?

33) You mean DMC1 sites in meiosis? Clarify to avoid confusion between transfected cells and meiotic cells.

34) “…although we show (see below) that ZCWPW1 is not directly involved in DSB positioning, this might suggest involvement of similar features to those it recognizes, in recruiting DSBs”. Is hard to follow. It will help if the last "it" was clarified to read "ZCWPW1".

35) Figure 6—figure supplement 1. Please label and explain the dashed horizontal lines.

36) The casual reader may be misled into thinking that the authors have tested for recruitment of ZCWPW1 to sites that have "dual" modifications. Unless I am mistaken, these are independent correlations to each mark, with no evidence that the histone marks exist coincidentally at the same locus in the same cell. Please clarify this section accordingly. Also, a direct interaction of PRDM9 with ZCWPW1 would be enough to explain specificity.

37) Figure 6—figure supplement 2. Please clarify the X-axis. Is the "+0.01" an offset added to each value in order to present on log scale? Please describe what analysis was performed and why.

38) Figure 6. Please clarify the analysis presented here. Do the 100bp windows cover the whole genome without other filtering? The legend is unintelligible, so it is very hard to interpret what is being presented. Untransfected means what: no PRDM9 and no ZCWPW1? "No untransfected H3K4me3" does not make sense. Both plots (red point and blue points) show ~2.5 fold increase from left to right (from 2 to 5 vs from 0.8 to 2). As such, is there really a difference if "predictive" power of the H3K4me3 at the red vs blue sites? Plotting the data on a Y-log scale (since enrichment is a ratio after all isn't it?) might help demonstrate this lack of difference more clearly. i.e. the blue curve only looks different to the red because it has been flattened by the linear Y scale.

Why is the distribution of H3K4me3 shifted to lower values in the red subset?

What does input fragment coverage mean? Do we agree that Input is chromatin before IP?

39) Figure S16. Please label panels. It was unclear what these data were attempting to test/demonstrate. Perhaps some manual annotations (and clarified legend and axes labels) would help? Explaining what the subsets are (and how they were obtained) is critical.

40) Figure 7C is referred to before 7B.

41) Figure 7A. The shaded labelling of 5' and 3' strands is unclear. Please replace with a clearer presentation style (different colours or dashed perhaps?). 7B presumably merges both strands, please clarify. Please homogenize X-axis labelling in A and B to be clearer (i.e. PRDM9 motif centre). Please use same units in both graphs (bp or kb).

42) Please show some raw DMC1 SSDS data from some example loci in order to demonstrate the lack of change ±ZCWPW1.

43) Paragraph two of subsection “DSBs occur at their normal locations in *Zcwpw1^-/-^* mice but show altered DMC1 persistence”. Broader? Based on what? Why is the DMC1 signal so irregular in the ZCWPW1 KO? What are the number of reads (Type I and Type II) in WT and KO?

44) Figure 7—figure supplement 1. This figure is very hard to understand. What does Heat>0 mean in legend? What does "Fraction of hotter hotspots" on Y-axis mean?

45) Figure 7C. Like many other figures, it is unclear what is being presented here. The legend has no explanation for the different colours (blue and black curves for example). The red/orange colour scale itself means what? What are the units? Why is the H3K4me3 relevant here?

In addition, it should be made to the reader that interpretation form this figure requires one additional assumption: i.e. that SPO11 oligos levels are not affected in ZCWPW1 KO. The authors should make clear that only WT data of SPO11 oligos is available.

46) Figure 7—figure supplement 2. Please label panels A, B, C. Please revise the text in the legend. It is very hard to follow. Adding manual annotations to the plots would help greatly – something as simple as: Hotspots relative to PRDM9 binding motif: upstream (red), downstream (black), central (green).

47) The Lange data would be more accurately described as SPO11-oligo seq.

48) Please refer to Figure 7C.

49) Figure 7—figure supplement 3. Please add panel labels. Please add units for axes. Are they log-scale? Please add wild type presented in exactly the same way as the two mutants for comparison.

It seems that the DMC1 data is interpreted as indication that ZCWPW1 promotes repair at hotter hotspots preferentially (the Abstract says: persistent DMC1 foci, particularly those more strongly bound by PRDM9, see also comment about this sentence). This conclusion is not convincing, ZCWPW1 KO may have the same effect as *Hop2* KO on DMC1 persistence (although the conclusion that the pattern is similar is an oversimplification, Y axis value are very different in the two strains): as a result of DSB repair defect, DMC1 level is proportional to DSB activity.

In addition, the authors cannot exclude that ZCWPW1 has a role on DSB activity and that DSB activity is increased in ZCWPW1 KO: it could be a direct effect: ZCWPW1 could compete with other readers, and/or indirect: synapsis defect allowing for a prolonged window for DSB activity with effects that may not be uniform across hotspots.

Accordingly, the statement, that DSB formation and positioning are completely unaffected is too strong.

50) Figure 7—figure supplement 4. Please present the other correlation plots that are mentioned in the text (DMC1 and SPO11) for comparison. Y-axis is a ratio so should be in log scale (or Log2 values)?

51) Please explain where the 9-fold effect is shown in Figure 7—figure supplement 4. Please clarify how the authors concluded this implies a "very strong effect" and a very strong effect of *what*? Of slower repair? Of H3K4me3? Of DMC1?

52) It is worth noting that the correspondence of ZCWPW1 occupancy with Alu repeats is also readily apparent in Figure 5C. And easier to visually interpret than the data presented in Figure 8.

53) Figure 5—figure supplement 2. Like many others, this figure needs revision. The labelling and presentation are poor and confusing: What utH3K4me3 means? In the middle, cells are cotransfected with ZCWPW1 and PRDM9 but peaks in red are ZCWPW1 alone? How is this possible?

54) Subsection “ZCWPW1 binds CpG dinucleotides” paragraph one. Please explain in the main text how the Figures 8A and Figure 5—figure supplement 2 support this statement.

55) Is it possible that the apparent Alu and CpG enrichment of ZCWPW1 by ChIP is due to lack of antibody specificity?

56) Use of "in vitro" is misleading. Instead ex vivo could be used, or in somatic cells.

57) Figure 8. Is this CpG binding still detected in the presence of *Prdm9*?

58) Extract the curve of ZCWPW1 enrichment at non-*Prdm9* sites in the presence of *Prdm9*, and test whether it follows the same distribution as in Figure 8A.

59) Discussion paragraph three: present differences with the cited paper and explain.

60) Discussion paragraph four. To quote HEK293 cells as a model system is very misleading. It is in no way a model system for meiosis. The authors have to indicate that this cell nuclei have a completely different environment as compared to meiotic cells. Therefore, extrapolation of any data to meiosis is only hypothetical, even it is coherent and follows some expectations.

61) ZCWPW1 does not directly recognize *Prdm9* binding sites: A *Prdm9* binding site is a DNA sequence.

62) Since the authors cite Spruce et al. paper they cannot write that ZCWPW1 is the first protein that recognizes regions bound by PRDM9.

63) Graphical abstract: how could the authors write that the functional importance of the marks has remained unknown? They have been shown to be essential for meiotic DSB formation.

---

## [Author Response]

Summary:[…]Importantly, as the authors have submitted their manuscript at the same time as the ones from Mahgoub et al. and Huang et al. , the results from these two papers have to be integrated in the Discussion to provide a more coherent view or to point out discrepancies if any — i.e.: what is the meaning of the ZCWPW1 enrichment detected in HEK as compared to the signal detected in spermatocytes? What are the potential implications on the role of ZCWPW1 on gene and TE expression? What is the dynamic of RPA2 in ZCWPW1 KO?

We thank the reviewers for their kind words, and we believe we have thoroughly addressed these substantive points and the specific points in our revised manuscript. All changes are tracked in the revised manuscript file.

As requested, we have now added a number of references to the related work on ZCWPW1 by Maghoub et al. and Huang et al. in the Results and Discussion sections, highlighting relevant evidence to integrate these studies into a coherent view:

i) “While we cannot rule out a direct interaction between ZCWPW1 and PRDM9, others have shown that dual modified peptides have a greater affinity to ZCWPW1 than peptides carrying the single modifications, supporting that the marks themselves are responsible for ZCWPW1’s recruitment (Mahgoub et al., 2020).”

ii) “Like Mahgoub and colleagues in a similar analysis (Mahgoub et al., 2020), we find co-evolution of ZCWPW1 with PRDM9, consistent with proteins functioning in the same pathway.”

iii) “In parallel with this work, two other studies of ZCWPW1 (Mahgoub et al., 2020; Huang et al., 2020) have reached complementary conclusions, and together with our own results allow some clear conclusions to be drawn.”

iv) “We find that in co-transfected HEK293T cells ZCWPW1 is strongly recruited to PRDM9-bound sites, in an allele-specific manner. Mahgoub et al., 2020, also report *PRDM9* allele-specific recruitment of ZCWPW1, but in hybrid mice carrying *Prdm9^Dom2^* and *Prdm9^Cast^* alleles which encode proteins with distinct zinc finger array DNA binding specificities. Together with ZCWPW1 binding measurements from Huang et al., 2020, these results imply that across *PRDM9* alleles and species – humans, mice, and likely chimpanzees – PRDM9 binding alone is sufficient to strongly recruit ZCWPW1 without requiring DSB formation (which does not occur in our system) or meiosis-specific co-factors (which would not be expected to be present in our cellular system).”

v) “In HEK293T cells we observed ZCWPW1 recruitment depends on the levels of H3K4me3 and H3K36me3 that PRDM9 deposits, while Huang et al., 2020, reported that in mice with a non-functional CW domain no ZCWPW1 binding peaks were observed, suggesting that H3K4me3 recognition at least might be necessary for the recruitment of ZCWPW1 at hotspots in vivo.”

vi) “Indeed in cells with PRDM9, the strongest ZCWPW1 binding sites are almost all PRDM9-bound sites. Similarly in mice, >85% of ZCWPW1 peaks overlap both marks concomitantly (Huang et al., 2020; Mahgoub et al., 2020). Mahgoub et al., 2020, additionally reported that in vitro binding affinity of recombinant mouse ZCWPW1 is at least 5 fold greater for dual-modified H3K4me3/K36me3 peptides compared to peptides carrying the single modifications.”

vii) “Although it was suggested that ZCWPW1 might recruit the DSB machinery to PRDM9 binding sites (M. Li et al., 2019; Spruce et al., 2020), we and Mahgoub et al., 2020, (using distinct assays) find that DSB positioning is completely unaffected by loss of ZCWPW1.”

viii) “Although more modest, Huang et al., 2020, did also report enrichment of ZCWPW1 at B1 elements in mice, the closest equivalent of human Alus. Unlike Mahgoub et al., 2020, Huang et al., 2020, also observed overlap of ZCWPW1 peaks with CpG islands (CGI) more frequently than expected. Given (i) they and Mahgoub et al., 2020, detect (Mahgoub et al., 2020) a much lower number of peaks (14,688 and 4,487 respectively) than we do in our HEK293T system (800,000), and (ii) we only find Alu enrichment in weaker ZCWPW1 peaks (of our top 10,000 peaks, >90% would instead be PRDM9-bound sites), these results are not conflicting, and instead might depend on detection/behaviour of weaker peaks, and/or presence of PRDM9 in ZCWPW1-containing cells.”

In their published manuscript Huang et al., 2020, conclude from their RNA-seq analysis of *Zcwpw1^-/-^* mice “These results strongly suggest that ZCWPW1 may not affect the transcription level of many genes sharing promoter overlap with ZCWPW1”, and so we have not commented on the implications of ZCWPW1 as a transcription co-factor as previously speculated by Huang et al. in their preprint.

See point 13 regarding the analysis of RPA2 dynamics in the *Zcwpw1* KO mouse.

Essential revisions:1) Show the specificity of the antibody for ZCWPW1 vs ZCWPW2.

We agree it is essential to exclude any potential cross-reactivity with ZCWPW2 of our polyclonal antibody. In Figure 2—figure supplement 1, we now include evidence (previously mentioned only as “data not shown”) that: (i) the pre-immune serum does not detect any signal in mid-zygotene or early pachytene cells strongly labelled by the ZCWPW1 antiserum by immunofluorescence (IF) in B6 mouse testis nuclear spreads (panel B); (ii) there is no detection of ZCWPW2 by IP and WB at the 38KDa molecular weight range predicted for mouse ZCWPW2, only a single band is observed at the expected molecular weight for ZCWPW1 (panel C). We also (iii) use HEK293T cells transfected with either FLAG-tagged mouse ZCWPW1 or ZCWPW2 to show detection of ZCWPW1 by IP and WB (panel D) and IF (panel E), but not ZCWPW2, by the antiserum (and no detection by the pre-immune serum), even at the high expression levels likely in this system.

Because we think it is likely ZCWPW2 (whose expression timing is similar to ZCWPW1) is still expressed in the *Zcwpw1* KO mouse, the complete loss of signal detected in IF staining by the ZCWPW1 antiserum in this mouse also suggests strong specificity to ZCWPW1.

The validation data are now referred to, in the relevant section, as: “To investigate the role of ZCWPW1 during meiosis in vivo, we produced and validated the specificity of an antibody against the full-length recombinant mouse protein (Figure 2—figure supplement 1), and … meiotic recombination.”

2) Revise cytology of ZCWPW1 KO: solve the discrepancy about DMC1 vs RPA2, and refer to other manuscripts for complementary data.3) Revise interpretation about DSB repair phenotype: Conclusion should be toned down. ZCWPW1 may also affect DSB formation4) Clarify several points in data presentation (in particular HEK cells data, genomic data and figures). It is unfortunate that many figures are poorly prepared, difficult to understand and thus requiring a lot of time to read and evaluate.5) Consider presenting DSB analysis by ChIP-seq DMC1 after the cytology results.

We address these points where they arise below.

Other points:1) General comment about mouse data presentation: for all data, provide age, number of mice and number of nuclei counted (complete the transparent reporting form accordingly).

We have revised the legend of all relevant figures (Figures 2-4, 7 and Supplements) to include this information which is also provided for each individual mouse used in the study in Figure 3—source data 1 and 2. For instance, Figure 2 legend now reads “Nuclear spreads from 9-10 weeks old B6 mice were immunostained with antibodies against ZCWPW1… These images are representative of the results obtained in 3 mice.” We have completed the transparent reporting form accordingly.

2) General comment about NGS data: for all figures, make sure to label properly x and y axes with explanation of units and scales, explain color labels and how curves were calculated, explained which regions are being analysed.

Thanks. The legends for Figures 1, 5 (and supplement 3 and 6), 6 (and supplements 1 and 2), 7 (and supplements 1, 2, 3 and 4), and 8 have been revised to include this information on the graphs and/or in the relevant legend. For example, we revised the legend of Figure 7C to add the sentences “Black dashed line is y=x for reference. SPO11 and DMC1 enrichment have been scaled by dividing by the mean autosomal enrichment. Large blue and dark red points show mean DMC1 signal binned into groups containing equal numbers of hotspots by WT SPO11 signal (vertical lines: corresponding 95% CIs), for X and autosomal data respectively.” in addition to revising the axis titles to specify “Scaled” and adding a legend for the (now simplified) colour scheme of the points.

3) Abstract: "persistent DMC1 foci, particularly those more strongly bound by PRDM9": This sentence is a very incorrect shortcut: persistence of DMC1 at specific hotspots has not been evaluated by cytology but by ChIP, and no direct correlation with strength of PRDM9 binding is presented.

We agree there are some caveats needed for this sentence, so due to lack of space in the Abstract we have simply deleted the phrase “particularly those more strongly bound by PRDM9”. We do think there is some evidence for this statement in the sense that we observe persistent foci by cytology but fairly normal break formation (both in the cytology and normal DMC1 ChIP-seq hotspot locations), implying an overall increase in DMC1 persistence. Then we observe a similar behaviour in DMC1 data for the X-chromosome and autosomes in the KO mouse, with a relative increase in those autosomal hotspots which are more active in DSB formation – which we would argue generally reflects stronger binding, and noting this phenomenon happens widely across all hotspots fairly similarly. It is then difficult not to conclude both (i) persistence increases overall for all hotspots, and (ii) persistent breaks occur particularly at the hotter hotspots (which will also contribute most of the breaks overall, i.e. cytologically visible signals of recombination). Indeed, many weaker hotspots become less detectable in our DMC1 ChIP-seq data in the KO mouse, so common, persistent breaks cannot be occurring at these hotspots even if the explanation is, they don’t break so often.

4) Introduction final paragraph. There are many proteins that are positioned in response to DSBs generated at PRDM9-binding sites (e.g. MRE11, RAD51, DMC1 etc). Please revise.

As suggested, we have edited this to read “directly positioned by PRDM9’s dual histone marks”.

5) Results first paragraph. It would help to have a clearer description of these data and how they relate to the graph in Figure 1B. What do the values on the Y-axis indicate?

We have added a note to the legend to properly define the Y-axis: “Jensen-Shanon divergence normalised to mean of 0 and standard deviation of 1 is shown on the y-axis (a measure of sequence conservation, see (Capra and Singh, 2007; Johansson and Toh, 2010))”.

6) Results paragraph two-four are hard to follow without small summary tables and/or simple bar plots or possibly pie-charts. Pointing at the large Table S1 was not helpful.It is not fully clear if the six independent losses are birds + crocodiles, three groups of fishes, canids, amphibians; please clarify. Then, what is defined as the first four losses? Birds + crocodiles and fishes? What are these mutations? Are they predicted to be loss of function? The succession of comparisons of groups of species is confusing. What is the group discussed and mentioned as: "they mainly retain ZCWPW1 "?. What is the need to add this statement at this point of the presentation of the results? In addition, it is not correct that this pattern is expected. The a priori expectation depends on whether ZCWPW1 is the only reader of those marks, for instance, it could also depend on whether this reading activity is essential.

This section has now been reworded throughout, to clarify the points raised. We have also added two new panels to Figure 1, showing bar plots summarising overlap between different groups of species and (i) whether PRDM9 is observed, when ZCWPW1 is present (panel C), and (ii) whether ZCWPW1 is observed, when PRDM9 is present and shows various independent evolutionary events (panel D). For the last point above, we wanted to relate the co-evolution data to the various domains of the two proteins and their predicted functions, but agree that the a priori expectation was clumsily worded, and it is now stated differently.

We now state that the likely “complete” losses (of PRDM9’s ability to catalyse H3K4me3/K36me3 deposition) are in the three groups of fishes (3 distinct losses), bird/crocodiles, amphibians, canids. We also now more clearly plot comparable loss events of other PRDM9 domains. We also show the loss events on the bar plot (the evolutionary history of PRDM9 is complex, involves many duplications and complete or partial losses of the protein, and there is considerable uncertainty for individual species, but we have attempted to summarize accurately the known events and overall patterns).

7) Figure 2: The pachytene nucleus should be defined: if it is neither early nor late, it should be mid-?Add antigen names on figure. Provide separate channels in supplement.

Figure 2 was revised as suggested, now indicating “mid-Leptotene”, “mid-Zygotene”, “mid-Pachytene” and mid-Diplotene stages, and the target proteins have been added directly to the panels. Separate channels are now provided for this staining in revised Figure 3—figure supplement 1B, and this is referred to in Figure 2 legend as “Images for the individual channels are provided in Figure 3—figure supplement 1”.

8) Paragraph two subsection “Localisation of ZCWPW1 in meiosis and details of asynapsis in infertile male Zcwpw1^-/-^ mice” (with Jung reference). This is confusingly placed. I suggest opening this paragraph with this published information, then describe the new observations within the manuscript.

The paragraph was revised according to the suggested structure.

9) Figure 3—figure supplement 1B is referred to before Figure 3—figure supplement 1A.

Thanks. We have revised this Figure (now Figure 3—figure supplement 1), swapping panels A and B to respect the chronological order of the data described in the text.

10) The text mentions lack of MLH1 foci (Figure 3—figure supplement 2), yet no mention of these observations within this figure or elsewhere can be found in the manuscript. Please amend/clarify.

We amended Figure 3—figure supplement 2, adding the missing data in panel B which shows lack of MLH1 foci in pseudo-pachytene cells in the *Zcwpw1* KO mouse.

11) Please add relevant references to the PRDM9 phenotype/observations.

We have added a reference to Hayashi et al., 2005.

12) It is not the best definition of RAD51 to qualify it as a ssDNA binding protein. It is above all a strand exchange protein that binds to both ss and dsDNA (and three-strand DNA as well).

We agree and have corrected the description to “a strand exchange protein”.

13) “In contrast, there was no difference in the levels of RPA2 (Figure 4—figure supplement 2).”. This statement is inappropriate without clear evidence (and in contradiction with Li et al.). Please quantify RPA2 foci as graphs in a similar way to the DMC1 and RAD51 data. If unclear, the authors could also test MEIOB foci.

We have now quantified RPA2 foci in the *Zcwpw1* KO mouse across the meiotic stages of prophase 1. The counts are presented as a graph in Figure 4—figure supplement 2B with the raw data in Figure 4—source data 1. In the light of the quantification results, the previous statement has been revised to: “Like DMC1, RPA2 levels were also significantly elevated in the *Zcwpw1^−/−^* mouse from zygotene onwards (Figure 4—figure supplement 2).”

14) Figure 3E. I assume it is implicit that each value is for one mouse, and data are in triplicate or duplicate for Prdm9? Please clarify.

Correct. We have added the clarification “each datapoint represents one mouse, each with n≥49 cells analysed.” to the legend. The corresponding raw data are presented in Figure 3—source data 3.

15) Figure 4. How were the categories Early Z, Z (MidZ?), Late Z defined?

Staging and sub- (early, mid and late) staging of spermatocytes during prophase I was carried out according to defined staining patterns against SYCP1 and SYCP3, and morphological criteria previously published (Gaysinskaya et al., 2014). We added more information about cell staging in section “Immunostaining of spermatocytes” of the Materials and methods.

16) Please aid reader by clarifying what asynapsed chromosomes are (HORMAD2-positive). Question: Sometimes unsynapsed is used, sometimes asynapsed is used. If there is a distinction, please clarify for the reader what is meant, or instead, please consolidate and choose a single term for clarity.

For consistency, the word “asynapsed” is now used across the manuscript. As suggested, we have also revised this sentence to read: “Indeed, we observe late unrepaired DMC1 foci mainly on HORMAD2-positive asynapsed chromosomes (Figure 4—figure supplement 3).”

17) Please define FISH.

We have expanded the FISH acronym (Fluorescent In-Situ Hybridisation).

18) Figure S9. Why do most (but not all) groups overlap? (e.g. Chr 2, 2-3, 3-4, 4-5, 6-7, 7-8, 8.9, 16-17, 17-18, 18-19). Are observations sometimes placed in two bins, and sometimes not? This seems problematic and unnecessary to make the point.

A single observation (one chromosome, in one cell) was only placed in one bin. However, because of the difficulty in measuring chromosome length with accuracy via microscopy, these bins overlapped several possible chromosomes. For simplicity, we have now decided to remove this analysis altogether.

19) Comparisons of synapsis between ZCWPW1 KO and Prdm9 KO. An alternative hypothesis is for the distinct phenotype is a difference in the genetic background. ZCWPW1 is B6N and Prdm9 is B6x129 hybrid probably with a certain number (?) of backcrosses. Subtle differences in checkpoint, cell cycle progression could affect the phenotype. Were mice compared at identical ages? Maybe complementary information could be available from other Prdm9 deficient alleles. What about sterile hybrids?

We agree differences in genetic background are a potential source of phenotypic variation, and indeed could explain partly or fully the apparent differences in meiotic defects observed between the two mutants. We have revised the paragraph to include this alternative. The mice from the 2 mutant lines were reasonably age-matched (within 2-3 weeks, see detail in Figure 3—source data 1). For the last point we agree, though time has not permitted us so far to perform a paired comparison of the meiotic phenotype of the *Zcwpw1* KO mouse with that of other *Prdm9* deficient strains (KRAB or SSXRD deletion mutants for instance), or mice with different combinations of alleles such as (PWDxB6) F1 hybrid infertile males where high levels of asymmetric PRDM9 binding are present. We hope to do so in future.

20) Supplementary table 1 should have a legend, explain columns titles (for instance Y276, 341, 357, Max sum…)

We did have some explanation of this in “Notes” within this table:

“Data are reproduced from Baker et al., 2017; for each species, we calculated the maximum domain architecture and sum of matching amino acids in the SET domain among all identified PRDM9 orthologues in that species (columns D and E)”. However, we have now reworded for increased clarity, and also added additional references to this note for each column separately, alongside retitling “Notes” to “Notes on table columns” in now Figure 1— source data 1 and 2.

21) Figure 2—figure supplement 2. Revise: This spread must be from WT not from ZCWPW1^-/-^.

It is indeed a wild-type sample. Note that this panel has been removed from the revised figure.

22) ZCWPW1 staining is only at one of the two ends and it looks as if it is the centromeric end; please provide data to solve this ambiguity by labeling centromeres for instance. Also, as mentioned above, prove that this signal is not ZCWPW2.

We do observe ZCWPW1 foci at both ends of the synaptonemal complex in many chromosomes, but not all. To quantify whether foci are at one, both ends or neither, in the revisions we quantified the number of chromosomes showing ZCWPW1 foci at either, or both, ends by FISH using telomeric and centromeric probes (revised Figure 2—figure supplement 2; raw data in Figure 2—source data 1). The results show that in some stages (late pachytene-diplotene) most chromosomes exhibit ZCWPW1 foci at both ends, while if foci (at least those observable via microscopy) are at one end only, this is mainly the centromeric end. We have revised the text now reading “Using Fluorescent In-Situ Hybridisation (FISH) to label centromeric and telomeric (distal and proximal to centromeres) regions of chromosomes, we established that these discrete foci of ZCWPW1 are consistently positioned at the ends of the synaptonemal complex, with the majority at both ends simultaneously, lying close to telomeres and centromeres themselves (Figure 2—figure supplement 2).” The specificity of the antibody for ZCWPW1 versus ZCWPW2 is now demonstrated in Figure 2—figure supplement 1.

23) SupFig4A. Where is the wild type? Wild type nuclei that have been treated in parallel and image acquisition with identical parameters must be shown.

Agreed; this dataset is now included in Figure 3—figure supplement 1B.

24) Data not shown: please show data referred to in lanes 855 (western blots), 891 (IP with pre-immune serum).

These data are now provided in Figure 2—figure supplement 1.

25) Supplementary figure 9: This analysis requires more documentation: representative images of the nuclei with immuno and FISH signals, and table of raw data (values for axis lengths and synapsis). Why are 18 and 19, or 18 and 17 not distinguished?

Chromosome ID was assigned based on the best match by predicted (Mb) length with a +/- 10% margin (calculated based on an inch to Mb ratio for chromosome 18 as a reference in the same cell), excluding chromosomes outside these boundaries. Where bins overlapped two possible chromosomes, these were grouped. For consistency across the analysis and the data presentation, we had retained (in the original Supplementary Figure 9) the grouping of chr17 or chr19 with chr18 (their closest chromosome in size) despite, in these instances, a potential ID as chr18 being excluded based on its identification by FISH. Due to the inaccuracy in length measurements, and likely high error rate in chromosome assignment, we have now removed this figure – see response to comment 18.

26) Figure 5—figure supplement 3. Clarify the legend title. It is unclear where the conclusion comes from. One assumes blue hexagons (actually misleading: the shape is invisible, just look like spheres) are Prdm9 bound.

We have added the clarification to the caption of what is now Figure 5—figure supplement 3: “(there are no peaks with high co-transfected enrichment [y-axis] when the untransfected enrichment [x-axis] is close to 0)”. Sorry about the figure resolution. The hexagons are clearer in the publication-quality image.

I also understand the X axis is data from Zcpwp1 transfected alone. So, the binding at these non-Prdm9 site does not change when Prdm9 is co-transfected. Please clarify.

The reviewer’s assumptions are correct. The red shows locations with no evidence of PRDM9 binding, and indeed these sites do not change much if at all when PRDM9 is co-transfected. The blue (locations with evidence of PRDM9 binding) shows many peaks above the diagonal hence showing that these sites do change when PRDM9 is co-transfected. i.e. some of the peaks from the ZCWPW1 transfected alone experiment do not change when PRDM9 is co-transfected (the ones which do not occur in locations with evidence of PRDM9 binding); but some of the peaks from the ZCWPW1 transfected alone experiment do change (those which have evidence of PRDM9 binding).

It is probably also worth pointing out that because ZCWPW1 binds at some level at many genomic sites, the sites of ZCWPW1 binding when transfected alone include a mixture of sites that are not bound by PRDM9, and sites that *are* bound by PRDM9, and both types are shown on this plot, not just sites not bound by PRDM9.

27) Are ZCWPW1, ZCWPW2 and Prdm9 expressed in HEK cells? If yes, at which level and can it have consequence on interpretation of the data?

All three genes have a highly restricted expression to reproductive glands in humans with comparatively low to no expression in kidney where the HEK293T cell line originates from (data from gtexportal PRDM9 https://www.gtexportal.org/home/gene/PRDM9; ZCWPW1 see Figure 1—figure supplement 1; ZCWPW2 https://www.gtexportal.org/home/gene/ZCWPW2). To ensure this is the case in our cell line stock, we have carried out a quantitative analysis of transcript expression by RT-PCR. The results presented in Figure 5—source data 1 show extremely low (*PRDM9* and *ZCWPW1*) to no (*ZCWPW2*) detectable gene expression in untransfected cells. Such levels are highly unlikely to confound the interpretation of the data in cells transfected with ZCWPW1 and/or PRDM9 where the expression is on average 800-1000 or 75,000-84,000 fold higher, respectively. Similarly, in a dataset from 10X genomics (https://support.10xgenomics.com/single-cell-gene-expression/datasets/1.1.0/293t) these three genes have very low, almost undetectable, expression in HEK293T cells (mean transcript count per cell of 0.036, 0.013, and 0.001, where median total transcript count per cell was 14,338). We added a statement in the relevant section of the results : “To test this, we co-transfected HEK293T cells … and then performed ChIP-seq against the ZCWPW1 tag (the endogenous levels of these proteins in HEK293T cells, and of the closely related family member ZCWPW2, are extremely low so unlikely to confound the analysis – see Figure 5—source data 1).”

**Author response image 1. sa2fig1:** 

Is the HA-ZCWPW1 fully functional?

We are uncertain of all the functional properties of this protein but note that we validated the predicted function of full length (HA-tagged) human ZCWPW1 as a dual histone methyl reader by demonstrating its preferential recruitment to PRDM9-bounds sites enriched in both histone marks. These results are in line with the expected behaviour. The C-terminal 9 amino acid small HA tag is located away from the internal CW and PWWP functional domains, and so is we hope unlikely to strongly confound the normal function of the protein. We note that others have specifically demonstrated binding of recombinant (His-tagged) mouse ZCWPW1 (residues 1-440 encompassing the CW and PWWP domains) to dual-modified H3K4me3/K36me3 peptides with a greater (over 5 fold) affinity compared to peptides carrying the single modifications in histone peptide assays in vitro (Mahgoub et al., 2020). We have added a reference to this work in the revised Discussion.

28) Figure 5A. How is "Enrichment" calculated? It is mentioned with vs without PRDM9. Is it a ratio? Is this a log scale?

We have added a clearer definition in the Materials and methods section which now reads “Each profile was normalised by the total coverage of all of the fragments and the normalised sample profile was divided by the relevant normalised comparison / control dataset to calculate enrichment.” It is a ratio and is presented on a log scale, which we have added a notification of in the legend as we agree with the point, that the axis breaks don’t make this clear: “Y-axis is log10 scale (y-axis labels remaining in linear space).”

For ZCWPW1 around human PRDM9 sites, it appears that the enrichment spans much more than just the peak region where PRDM9 binds. It would be extremely helpful to be able to compare these analyses to a similar aggregation of PRDM9 and H3K4me3 around the same sites. Y axis: homogenize: 2 or 2.0.

The profile plots (and heatmaps) of PRDM9 and H3K4me3 are shown in panel B of this figure. Indeed, they show a distance scale broadly consistent with binding of ZCWPW1 to H3K4me3 (and H3K36me3), but wider than binding by PRDM9 itself. (Given that the association between ZCWPW1 and chromatin likely reflects this indirect binding/recruitment, it is not clear in advance exactly what enrichment distance scale we expect.) Scale on Y axis is now consistent between panels.

Why is the enrichment not down to 1 with Prdm9 transfection?

The profile in Author response image 2, in orange, shows random sites and hits 1, so this is not simply an issue in normalisation. Instead, the higher background enrichment reflects higher regional (but not fully local) background effects, for example driven by some broad regions of the genome being more generally accessible, and other low level peaks also falling in the region surrounding one peak, on average. To demonstrate this, if we instead take locations randomly between 15kb and 10kb downstream of each peak (blue line) and calculate the profile there and overlay, we find that the locally higher background is consistent with the value the peak profile reaches.

**Author response image 2. sa2fig2:** Black: Profile of peaks. Blue: profile of locations 15 to 10kb downstream. Red: profile of globally (whole genome) random locations.

Provide western blots to assay whether ZCWPW1 levels are identical in different analysis (with normalization for transfection efficiency).

Unfortunately, we no longer have any material from these experiments to carry out the requested western blot quantification. At the time of the experiment, we verified similar transfection efficiency/levels of protein expression across the various experimental samples by immunofluorescence staining (on duplicate transfections). We have revised Figure 5—figure supplement 4 to include representative images of the transfected samples used in the study.

Analyze quantitatively the correlation between ZCWPW1 and Prdm9 binding.

We have calculated the correlation between PRDM9 enrichment and ZCWPW1 enrichment at sites of PRDM9 binding, and refer to this analysis as “ZCWPW1 shows a strong enrichment at human PRDM9 binding sites with higher enrichment at sites with higher PRDM9 enrichment (Figure 5A, B, Pearson's correlation = 0.43 – Figure 5—figure supplement 1)” where Figure 5—figure supplement 1 is now a scatter plot of PRDM9 enrichment against ZCWPW1 enrichment showing a linear effect.

29) Please clarify what is meant by "sequence-specific binding", and please describe how the data presented in Figure 5C and Figure 5—figure supplement 3 support this interpretation.

We have removed “sequence-specific” as we do not intend to imply a sequence motif per se which we suspect is what the reviewer is concerned about. We have also replaced the Figure 5—figure supplement 3 citation with Figure 5—figure supplement 2 to support the following new statement “Notably, even without PRDM9 we observed many ZCWPW1 binding peaks across the genome, some coinciding with PRDM9 binding sites (Figure 5C and Figure 5—figure supplement 2)”, and have moved the S10 (now Figure 5 —figure supplement 3) citation further down now reading “Hence, PRDM9 is able to strongly reprogram ZCWPW1 binding (Figure 5C, see also Figure 5—figure supplement 3), suggesting that peaks in cells without PRDM9 transfection are bound more weakly.”

30) “However, upon transfection, 91% (98%) of the strongest 10,132 (3,016) peaks are PRDM9 binding sites (cf. 7% expected overlap with randomised peaks), so PRDM9 is able to strongly reprogram ZCWPW1 binding, suggesting that non-PRDM9 peaks are bound more weakly.”. What does the value in parentheses (3,016) mean? This sentence as written is hard to follow. Please simplify.

Apologies. The parenthetical value corresponds to the 98% figure also in parentheses earlier in the sentence. We have re-written this sentence to clarify: “However, upon co-transfection with PRDM9, 92% of the strongest 10,000 ZCWPW1 peaks are at PRDM9-bound sites (98% of the strongest 3,000, vs 7% expected overlap with randomised peaks, and only 47% overlap for the strongest 3,000 ZCWPW1 peaks in cells without PRDM9 transfection).”

31) Figure 5—figure supplement 5. Add labels to A and B panels.

Added, thanks.

Panel A: It seems that ZCWPW1 correlates poorly with cPRDM9. Why is this?

Overall, ZCWPW1 in cells co-transfected with cPRDM9 still exhibits a large and clear enrichment at cPRDM9 sites, so we certainly do see allele-specific co-localisation. However, there are quantitative differences, with ZCWPW1 showing weaker enrichment for cPRDM9 than hPRDM9, in co-transfected cells. Although we do not know with certainty the reasons for this effect (which might operate differently in true meiotic cells with a different background chromatin environment), we have noted that cPRDM9 binds at sites with different broad-scale properties to hPRDM9, including more heterochromatic sites. Also, for example, cPRDM9 binding is not enriched at promoter regions. These properties might impact PRDM9’s deposition of histone modifications, and through this or another mechanism slightly weaken ZCWPW1’s ability to recognize bound sites.

What about transfecting chimp ZCWPW1?

We have not been able to explore chimp ZCWPW1 in this publication for reasons of time, but we agree it would indeed be an interesting avenue of research for the future.

Do the same correlation plot with human proteins.

We believe the reviewer means the heatmaps for human PRDM9 locations, which are provided as Figure 5B (see also comment below about panel B in reply to 31f).

Could there be some species-specific interactions, despite the nearly identical sequences of human and chimp proteins?

Although this is a good question, we don’t yet have data to answer it. Our working model is that the mechanism of action of ZCWPW1 is through the histone methylation marks deposited by PRDM9 rather than by directly binding PRDM9 itself – for example allowing flexibility in populations where PRDM9 is polymorphic. In this case the interaction might be expected to be quite similar for both human and chimp PRDM9. It is though possible that PRDM9 may directly bind ZCWPW1, or ZCWPW1 might have additional specificities (e.g. the CpG recognition) and/or recruit different proteins depending on human and chimp versions, but we have not explored this hypothesis.

Please provide the plot (heatmap/average plot) of H3K4me3 at cPRDM9 peaks.

H3K4me3 (without any transfection) at cPRDM9 peaks is shown in column 7 “Untransfected H3K4me3”. We do not have data available for H3K4me3 when co-transfected with cPRDM9, so have not been able to present this.

Panel B: What is the difference with Figure 5B?

Panel B shows the top 25% of all ZCWPW1 peaks (when co-transfected with human PRDM9). Figure 5B is focussed on PRDM9, so shows the top 25% of human PRDM9 peaks. There will be overlaps between these locations, but they are different in general.

32) Paragraph three of subsection “ZCWPW1 is recruited to PRDM9 binding sites in an allele-specific manner”. Please expand this section so that the average reader can appreciate and understand the analysis that was performed. Figure 5—figure supplement 6 is hard to follow, please clarify.

We have modified the section to clarify, it now reads: “Notably, the strength of ZCWPW1 binding in HEK293T cells at human PRDM9 binding sites provides a better predictor of human meiotic DMC1 binding status (a proxy for DSB formation) than does human PRDM9 binding strength itself (Figure 5—figure supplement 6). We show (see below) that ZCWPW1 is not directly involved in DSB positioning, this result might therefore instead suggest that similar features to those ZCWPW1 recognises are involved in recruiting DSB formation machinery.”

We also introduced additional details in the legend to interpret Figure 5—figure supplement 6 which now reads: “At identified human PRDM9 binding sites, we identified those at which male recombination hotspots occur, defined by the presence/absence of an overlapping human DMC1 peak, and fit a linear model to predict this hotspot status based on PRDM9 binding strength (PRDM9 Only), ZCWPW1 enrichment (with human PRDM9 vs without, referring to enrichment of ZCWPW1 cotransfected with PRDM9 relative to ZCWPW1 transfected alone), or both (see Materials and methods “DMC1 prediction”). […] Estimated PRDM9-dependent ZCWPW1 enrichment (green) provides a better predictor than does PRDM9 binding strength (blue).”

Where is the logistic regression methods?

The corresponding methods were under the title of “DMC1 prediction”. It was hard to find these, so we have added the term “logistic regression” in the Materials and methods section to make this section easier to find by searching as well as quoting the method section title in the caption of Figure 5—figure supplement 6 “(see Materials and methods “DMC1 prediction”)”.

33) You mean DMC1 sites in meiosis? Clarify to avoid confusion between transfected cells and meiotic cells.

Correct, we have added the clarification “human meiotic” here.

34) “…although we show (see below) that ZCWPW1 is not directly involved in DSB positioning, this might suggest involvement of similar features to those it recognizes, in recruiting DSBs”. Is hard to follow. It will help if the last "it" was clarified to read "ZCWPW1".

Thanks. Changed as suggested.

35) Figure 6—figure supplement 1. Please label and explain the dashed horizontal lines.

We have added the phrase “Dotted lines show overall means for each colour.” to the legend.

36) The casual reader may be misled into thinking that the authors have tested for recruitment of ZCWPW1 to sites that have "dual" modifications. Unless I am mistaken, these are independent correlations to each mark, with no evidence that the histone marks exist coincidentally at the same locus in the same cell. Please clarify this section accordingly. Also, a direct interaction of PRDM9 with ZCWPW1 would be enough to explain specificity.

Agreed (in principle). We have added the caveat “tested separately, not necessarily “dual” marks coincident within an individual cell, although locations of these marks are rarely coincident except at hotspots (Powers et al., 2016)”.

37) Figure 6—figure supplement 2. Please clarify the X-axis. Is the "+0.01" an offset added to each value in order to present on log scale? Please describe what analysis was performed and why.

We have added clarification to the plot legend of this figure: “This is in some sense opposite (but complementary) to Figure 6—figure supplement 1 in which the subject of the axis is reversed.” and “0.01 has been added to the x-axis values in order to display enrichment estimates of zero on the log scale.”

38) Figure 6. Please clarify the analysis presented here. Do the 100bp windows cover the whole genome without other filtering?

Correct, the only filtering applied is that which is already stated in the caption “defined as indicated in the legend… with the additional constraint of requiring input fragment coverage >5 for ZCWPW1 and >15 for H3K4me3.” To eliminate and check for e.g. X-chromosome artifacts we have additionally re-run this analysis excluding the sex chromosomes, resulting in a near-identical plot.

The legend is unintelligible, so it is very hard to interpret what is being presented. Untransfected means what: no PRDM9 and no ZCWPW1?

Correct: untransfected means no transfection with either PRDM9 or ZCWPW1. We have changed this in the legend to “pre-existing”, see next response.

We have also added the following sentence to the caption “(where “p” is the p-value from peak-calling required for a window to be included in the subset)”.

"No untransfected H3K4me3" does not make sense.

We have changed this to “No pre-existing H3K4me3” and hope that this is clearer (it means no H3K4me3 is seen in untransfected cells). An alternative would be “No PRDM9-independent or ZCWPW1-independent H3K4me3”, we settled on the former and added an explanation to the legend: “Pre-existing H3K4me3 refers to H3K4me3 that is present without transfection (of either PRDM9 or ZCWPW1), which is mainly found at promoter regions.”

Both plots (red point and blue points) show ~2.5 fold increase from left to right (from 2 to 5 vs from 0.8 to 2). As such, is there really a difference if "predictive" power of the H3K4me3 at the red vs blue sites? Plotting the data on a Y-log scale (since enrichment is a ratio after all isn't it?) might help demonstrate this lack of difference more clearly. i.e. the blue curve only looks different to the red because it has been flattened by the linear Y scale.

It may be that there is a misunderstanding here: we are not claiming that there is a difference in the slope, i.e. the fold-impact of increasing H3K4me3 on relative ZCWPW1 presence. Instead, we are noting the *absolute* level of ZCWPW1 is much higher, for any given level of H3K4me3, at PRDM9 binding sites i.e. that the “intercept” is higher for PRDM9 bound regions. The parallel behaviour emphasizes that this property remains for any level of H3K4me3. The relevant variable is the presence of PRDM9 (perhaps mediated by H3K36me3 or another mark), and not H3K4me3 which is similarly present in both subsets. We have as suggested replaced the plot with a y-log version although as explained above our interpretation of the data is unchanged: the red and blue subsets are still almost completely separated by the black dotted line. We have added the following sentence to the legend which we hope will guide the reader on the interpretation of this plot “For any given level of H3K4me3 (x-axis), ZCWPW1 enrichment (y-axis) is higher at PRDM9 bound regions (red) than regions with pre-existing H3K4me3 (promoters, blue).”

Why is the distribution of H3K4me3 shifted to lower values in the red subset?

The blue subset is, in effect, promoter regions – which have very high H3K4me3 levels (plausibly close to ~100% of some histones being marked at the highest levels). The red subset has H3K4me3 only due to PRDM9 binding nearby, which is limited by both PRDM9 binding strength and also our achieved transfection efficiency, so in practice appears unable to achieve the same level.

What does input fragment coverage mean?

Fragment refers to the DNA fragment that was sequenced as inferred by the two ends of paired end sequencing, and coverage is the number of reads covering a given position.

Do we agree that Input is chromatin before IP?

Yes, Input refers to the starting material (crosslinked, sonicated chromatin from lysed cells) before IP.

39) Figure S16. Please label panels. It was unclear what these data were attempting to test/demonstrate. Perhaps some manual annotations (and clarified legend and axes labels) would help? Explaining what the subsets are (and how they were obtained) is critical.

For simplicity, we have just removed this faceted figure – as it made a similar point to Figure 6.

40) Figure 7C is referred to before 7B.

We have deleted this additional forward reference, as it is not essential.

41) Figure 7A. The shaded labelling of 5' and 3' strands is unclear. Please replace with a clearer presentation style (different colours or dashed perhaps?).

We have changed to use dotted lines as suggested.

7B presumably merges both strands, please clarify.

Yes – we have added “(Both strands combined)” to the legend of Figure 7B.

Please homogenize X-axis labelling in A and B to be clearer (i.e. PRDM9 motif centre).

Thanks – we have homogenized the axis labels to say “Position relative to PRDM9 motif (kb)”.

Please use same units in both graphs (bp or kb).

Done.

42) Please show some raw DMC1 SSDS data from some example loci in order to demonstrate the lack of change ±ZCWPW1.

In Author response image 3 are four representative regions among those with high coverage, showing identical general patterns. We believe it’s not clear what this figure adds in addition to Figure 7, because this figure shows the lack of change without the additional noise when looking at individual hotspots. However, we could add this as a supplement if the reviewer thinks it would be helpful having seen Author response image 3:

**Author response image 3. sa2fig3:** 

43) Paragraph two of subsection “DSBs occur at their normal locations in Zcwpw1^-/-^ mice but show altered DMC1 persistence”. Broader? Based on what?

We have removed the broader comment here, as this is discussed with more precision in the following paragraph “slightly wider […] (Figure 7B).”

Why is the DMC1 signal so irregular in the ZCWPW1 KO? What are the number of reads (Type I and Type II) in WT and KO?

As guessed by the reviewer the KO DMC1 signal has more variance, due to the lower number of both read types obtained (< ½), from a single infertile mouse:

44) Figure 7—figure supplement 1. This figure is very hard to understand. What does Heat>0 mean in legend?

As per the legend, heat refers to DMC1 signal, so Heat>0 therefore refers to hotspots with a DMC1 signal above 0. We have relabeled to “DMC1>0” for ease of understanding of this figure.

What does "Fraction of hotter hotspots" on Y-axis mean?

We have changed the axis titles to be clearer as follows. Y: “Fraction of WT hotspots falling in each KO DMC1 bucket for all hotspots with WT DMC1 ≥ a given x-axis value” and X: “WT hotspot DMC1 heat relative to strongest”. We also added the following explanation to the caption to guide the reader in how to read the plot: “Y-axis values at x = 0 show the fraction of all hotspots falling into the buckets shown in the inset colour legend. […] Therefore, almost all WT hotspots with activity >20% of the hottest hotspot are observed, and non-observed hotspots show only weak activity in WT, and so our power to detect them is expected to be reduced.”

What this figure shows is that although we do not detect all hotspots present in the WT mouse in the KO, we do detect many, and almost all the high-heat hotspots, suggesting those not observed are explained by being weaker and fewer reads meaning we lack power in the KO.

45) Figure 7C. Like many other figures, it is unclear what is being presented here. The legend has no explanation for the different colours (blue and black curves for example). The red/orange colour scale itself means what? What are the units? Why is the H3K4me3 relevant here?

Apologies: we have simplified this figure by removing H3K4me3 colouring, so the colours of the points representing the binned means are more easily pairable with the data points. We have also added a legend directly to the figure to label the autosomal vs X chromosome points. We have also added the following to the caption:

“Unlike WT mice, DMC1 signals in *Zcwpw1^-/-^* mice are approximately linearly associated with WT SPO11. […] Large blue and dark red points show mean DMC1 signal binned into groups containing equal numbers of hotspots by WT SPO11 signal (vertical lines: corresponding 95% CIs), for X and autosomal data respectively.”

In addition, it should be made to the reader that interpretation form this figure requires one additional assumption: i.e. that SPO11 oligos levels are not affected in ZCWPW1 KO. The authors should make clear that only WT data of SPO11 oligos is available.

We agree and have added the following clarification to the main text “assuming SPO11 levels are not greatly affected by ZCWPW1 KO”.

46) Figure 7—figure supplement 2. Please label panels A, B, C. Please revise the text in the legend. It is very hard to follow. Adding manual annotations to the plots would help greatly – something as simple as: Hotspots relative to PRDM9 binding motif: upstream (red), downstream (black), central (green).

We have added panel labels and reworded the legend including adding the suggested phrase to Figure 7—figure supplement 2.

47) The Lange data would be more accurately described as SPO11-oligo seq.

We have corrected this.

48) Please refer to Figure 7C.

We have added the reference.

49) Figure 7—figure supplement 3. Please add panel labels. Please add units for axes. Are they log-scale? Please add wild type presented in exactly the same way as the two mutants for comparison.

We have added panel labels, explained the scaling in the legend ("SPO11 and DMC1 enrichment have been scaled by dividing by the mean autosomal enrichment.”) and added a WT version as panel C to Figure 7—figure supplement 3.

It seems that the DMC1 data is interpreted as indication that ZCWPW1 promotes repair at hotter hotspots preferentially (the Abstract says: persistent DMC1 foci, particularly those more strongly bound by PRDM9, see also comment about this sentence). This conclusion is not convincing, ZCWPW1 KO may have the same effect as Hop2 KO on DMC1 persistence (although the conclusion that the pattern is similar is an oversimplification, Y axis value are very different in the two strains): as a result of DSB repair defect, DMC1 level is proportional to DSB activity.

We think the alternative hypothesis suggested by the reviewer “ZCWPW1 KO may have the same effect as *Hop2* KO […] as a result of DSB repair defect, DMC1 level is proportional to DSB activity.” is exactly what we are proposing. It’s because in the *absence* of a DBS repair defect hotter hotspots are repaired faster (in a non-linear manner, Figure 7C) (i.e. *normally* there is a non-linear effect), so if interpreted as a delay or failure in repair increasing DMC1 signal (as observed via microscopy also), the difference/fold-effect of the KO is greater at the hottest hotspots. The y-axis scaling is arbitrary (based on number of ChIP-seq reads, for the previously published *Hop2* KO data), and so does not impact this interpretation.

In addition, the authors cannot exclude that ZCWPW1 has a role on DSB activity and that DSB activity is increased in ZCWPW1 KO: it could be a direct effect: ZCWPW1 could compete with other readers, and/or indirect: synapsis defect allowing for a prolonged window for DSB activity with effects that may not be uniform across hotspots.Accordingly, the statement, that DSB formation and positioning are completely unaffected is too strong.

We have removed “formation” so it currently reads “DSB positioning is completely unaffected”.

50) Figure 7—figure supplement 4. Please present the other correlation plots that are mentioned in the text (DMC1 and SPO11) for comparison. Y-axis is a ratio so should be in log scale (or Log2 values)?

We have added these new plots as additional panels to Figure 7—figure supplement 4. We have also used log scales for these plots (although the relationship appears no longer linear on this scale).

51) Please explain where the 9-fold effect is shown in Figure 7—figure supplement 4.

We have scaled each axis by dividing by the median, to make the 9 fold effect more obvious. Now the mean ratio starts at 1 on the left (y-axis) and increases to around 9 on the right.

Please clarify how the authors concluded this implies a "very strong effect" and a very strong effect of what? Of slower repair? Of H3K4me3? Of DMC1?

A “very strong effect” is based on the 9 fold change (almost an order of magnitude) which seems reasonably described as very strong.

We have added the clarification “of PRDM9’s “assistance” in aiding repair”, regarding the effect referred to.

52) It is worth noting that the correspondence of ZCWPW1 occupancy with Alu repeats is also readily apparent in Figure 5C. And easier to visually interpret than the data presented in Figure 8.

We thank the reviewer(s) for pointing this out, and we have additionally referenced 5C to aid the reader.

53) Figure 5—figure supplement 2. Like many others, this figure needs revision. The labelling and presentation are poor and confusing: What utH3K4me3 means?

We have modified Figure 5—figure supplement 2 legend by expanding all abbreviations rather than explaining them in the caption – using "Pre-existing” H3K4me3 instead of “untransfected”, and visually structuring the colour categories. We have also updated the caption to provide a more explicit description of what the panels and colours within show, now reading “For example dark green peaks are those which overlap with ZCWPW1 peaks when transfected alone, but not overlapping Human PRDM9 peaks, and not overlapping pre-existing H3K4me3 peaks but do overlap with Alu repeats. Pre-existing H3K4me3 refers to peaks found without PRDM9 or ZCWPW1 transfection.” and “In cells expressing ZCWPW1 in the presence (middle plot) but not in the absence (left plot) of PRDM9, the strongest peaks are dominated by PRDM9-bound sites marked by H3K4me3 (pink), while ZCWPW1 occupancy increases occur nearly exclusively at these sites, following co-transfection with PRDM9 (right plot).”

In the middle, cells are cotransfected with ZCWPW1 and PRDM9 but peaks in red are ZCWPW1 alone? How is this possible?

The graph shows “Proportion of ZCWPW1 peaks, ordered by enrichment of ZCWPW1 binding over input, overlapping various other marks”; only the light blue category represents peaks “Not [in] ZCWPW1 Alone” i.e. the co-transfected peaks are almost completely a subset of the “ZCWPW1 Alone” peaks (to rephrase the reviewers interpretation, the peaks in red are *also* present in “ZCWPW1 Alone”, see also Figure 5—figure supplement 3) resolving the impossibility.

54) Subsection “ZCWPW1 binds CpG dinucleotides” paragraph one. Please explain in the main text how the Figures 8A and Figure 5—figure supplement 2 support this statement.

We looked at this – the Y-axis of Figure 8A directly shows “Fraction of ZCWPW1 peaks that overlap Alu repeats” with values of greater than 75% for weaker ZCWPW1 peaks e.g. enrichment <1 (much greater than expected by chance, as stated in the following sentence). Similarly, the y-axis of Figure 5—figure supplement 2 shows the fraction of ZCWPW1 peaks overlapping various marks, including Alus (in dark green). So, we think this is clear, but would be happy to include a specific edit if preferred.

55) Is it possible that the apparent Alu and CpG enrichment of ZCWPW1 by ChIP is due to lack of antibody specificity?

The anti-HA antibody from Abcam (Ab9110) used for ChIP against (HA-tagged) ZCWPW1 is a highly specific and widely used ChIP-grade antibody. The negative control ChIP anti-HA in untransfected HEK293T cells for this experiment did not yield DNA in quantifiable amounts (below the range of detection of the Qubit dsDNA high sensitivity assay), therefore we believe ZCWPW1 does indeed (often weakly) bind Alus and CpGs.

56) Use of "in vitro" is misleading. Instead ex vivo could be used, or in somatic cells.

We have replaced this with “in HEK293T cells”. (We did not opt for “ex vivo” as this is preferred by some to only be used for experiments performed on *primary tissue* outside of the organism.)

57) Figure 8. Is this CpG binding still detected in the presence of Prdm9?

In these experiments, yes. However, we note that as they represent an average of many cells, with variable transfection efficiency, some cells have PRDM9 and some don’t. It may then be that in individual cells with PRDM9, ZCWPW1 is very strongly sequestered to PRDM9 sites away from non-hotspot CpG sites.

58) Extract the curve of ZCWPW1 enrichment at non-Prdm9 sites in the presence of Prdm9, and test whether it follows the same distribution as in Figure 8A.

In Author response image 4 we have calculated the curve both in the presence of PRDM9 (left column) and in absence (right column), as well as at all sites (top row) and at sites not bound by PRDM9 (bottom row) [as even in the absence of PRDM9 many ZCWPW1 binding sites overlap PRDM9 binding sites]. The curves all show effectively the same thing which is expected given the large overlap of site positions.

**Author response image 4. sa2fig4:** 

59) Discussion paragraph three: present differences with the cited paper and explain.

In the revised manuscript, these differences are now discussed.

60) Discussion paragraph four. To quote HEK293 cells as a model system is very misleading. It is in no way a model system for meiosis. The authors have to indicate that this cell nuclei have a completely different environment as compared to meiotic cells. Therefore, extrapolation of any data to meiosis is only hypothetical, even it is coherent and follows some expectations.

We have revised this paragraph to give a clearer and more accurate description of this cellular system and its intended use. Given the mitotic nature of HEK293T cells, of course we never meant to claim these could be considered a model system of *meiosis* (as a process). However, they provide a suitable cellular system in which to study the *direct* DNA and histone binding properties of ectopically-expressed meiotic proteins, which are our focus in those experiments. The absence of a large number of meiotic proteins within these cells is consistent with the idea that recruitment of ZCWPW1 might require only PRDM9 and certainly rules out their necessity, and so in that sense is advantageous. In practice we do observe a strong overlap between human meiotic recombination hotspots and binding sites of both PRDM9, and (more strongly) sites where ZCWPW1 is present, attesting that though very different, mitotic cells can nevertheless give useful insights into molecular properties of these proteins.

61) ZCWPW1 does not directly recognize Prdm9 binding sites: A Prdm9 binding site is a DNA sequence.

We have changed this to “PRDM9-marked sites”, making reference to the histone marks more explicit.

62) Since the authors cite Spruce et al. paper they cannot write that ZCWPW1 is the first protein that recognizes regions bound by PRDM9.

We are making a distinct claim here. HELLS is required for stable PRDM9 binding and recognises PRDM9 itself, independently of whether PRDM9 is bound to DNA. Once PRDM9 has bound however, ZCWPW1 is the first protein which recognises these PRDM9 marked sites. To clarify this we have reworded to “ZCWPW1 likely directly recognises sites marked by the dual histone marks deposited by the PRDM9-HELLS complex (Spruce et al., 2020), the first such protein identified.”

63) Graphical abstract: how could the authors write that the functional importance of the marks has remained unknown? They have been shown to be essential for meiotic DSB formation.

Apologies for this slip – we have changed this to state “but the mechanistic role of these marks”, because the protein(s) recognizing the marks have remained unclear.